# Elucidating the reaction mechanism of a palladium-palladium dual catalytic process through kinetic studies of proposed elementary steps

Anže Ivančič [1], Janez Košmrlj [1] & Martin Gazvoda [1✉]

In the synergistic dual catalytic process, the kinetics of the catalytic cycles must be balanced for the successful outcome of the reaction. Therefore, the analysis of the kinetics of the independent catalytic cycles is essential for such reactions, as it enables their relational optimization as well as their design. Here we describe an analysis of the mechanism of a catalytic synergistic bimetallic reaction through the experimental study of a palladium-catalysed cross-coupling of aryl halides with terminal alkynes, an example of a monometallic dual catalytic process. The proposed mechanism of the investigated reaction was disassembled into two palladium catalytic cycles and further into elementary reactions, and each step was studied independently. The described mechanistic analysis allowed us to identify the rate-determining step of the catalytic process by comparing the rates of the elementary reactions under similar reaction conditions, balanced kinetics of the palladium catalytic cycles, and also in which step which reagent enters the catalytic cycle and how.

[1] University of Ljubljana, Faculty of Chemistry and Chemical Technology, Ljubljana, Slovenia. ✉email: martin.gazvoda@fkkt.uni-lj.si

Methods for analysing the mechanisms of catalytic reactions are usually based on the kinetic analysis of the reaction under synthetically relevant conditions, i.e., by reactions of starting reagents and (pre)catalysts[1,2]. Advanced methods of this type, such as reaction progress kinetic analysis (RPKA)[3] and variable time normalization analysis (VTNA)[4,5], lead to these conclusions with a minimal number of experiments required. However, the problem of elucidating mechanistic pathway becomes apparent when complex reactions based on two or more interconnected catalytic cycles are involved. Such reactions include so-called synergistic dual catalysis, also known as bimetallic catalysis or cooperative catalysis, in which both starting substrates are activated in situ with two separate metal catalysts to form two reactive organometallic species that combine in the transmetallation step[6,7]. Elucidating the mechanism of such a reaction is even more difficult when the same metal activates two different substrates and mechanistic analysis of such a reaction using the above methods cannot shed light on the reaction pathway (vide infra).

Palladium-catalysed cross-coupling reactions have transformed the field of synthetic chemistry and the way molecules are made[8–10]. Synergistic dual catalysis, in which both coupling partners are activated in situ without the need for stoichiometric amounts of either metal, an alternative to traditional mono-catalytic cross-coupling, is emerging as a powerful synthetic tool[6,7]. The seminal example of synergistic dual catalysis is the Sonogashira reaction, in which catalytic amounts of palladium and copper precatalysts are used to activate aryl halide and terminal alkyne, respectively[11,12]. Coincident with the Pd-Cu reaction, the copper-free variant was independently described by Cassar and Heck in 1975 (Fig. 1a)[13,14]. The use of copper as a co-catalyst, while beneficial for efficacy, has several disadvantages, namely the use of environmentally unfriendly reagents, the formation of undesirable alkyne homocoupling by-products, and the need for oxygen exclusion, which is why the copper-free version has great potential[12,15]. A monometallic mechanism, similar to that of the Heck reaction[16] and Buchwald-Hartwig amination[17,18] was initially postulated for this reaction, in which there is no transmetallation step but ligand exchange in the palladium oxidative addition complex (Fig. 1b)[19]. Although several experimental and theoretical studies have been performed to support this mechanism[20–28], many questions remained unanswered. Experimental mechanistic studies have been performed under synthetically relevant conditions[21,24,28] using various methods and techniques for mechanistic analysis, including trapping of the proposed intermediates[21], competitive studies with different starting substrates[21,24] and attempted elucidation of reaction pathway by MS using labelled starting reagents[28], but have not unambiguously confirmed the mechanistic proposal.

As an alternative to this monocyclic mechanism, we have recently shown that two separate palladium catalytic cycles connected in a palladium-palladium transmetallation step[29–31] may also operate (Fig. 1c)[32]. In this scenario, one palladium pre-catalyst activates the (het)aryl halide and another activates the terminal alkyne. This mechanistic proposal places the copper-free Sonogashira reaction (Heck-Cassar alkynylation) among bimetallic catalytic reactions in which the same metal serves two distinct functions[33–35]. To our knowledge, there have been very few reports postulating this type of catalysis, all based on the palladium-palladium system[36–39], i.e., the arylation of pyridine N-oxide with aryl bromides[36], the cyanation of aryl bromides with HCN[37], the aerobic oxidative coupling of arenes[38], and the carbonylative Sonogashira cross-coupling[39].

Understanding the mechanism of a reaction is critical to its development. The postulated dual catalysis in case of palladium-catalysed coupling of aryl halides and terminal alkynes[32] enabled us to improve the synthesis protocol by simultaneously introducing two different palladium pre-catalysts[40]. As described above, traditional mechanistic analyses have failed to elucidate the mechanistic pathway for such a complex reaction. Alternatively, the reaction mechanism can be disassembled into postulated separate catalytic cycles or even further into elementary steps that can be studied independently using traditional methods of kinetic analysis. However, for this type of analysis, the proposed reaction intermediates should be stable enough to be isolated or selectively formed in situ. This approach has been used sporadically to confirm and understand key steps[41], i.e., oxidative addition[42,43], transmetallation[44–47], and reductive elimination[48], of the mechanisms of cross-coupling reactions. It is noteworthy that a mechanistic disassembly and independent kinetic analysis of each of the elementary steps for a synergistic bimetallic catalytic process as described herein has not been reported before.

Here, the proposed mechanism was divided into elementary steps and investigated using independently prepared reaction intermediates (Fig. 1d). The mechanistic analysis described allowed us to identify the rate-determining step of the catalytic process by comparing the rates of elementary reactions under similar reaction conditions, balanced kinetics of the palladium catalytic cycles, and also in which step which reagent enters the catalytic cycle and how.

## Results and discussion

**Organometallic nucleophile (palladium bisacetylide) (re)formation.** We started our mechanistic analysis with attempt to describe organometallic nucleophile (re)formation in the proposed 'Pd catalytic cycle II' in which palladium acetylides are generated (Fig. 2a). For this reason, we divided the proposed 'Pd catalytic cycle II' into two elementary reactions (Fig. 2b, c). The first reaction describes the initial formation of $C$ from the $Pd^{II}X_2$ species usually present in the reaction mixture (Fig. 2c) and the second reaction describes the regeneration of the palladium bisacetylide $C$ from the monoacetylide $D$ derived in transmetallation (Fig. 2b). Apart from our earlier reports[32,40], palladium bisacetylides have not been described in the context of cross-coupling catalysis. There has been considerable interest in (transition)metal based acetylides[49,50], while there are few reports on palladium counterparts[51–58]. We have previously described that under identical conditions the reaction of palladium oxidative addition complex $A$ with palladium bisacetylide $C$ into tolan product 3 occurs instantly, whereas induction period was present for the reaction of palladium oxidative addition complex $A$ with terminal alkyne 2 in the presence of base, in which palladium acetylides are formed to initiate the reaction[32].

For the species mimicking intermediates $D$, we chose palladium complexes 5 with archetypal triphenylphosphine (PPh$_3$) ligands as shown in Fig. 3. The required iodoalkynes[59], bromoalkynes[60] and chloroalkynes[61] were prepared using the modified literature procedures. Their reactions[62] with Pd(PPh$_3$)$_4$ in benzene as solvent at room temperature gave various iodide (**5a, 5d, 5e, 5 h, 5j, 5k**), bromide (**5b, 5 f, 5i**) and chloride (**5c, 5 g**) (het)aryl or alkyl palladium acetylides in 19–81% isolated yields (Supplementary Note 1). The synthesized 5 served as mimics for the structures $D$ of the proposed mechanism, as well as diagnostic tools for the subsequent monitoring of transmetallation reactions. They were stable in CDCl$_3$ solutions for about 1 h at room temperature, after which they began to slowly decompose as shown by $^1$H and $^{31}$P NMR. The isolated samples of 5 were stable under argon atmosphere in the freezer at −20 °C for about 7 days. Despite this, compounds 5 were characterized by IR, HRMS and NMR ($^1$H, $^{13}$C, $^{31}$P) (Supplementary Methods, Supplementary Note 1 and Supplementary Data 1).

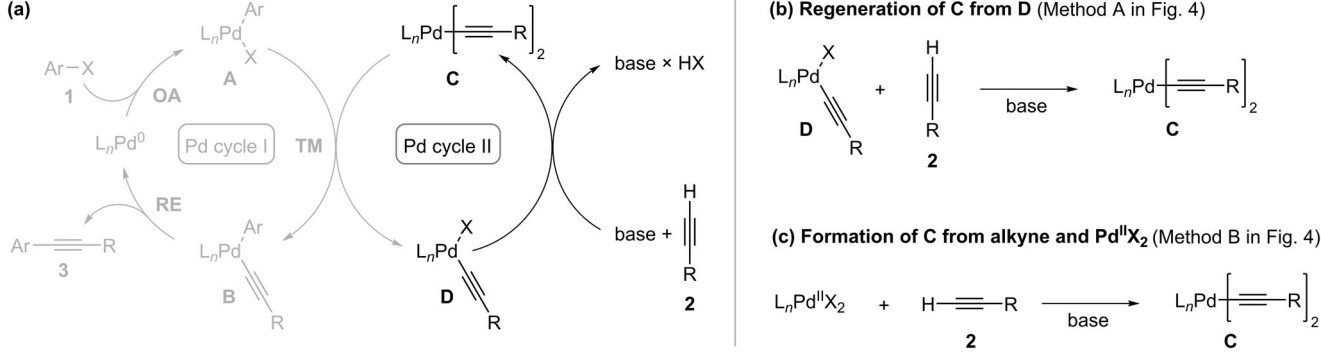

**Fig. 1 Investigated reaction and its mechanistic analysis. a** Palladium catalysed cross-coupling between aryl or vinyl halides and terminal alkynes. **b** Monometallic and **c** bimetallic mechanistic proposal. **d** Disassembly of the proposed bimetallic mechanism on elementary reactions. *Cis–trans* isomerization steps are omitted for clarity. OA oxidative addition, TM transmetallation, RE reductive elimination.

**Fig. 2 Proposed formation of organometallic nucleophile.** Breaking **a** highlighted 'Pd catalytic cycle II' into reaction for **b** regeneration of palladium bisacetylides **C** from palladium monoacetylides **D** and **c** initial formation of palladium bisacetylides **C** from $L_nPd^{II}X_2$ source.

With the palladium monoacetylides **5** in hand, the synthesis of palladium bisacetylides **6**, the mimics of **C**, was carried out by two different methods (Fig. 4). In Method A, the above mono-acetylides **5** were subjected to a base-mediated reaction with alkynes **2**. In the proposed bimetallic mechanism, this reaction is also considered as a regeneration pathway of **C** from **D** in 'Pd catalytic cycle II'. In this way, we were able to prepare **6** in good to excellent isolated yields (28–95%) from iodo, bromo as well as chloro monoacetylides **5**. The halogen ligand in **5** had no discernible effect on the course of the reaction (Fig. 4). For

example, **6a** was synthesized from iodo **5a**, bromo **5b** and chloro **5c** monoacetylides in 80%, 95%, and 68% isolated yields, respectively. The substituent on the alkyne ligand also had little effect on the isolated yields of the bisacetylides **6**, except for **6l** (28%) and **6p** (38%) from **5e** and **5h**, respectively. Method B, on the other hand, consists of a base-mediated reaction between alkyne **2** and $L_nPd^{II}X_2$ as a palladium source, which is also the first step in the formation of palladium acetylide intermediates that initiate the catalysis. The palladium bisacetylides **6** were isolated by simple filtration in 27–93% yield (Fig. 4).

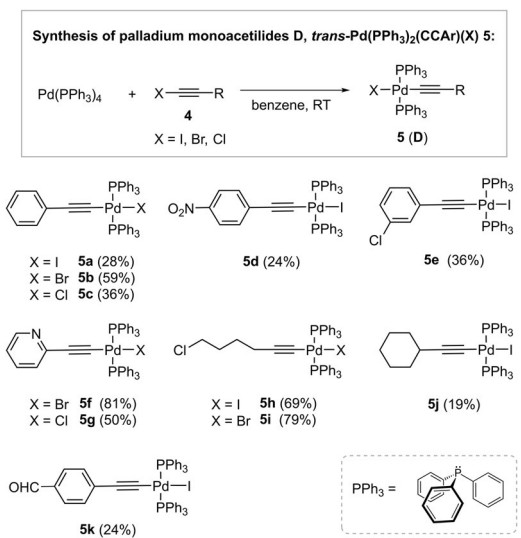

**Fig. 3 Preparation of palladium monoacetylides *trans*-Pd(PPh₃)₂(CCAr)(X) (5) that mimic intermediate D.** Synthetic details are described in Supplementary Information.

Using Methods A and B, we were able to prepare 17 symmetrical bisacetylides **6** with alkyne ligands bearing aryl, heteroaryl or alkyl substituents. Method A was also amenable for the preparation of unsymmetrical bisacetylides **6r–6 u** (76–93% yields) bearing alkynes with aryl, heteroaryl or alkyl substituents (Fig. 4). The palladium bisacetylides **6** are bench stable solids under ambient conditions. There was no noticeable degradation of **6a** over a period of 6 months as indicated by NMR analysis ($^1$H and $^{31}$P) when we stored it in a vial on a bench.

**Insight into Pd-Pd transmetallation**. The ability to isolate the intermediates prior to transmetallation provides a unique opportunity to study the critical step of the proposed mechanism, transmetallation. It is noteworthy that the studies on transmetallation reactions are scarce, with the transmetallations of stannane, boronate, organosilane and organozincate nucleophiles with palladium complexes being the most extensively studied[41,63]. The exchange of groups between Pd(II) complexes, including the transfer of alkynyl groups, has been studied previously, however, this process has not been placed in the context of catalytic cross-coupling between aryl halides and terminal alkynes[29–31].

With the palladium acetylides in hand, their reactivity in transmetallation with palladium oxidative addition complexes **7** (Fig. 5) was investigated. The derivatives **7**, including chloride, bromide, and iodide analogues, were prepared independently from aryl halides **1**, according to procedures described in the literature (Fig. 5)[64–66]. Catalytically active species are usually not stable enough to be isolated, so their precursors are usually isolated. For example, Pd oxidative addition complexes **7**, *trans*-Pd(PPh₃)₂(Ar)(X) are isolated as precursors to catalytically active tricoordinated Pd(PPh₃)(Ar)(X) species. Precursors to catalytically active species are usually observed in reactions as resting states of catalytic cycles.

To get an idea of the generality of the transmetallation, Pd oxidative addition complexes **7** with different halogen substituents (I, Br, Cl) and aryl ligand consisted of *para*-substituted phenyl ring with the substituents having different electronic effects, from strongly electron withdrawing to electron donating, were selected for screening (Fig. 6). Similarly, symmetrical Pd-acetylides **6** with various substituents on alkynyl groups were selected for screening. The progress of the reactions was monitored by $^1$H and $^{31}$P NMR spectroscopy. The former was used in conjunction with comparison with the spectra of

independently prepared authentic samples to determine the conversion to tolane product **3**, while the latter provided in situ monitoring of the phosphorus-containing species in the reaction mixture. An example of the monitoring of the reaction by NMR is shown in Fig. 6a, b, and further examples can be found in Supplementary Information (Supplementary Figs. 1–12). In Fig. 6c are reported the conversions of the respective transmetallation to the corresponding product **3** after 2 h from the onset of the reaction. It is noteworthy that all of the studied reactions between **6** and **7** resulted in the corresponding internal alkyne products **3**. Although the **B**-like transmetallation intermediate should be formed along the reaction pathway (Fig. 1c), it was not detected in any of the reactions from Fig. 6c.

The following trend was observed in reactivity: the halogen ligand of **7** affected the reaction rate the most and was increased from iodine, bromine to chlorine. The more electron donating group (R¹ = OMe, Me) at **7** proved to be more favourable than the electron withdrawing one (R¹ = NO₂). The opposite was true for R² at the alkyne ligand in **6**, with the electron-withdrawing nitro group of **6j** having the most favourable effect on the formation of product **3**. Lesser effects were observed for the groups with moderately donating or withdrawing electronic properties, e.g., Me in **6c**, *tert*-Bu in **6d**, Cl in **6f**, as well as for the 2-thienyl ligand in **6n**. In sharp contrast, the pyridyl ligand in **6m** dramatically accelerated the reactions. This could be due to the coordination ability of the pyridine nitrogen atom, which can act as a bridging ligand and facilitates the transmetallation process[67]. The Pd acetylide with the olefin substituent **6q** reacted similarly to the acetylides with aromatic substituents, while the reaction of the acetylide **6p** with alkyl groups was the most sluggish. All reactions between **6** and **7** were clean up to 2 h after initiation and led to the formation of disubstituted alkyne **3**, the palladium monoacetylide **5**, and Pd⁰(PPh₃)₂ (**8**)[68], formed by reductive elimination, as the only products, as evidenced by $^{31}$P NMR analyses of the reaction mixtures (Fig. 6b). The alkyne ligands present in the solution can stabilize **8**[69]. After 2–4 h, additional resonances began to appear in $^{31}$P NMR spectra, which could be due to decomposition of complexes and side reactions between the species formed. However, these species are not relevant to the reactions studied, so we did not assign them further.

The systematic variation of substituents on both the palladium oxidative addition complexes and the palladium acetylides provides insight into the critical step of catalysis, transmetallation. To simplify the analysis, we performed the reactions without additives, which can interact with the palladium complexes as coordinating ligands and influence the reactions (vide infra).

First, the influence of the halogen ligand in **7** on the reaction rate was examined (Fig. 7). The resonance of the methyl group of the tolyl substituent in **7** was used as a probe for monitoring the reactions by $^1$H NMR due to its characteristic chemical shift in the oxidative addition complexes **7** (δ = 2.29 ppm) and product **3** (δ = 2.37 ppm). We performed the reactions in the NMR tubes and monitored the respective product **3** formation over time by $^1$H NMR at 5 min intervals. All reactions studied showed concave profiles of concentration **3** over time with maximum rates at the beginning of the reactions (Fig. 7 and Supp. Info). The determined rates indicated that the substituents in **7** accelerated the transmetallation reaction in the following order: Cl > Br > OAc > I (Fig. 7, entries 1–4). Similar studies of halide ligands in palladium oxidation addition complexes revealed that the more electronegative halide ligand facilitates the process in the Pd-Sn system[72], and the rate of cleavage (reaction) of tricoordinated palladium oxidation addition complexes with Cl, Br, and I halogen atoms in the order Cl > Br > I was observed with secondary amines[73] (see below for further discussion of the halide effect on the Pd-Pd transmetallation studied).

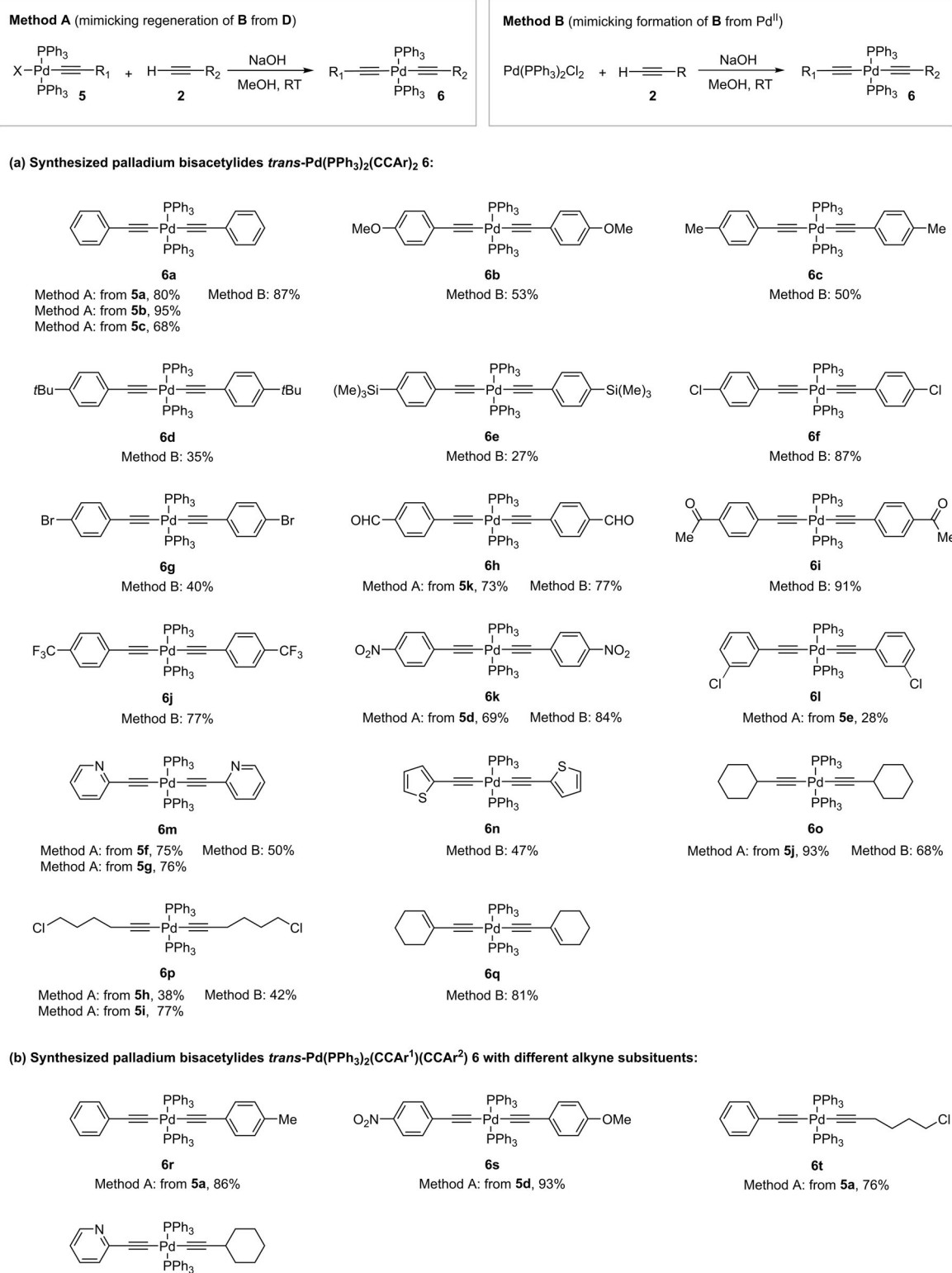

**Fig. 4 Independently prepared palladium bisacetylides 6 that mimic intermediate C. a** Synthesized symmetrical palladium bisacetylides **6** by Methods A and B. **b** Synthesized unsymmetrical palladium bisacetylides **6** by Method A. Method A mimics the regeneration of palladium bisacetylides **C** from monoacetylides **D**, and Method B the initial formation of palladium bisacetylides **C** from Pd[II]X$_2$ source. Synthetic details are described in Supplementary Information.

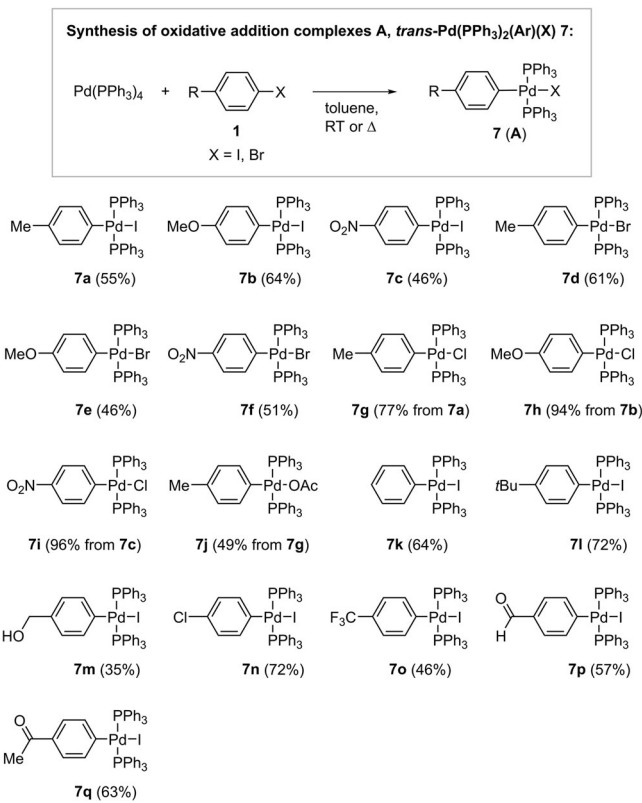

**Fig. 5 Synthesized palladium oxidative addition complexes 7, mimics of A.** For synthetic details see Supplementary Information (Supplementary Note 1).

In order to evaluate the productivity of the transmetallation reaction with respect to the acetylide reagent, we compared the reactivity of different types of palladium acetylides **5** and **6** in the reaction with **7a**, since both palladium monoacetylides **D** and **C** can potentially contribute to the formation of product **3** via the transmetallation reaction with **A**. The reactions of palladium monoacetylides **5a**, **5b** and **5c** bearing iodine, bromine and chlorine halogen substituents, respectively, with **7a** were sluggish compared to the reaction with bisacetylide **6a** (Fig. 7, entries 5–7). Moreover, compounds **5** were found to be highly reactive with terminal alkynes **2** in the base-mediated reaction, as we did not detect them as intermediates in the reaction mixtures during the preparation of **6** from alkynes **2** and $Pd^{II}X_2$ source (vide supra). The lower reactivity of **D** compared to **C** with respect to **A** could also be due to the fact that **C** has two alkyne handles while **D** has only one, and trans influence of halogen ligand in **D**, which lowers the reactivity of alkyne handle[74,75]. All this suggests that the palladium monoacetylides **D** contribute very little overall to product formation in the transmetallation reaction with **A**.

To gain further insight into the substituent effect on Pd-Pd transmetallation, we performed a Hammett substituent analysis[76] based on the initial rates of the reactions between **6** and **7** (Fig. 8). It is noteworthy that Hammett analyses of transmetallation reactions in general are rare to our knowledge, with that for the Pd-Si transmetallation of the Hiyama-Denmark reaction being the only example to date[77].

Plotting the relative rates for transmetallation reactions between the oxidative addition complexes **7** and palladium bisacetylide **6a** against the σ-values[76] for the substituents $R^1$ on the electrophile gives a negative Hammett constant (Fig. 8a, ρ = −0.42). Since this reaction is a complex multistep process, both the conversion of the starting material **7** and the product formation **3**

were monitored over time. In all cases, the maximum reaction rate was reached at the beginning of the reaction. The rates as well as the Hammett plots for conversion of starting **7** and product formation **3** were aligned, indicating direct involvement of **7** in the rate-determining step and that the electron-donating substituents in **7** accelerated the reaction rate.

Given the data in Fig. 6c, the electronic effect of palladium bisacetylide **6** substituents in the transmetallation step could be tentatively determined by comparing the conversions for a particular **7**. This analysis shows that palladium bisacetylides **6** with more electron-withdrawing substituents react faster (Fig. 6c). Hammett analysis using σ values[76] of the reaction between acetylides **6** with different substituents $R^2$ and **7a** revealed a linear relationship with a positive ρ value (Fig. 8b, ρ = 0.48), indicating that electron-withdrawing substituents increase the reactivity. The rates and the Hammett plots for the conversion of starting palladium acetylides **6** and the formation of product **3** were aligned, indicating direct involvement of **6** in the rate-determining step and that electron-withdrawing substituents in **6** accelerate the reaction rate.

Competition experiments using either a mixture of palladium oxidative addition complexes **7a** and **7c** with palladium bisacetylide **6a** as the limiting reagent or a mixture of palladium bisacetylide **6c** and **6k** with palladium oxidative addition complex **7k** as the limiting reagent showed formation of corresponding products **3** at rates (and ratios) similar to those in Hammett plots (Supplementary Figs. 20 and 21).

The reaction between **6** and **7** consists of two elementary steps of the proposed reaction mechanism in Fig. 1c, transmetallation and reductive elimination. Monitoring the reactions in Fig. 6 by $^{31}P$ NMR, we observed only palladium monoacetylide **5** and $Pd(PPh_3)_2$ (**8**), the products of the transmetallation and reductive elimination steps, in addition to the initial **7** and **6** in all the reactions studied. The absence of intermediate **B** in the reaction mixture suggests that reductive elimination of intermediate **B**[30–32] proceeds more rapidly than transmetallation and that the latter is most likely the rate-determining step for the reactions between **6** and **7** under the study.

On the basis of these results, we have postulated the most probable pathway of the transmetallations under study. It has already been shown that 4-coordinated palladium oxidative addition complexes, such as **7**, are more likely to undergo dissociation of a $PPh_3$ ligand and subsequent reaction with the organometallic nucleophile than a direct associative reaction[8,41,63]. *Cis*-species **7** were previously proposed as reactive species in the transmetallation step of the Stille reaction[78], but were not observed when monitoring the oxidative addition under study (Supplementary Figs. 22−24). The addition of excess $PPh_3$ to the reaction of **6a** and **7a** stopped the reaction completely, and slowed catalytic reaction of **1a** and **2a**, further indicating dissociative pathway (Supplementary Figs. 25,26). Therefore, the first step is most likely the dissociation of $PPh_3$ from **7** into coordinatively unsaturated **7−PPh₃**, which associates with **6** via **9** by π-coordination of the triple bond, to form the cyclic intermediate **10**, in which alkyne and iodine act as bridging ligands that bind to both palladium centers (Fig. 9). It was demonstrated that halogen ligands, such as iodide and chloride, promote cyclic transmetallation pathway in Pd-Sn and Pd-Au transmetallation, respectively[79,80]. Migration of the alkyne and iodide ligands could occur in two steps via the cyclic and open intermediates **11** and **12**, respectively, where stepwise ligand exchange occurs. Based on the results of Hammett correlations as well as previous DFT calculations[32], we postulated that dissociation of the Pd-Pd complexes from **12** is most likely the rate-determining step of the process under investigation. After

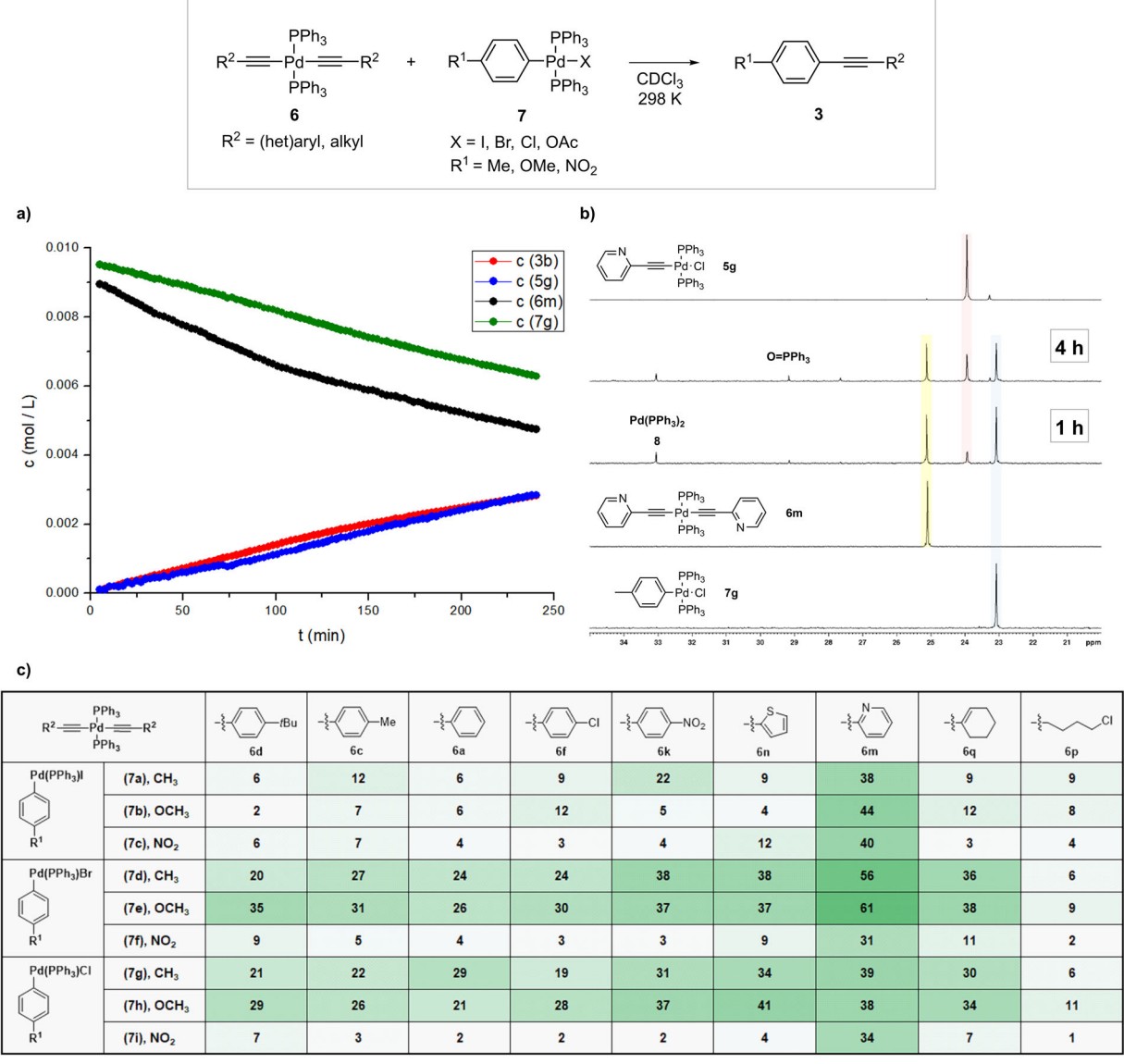

**Fig. 6 Probing transmetallation reactions of 6 and 7.** Example of monitoring the transmetallation reaction **a** concentration/time plot of key species (**3b**, **5 g**, **6 m**, **7 g**) and **b** stack of $^{31}$P NMR spectra of reaction between **6 m** (0.01 M) and **7 g** (0.011 M). **c** The conversion into tolan product **3** after 2 h at 298 K as determined by $^1$H NMR is reported. Conditions **6** (0.01 M) and **7** (0.01 M) in CDCl$_3$, 298 K, 2 h. Concentration of starting complexes was comparable to that of palladium precatalyst in catalytic version of the reaction[70,71]. CDCl$_3$ was chosen as the solvent for monitoring the transmetallation reactions because it proved to be the most suitable, as shown by the comparison of the transmetallations in different solvents (Supplementary Figs. 13–17).

dissociation of **12** into **5** and **B-PPh₃**, the latter could associate with PPh₃ before it reductively eliminates **3**, although reductive elimination from 3-cooridnated Pd-species could also be possible[32], with subsequent reassociation of Pd(PPh₃) with PPh₃ into Pd(PPh₃)₂ (**8**).

In the transmetallation described, similar types of bonds are broken and formed, so the driving force of the process could be the trans effect of the aryl and alkynyl ligands in **12**, with the aryl having a larger trans effect than the alkynyl ligand[74,81], as well as the irreversible reductive elimination of **3**. The order of reactivity Cl > Br > I of the halogen in **7** could be explained by the strength of the Pd-X bond. It was postulated that the strength of the dative Pd-(μ-X) bond should parallel the donor ability of the lone halide electron pairs and therefore be strongest when X = I[73]. A larger halide in the dimeric species **10** leads to a weaker Pd-alkynyl bond and disfavours the cleavage of the alkynyl ligand from the palladium bisacetylide, which could also lead to the observed

order of reactivity Cl > Br > I of the halogen in **7** in the transmetallation process. It is also possible that RE is the rate-determining step in which both **6a** and **7a** contribute to the formation of the key intermediate **B**, with the equilibrium on the left and therefore only **6** and **7** are observed.

**Kinetic studies of elementary steps of the mechanism.** We sought to determine the rate-determining step of the catalytic process by comparing the rates of the elementary steps of the proposed mechanism, i.e., oxidative addition, transmetallation with reductive elimination and palladium bisacetylide reformation, under similar reaction conditions as shown in Table 1. For this purpose, 4-iodotoluene (**1a**) and phenylacetylene (**2a**) were selected as model substrates, together with the corresponding oxidative addition complex **7a** and palladium acetylides **5a** and **6a**. For the catalytic reaction, Pd(PPh₃)₂I₂ was selected as a pre-catalyst to avoid the effect of halide ion exchange (vide infra),

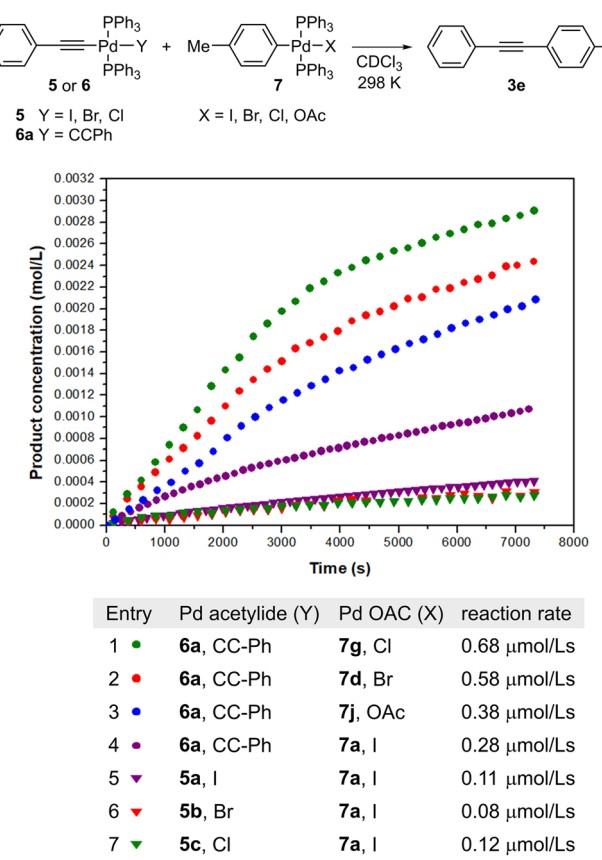

**Fig. 7 Investigating the influence of ligands in complexes 5 and 7 on the rate of the transmetallation reaction.** Conditions: **5** or **6a** (0.01 M), **7** (0.01 M), CDCl₃, 302 K.

while dichloromethane and pyrrolidine were selected as solvent and base, respectively. It is known that coordinating amine bases, such as pyrrolidine, can act as ligands that interact with palladium complexes to affect the speciation in the reaction mixture[32]. For example, pyrrolidine can replace one of the phosphines in **7**, leading to an equilibrium between **7** and **7a'(pyr.)**, Pd(PPh₃) (pyrrolidine)(Ar)(X) (Supplementary Fig. 28)[20,32,82]. On the other hand, pyrrolidine has little effect on structures **5** and **6** (Supplementary Figs. 29,30)[32].

The reaction of **1a** with Pd(PPh₃)₄ to form **7a** was immediate, consistent with examples of oxidative addition of aryl iodides to Pd⁰ reported in the literature (Table 1, Entry 1)[32,83,84]. Although *cis*-**7a** is initially formed, it could not be observed because it is in equilibrium with the thermodynamically favourable *trans*-isomer **7a**[85]. It could be that in the catalytic reaction the transmetallation described above with respect to the oxidative palladium addition complex occurs immediately after the dissociation of PPh₃ from the initially formed *cis*-**7a**, which would correspond to **7a-PPh₃**, before the formation of *trans*-**7a**. The addition of pyrrolidine to Pd(PPh)₄, followed by the addition of **1a**, also led to the immediate formation of **7a**, which was in equilibrium with **7a'(pyr.)**[32]. The rate of oxidative addition of aryl or pseudoaryl halides (Ar-X) usually follows the dissociation enthalpies of the corresponding bonds (C-X), and in this concept the order of reactivity is Ar-I > Ar-Br ≈ Ar-OTf > Ar-Cl[86]. This indicates that replacing the aryl iodide starting substrate with aryl bromide or chloride would completely change the rates of the elementary steps in the catalytic reaction, most likely resulting in oxidative addition being the rate-determining step. Similarly, it has been shown that substituents on the phenyl ring of the aryl halide or

alkyne reagent can affect the change in the rate-determining step of the catalytic process[21].

As indicated above, the tricoordinated palladium oxidative addition complex is the most likely the reactive species in the catalytic reaction, therefore we prepared Pd(P(*o*-tolyl)₃)(4-Me-C₆H₄)(I)[87] **7a'** as a surrogate for tricoorrdinated Pd(PPh₃)(Ar)(I) species, which formed the product **3e** in reaction with **6a** (Table 1, entry 2) at rate comparable to the catalytic reaction (entry 4). While the transmetallation between **6a** and **7a'** was the slowest of the reactions studied (Table 1, entry 2), due to the instability of **7a'**, i.e., aryl group scrambling and side reactions[88,89], a lower than potential rate was observed during the reaction (Supplementary Figs. 31,32); the rate would otherwise likely be higher than that observed in the formation of **6a** from **5a** described. The rate of formation of **6a** from **5a** and **2a** (Table 1, entry 3) was almost completely aligned with the rate of catalytic reaction, and it most likely the rate-determining step of the process under study. The rates of catalytic and stoichiometric reactions are in good agreement considering that the stoichiometric reactions of the complexes expected as intermediates are different from those under catalytic conditions[69]. It is noteworthy that the reactions of entries 1, 2 and 3 from Table 1 had the highest rates at the beginning of the reaction, while the rate of the catalytic reaction of entry 4 had the induction period at the beginning and its maximum rate was reached after 28 minutes.

To evaluate the mechanistic analysis approach described, we performed a kinetic analysis under synthetically relevant conditions to determine the order in each reagent. These were determined from the slopes of the linear functions of the ln/ln plots of the maximum rates, obtained by derivation of the sigmoid curves from the fit of the experimental data, against the corresponding concentration for each reagent and catalyst. In addition, we performed a VTNA analysis[5] for the catalytic reaction under study by plotting the concentration of **3e** against t × [Z]$^y$, where [Z] is the concentration of the reagent and y is its order. We calculated the order for each reagent by performing reactions with different initial concentrations of the reagents. Both analyses gave similar results, the orders of 1.0 in Pd, -0.1 in aryl iodide, 1.1 in pyrrolidine and 0.3 in phenylacetylene were obtained for the traditional kinetic analysis, while the orders of 1.2 in Pd, 0 in aryl iodide, 1.1 in pyrrolidine and 0.4 in phenylacetylene for the VTNA analysis (Supplementary Figs. 33-41). These results indicate that one of the transition states within the multistep process of formation of palladium bisacetylide **6a** from **5a** is most likely the rate-determining step of the catalytic reaction studied.

Although the order in Pd for the catalytic reaction studied is 1, it is reasonable to assume that the reaction proceeds via a Pd-Pd transmetallation step between palladium oxidation addition complex and palladium bisacetylide. This is indicated by the presence of palladium bisacetylides, as well as other proposed key intermediates, in the reaction mixture during catalysis, as shown by the ³¹P NMR[32], where the palladium bisacetylides are formed at relevant rates with respect to the catalytic reaction (Table 1). Moreover, and most importantly, under identical conditions, the reaction of the palladium bisacetylide with the palladium oxidative addition complex to give the tolane product **3** is instantaneous, while in the reaction of the palladium oxidative addition complex with the terminal alkyne **2** in the presence of a base, the induction period is present at the beginning of the reaction (in which palladium acetylides are formed to initiate the reaction)[32]. It should be noted that transmetallation need not be the rate-determining step to assume that it occurs in the process, since, for example, oxidative addition was found to be the rate-determining step in the Pd-Cu Sonogashira reaction studied[90], which does not preclude Pd-Cu transmetallation from proceeding in the reaction.

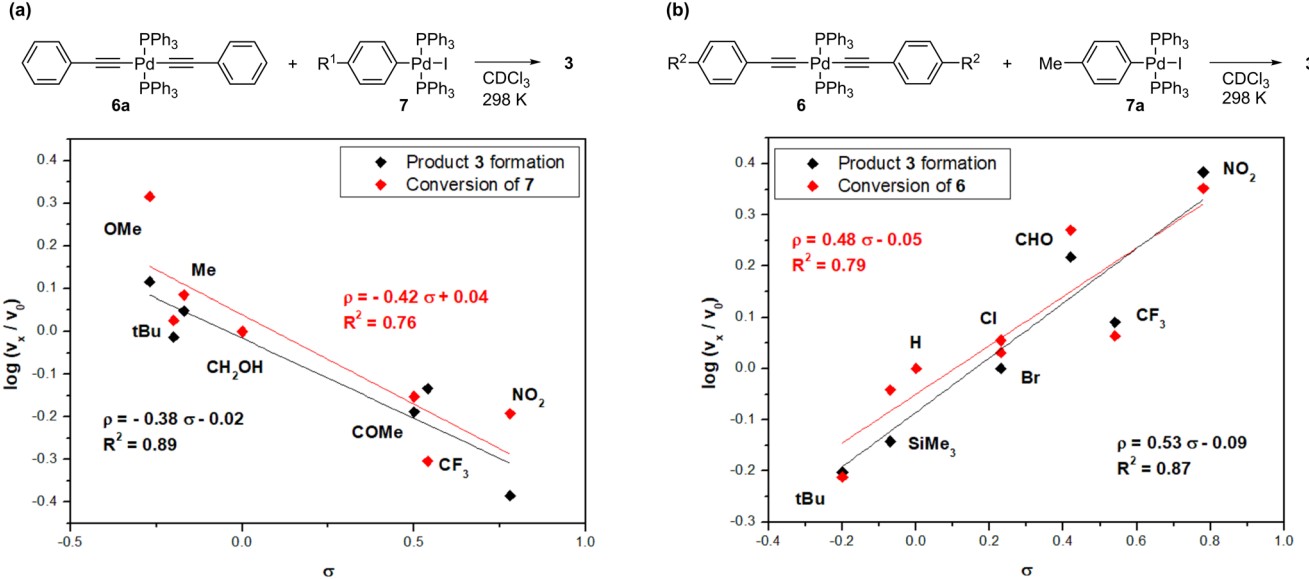

**Fig. 8 Hammett plots for the transmetallation reactions of 6 and 7. a** Reaction of **6a** with *para*-substituted oxidative addition complexes **7**. Conditions: **6a** (0.011 M), **7** (0.01 M), CDCl₃, 302 K. **b** Reaction of **7a** with *p*-substituted palladium bisacetylides **6**. Conditions: **6** (0.01 M), **7a** (0.011 M), CDCl₃, 302 K. For Hammett plots using $\sigma_p^+$ ($\sigma_p^-$) constants, see Supplementary Figs. 18,19.

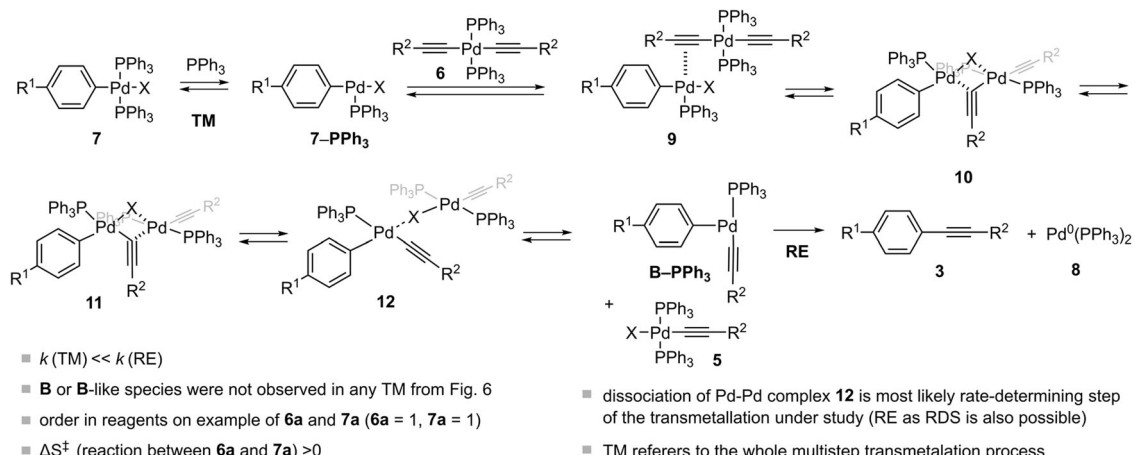

- $k$ (TM) << $k$ (RE)
- **B** or **B**-like species were not observed in any TM from Fig. 6
- order in reagents on example of **6a** and **7a** (**6a** = 1, **7a** = 1)
- $\Delta S^{\ddagger}$ (reaction between **6a** and **7a**) >0

- dissociation of Pd-Pd complex **12** is most likely rate-determining step of the transmetallation under study (RE as RDS is also possible)
- TM referes to the whole multistep transmetalation process

**Fig. 9 Postulated multistep Pd-Pd transmetallation under study.** Proposed transmetallation pathway based on the results of the reaction between **6a** and **7a**. The reaction most likely proceeds by initial dissociation of PPh₃ from **7a** and via the cyclic intermediate **10**, in which alkyne and iodine act as bridging ligands (Supplementary Table 1, Supplementary Fig. 27).

**Accelerating catalytic reaction by in situ halide metathesis**. The influence of halide ions on the rate of cross-coupling reactions is well known. For example, fluoride, hydroxy or other ions can add to organostannanes[91], boronic acids[92–94], organozinc[95], and organobismuth compounds[96] to form more reactive organometallic nucleophiles that accelerate cross-coupling reactions by accelerating the transmetallation step. The halogen ligand in the palladium oxidative addition complex was shown to have an effect on the rate of transmetallation with organostannanes[97,98], which was used to accelerate catalytic Stille reaction by in-situ halide metathesis[94]. Moreover, a distinct chloride effect was also described for stabilization of Pd species by in situ reduction of the Pd(II) precatalyst, leading to an anionic [Pd⁰Cl(L)₂]⁻ and in this way facilitating oxidative addition[84,99].

The catalytic system studied becomes increasingly complex when additional halide (chloride) ions are added to the reaction mixture, in addition to a coordinating base (pyrrolidine), since both can act as ligands and form an equilibrium species with palladium intermediates, giving rise to several potential reaction intermediates.

We performed the following experiments, summarized in Fig. 10. Cross-coupling between **1a** and **2a** was first conducted with Pd(PPh₃)₂I₂ as the palladium precatalyst to provide a reference reaction in which no chloride ions were present (Fig. 10, Table, entry 1). Upon addition of 0.2 or 1.1 equivalents of Bu₄NCl to the starting aryl iodide **1a**, we observed a 2- and 3-fold acceleration of the rate, respectively (Fig. 10, entries 3 and 4). Alternatively, the use of Pd(PPh₃)₂Cl₂ instead of Pd(PPh₃)₂I₂ accelerated the reaction due to release of chloride ions from Pd(PPh₃)₂Cl₂.

To further substantiate the role of chloride and pyrrolidine as potential ligands, we performed the elementary reactions with complexes **A**, **C** and **D** of the postulated mechanism of Fig. 10a with the addition of chloride ions and pyrrolidine (Fig. 11). Iodide exchange in the oxidative addition complex **7a** with chloride occurred immediately and with >95% in presence or absence of pyrrolidine (Fig. 11a). Pyrrolidine formed an equilibrium with both the iodine and chloro oxidative addition complexes **7a** and **7g**, with a greater effect on **7g**, as shown by the

**Table 1 Comparison of the rates of isolated elementary steps with the catalytic reaction between 1a and 2a.**

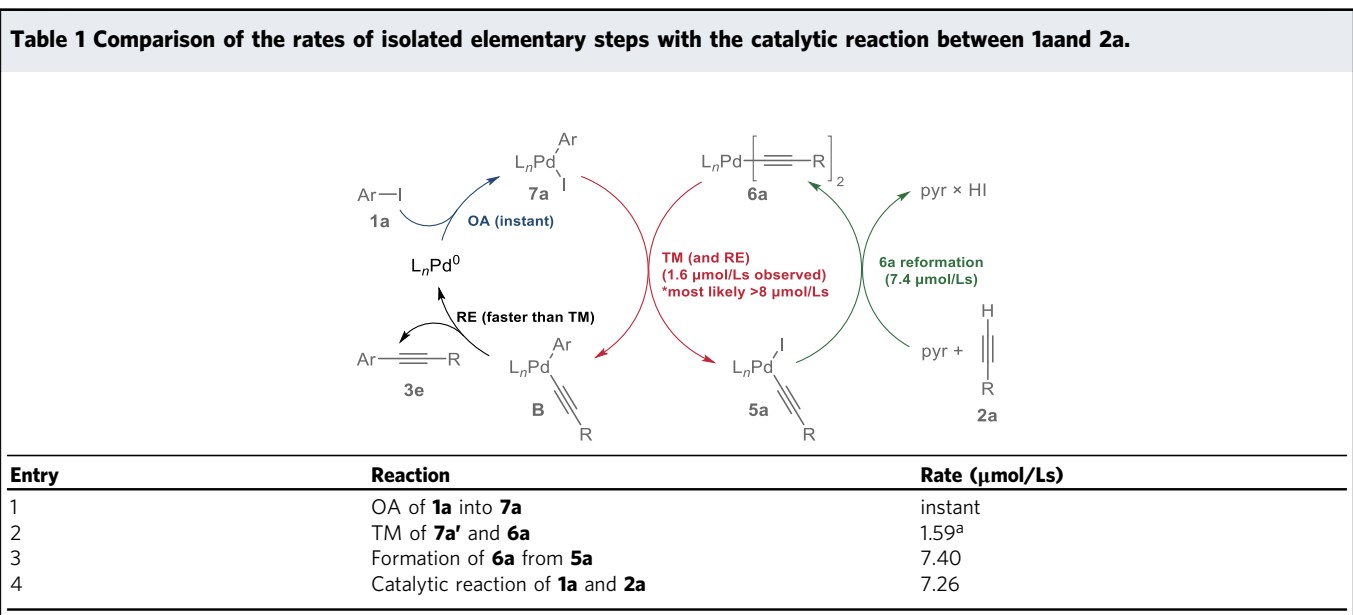

| Entry | Reaction | Rate (μmol/Ls) |
|---|---|---|
| 1 | OA of **1a** into **7a** | instant |
| 2 | TM of **7a'** and **6a** | 1.59[a] |
| 3 | Formation of **6a** from **5a** | 7.40 |
| 4 | Catalytic reaction of **1a** and **2a** | 7.26 |

In all cases the concentration of palladium precatalyst or reagent(s) was the 0.01 M for the compared reactions, i.e., Pd(PPh₃)₂I₂ for the catalytic reaction, Pd(PPh₃)₄ for oxidative addition into **7a**, and **6a** and **7a'** for transmetallation reaction. The reactions were performed in dichloromethane or CDCl₃ at RT (296 K) in the presence of pyrrolidine, apart from TM reaction. The rates of studied stoichiometric and catalytic reactions are similar in CH₂Cl₂ and CHCl₃ (CDCl₃). [a]The actual rate is most likely higher than the observed rate due to decomposition of **7a'**, without the decomposition of **7a'**, the rate would probably be > 8 μmol/Ls. For more details see the Supplementary Note 1. Pyr = pyrrolidine; Ar = 4-CH₃-C₆H₄-; R = Ph.

corresponding equilibrium constants (Fig. 11b,c). Similar observations were made for the palladium monoacetylide **5a**, which immediately converted to the chlorine analogue **5c** in the presence of chloride ions, with pyrrolidine having a greater effect on the latter than on **5a** (Fig. 11d-f, Supplementary Figs. 23,24,28,29,42,43). In contrast, chloride ions and pyrrolidine had no effect on the palladium bisacetylide **6a**, as determined by ³¹P NMR (Fig. 11g) (Supplementary Fig. 44).

Identifying the rate-determining step in such a complex system is therefore extremely challenging and will be the subject of further studies. Although chloride ions could facilitate any of the steps within the proposed mechanism, the increase in the rate (3-fold) of the catalytic reaction in the presence of chloride ions corresponded to the increase in the rate of the transmetallation reaction of **7a** and **6a** (3-fold) when treated with chloride ions (Supplementary Fig. 45). However, this could be a coincidence, since the rate-determining step in this reaction has not been clearly confirmed and could be within any (multistep) process of formation of key proposed mechanistic intermediates. Incidentally, we found that the presence of pyrrolidine affects the order of **7a** in the transmetallation reaction. We examined the order of **6a** and **7a** in the transmetallation reaction in the presence of pyrrolidine, which resulted in the order 1 in **6a** and 0.5 in **7a** (Supplementary Figs. 46-50). Since **7a** forms **7a'(pyr.)** in equilibrium with pyrrolidine, which is not a reactive species in the transmetallation steps as described above, the order in **7a** decreases. This suggests that if there is a large excess of pyrrolidine, the order of **7a** may be limiting toward 0, making it even more difficult to observe transmetallation as a rate-determining step by traditional methods of kinetic analysis (in case when transmetallation would be the rate-determining step of the process).

It is noteworthy that the results of Figs. 10 and 11 show the importance of careful selection of precatalysts, when they are in the form of halides, when the kinetics of such a reaction is studied under synthetically relevant conditions, since possible halide metathesis takes place in situ. This can influence the structure of key mechanistic intermediates and the rates of elementary reactions of the mechanism.

## Conclusion

An experimental mechanistic insight into a unique type of synergistic bimetallic catalytic process involving a single metal, namely, the palladium catalysed cross-coupling of aryl halides and terminal alkynes, is described. For the study, the proposed reaction mechanism was divided into two palladium catalytic cycles and further into elementary reactions and each step was studied independently. Palladium bisacetylides, not previously described in the context of cross-coupling catalysis, are described here in detail as organometallic nucleophiles and have proved to be a very useful tool for the study of the mechanism under investigation. Direct substitution of halide in Pd(PPh₃)₂X₂ by alkyne in the presence of a base provides a general and simple route to symmetrical palladium bisacetylides, whereas unsymmetrical bisacetylides can be accessed by oxidative addition of halogenated alkynes to Pd(PPh₃)₄ followed by halogen substitution with the alkyne ligand. We gained additional insight into Pd-Pd transmetallation by stoichiometric reactions of palladium bisacetylides and palladium oxidative addition complexes through Hammett plots. Kinetic studies of the postulated elementary steps of the mechanism revealed balanced kinetics of the palladium catalytic cycles, where comparable reaction rates were observed for the formation of key intermediates. The described approach allowed us to identify the rate-determining step of the studied catalytic process by comparing the rates of the elementary steps under similar reaction conditions, which was further evaluated by traditional kinetic and VTNA analyses. In the case of the studied catalytic reaction of tolyl iodide and phenylacetylene, this proved to be the palladium bisacetylide formation from palladium monoacetylide. It should be noted that under different conditions (e.g., different ligands and reaction conditions), other mechanisms could compete with or replace the one described here. Similarly, the use of different aryl halide and alkyne starting reagents would completely change the rates of the elementary steps of the catalytic reaction. Furthermore, since the stoichiometric reactions of the complexes expected as intermediates differ from those under catalytic conditions[69], attention should be paid to possible side reactions, as in the case of the Pd-Pd transmetallation studied, where decomposition of the palladium oxidative

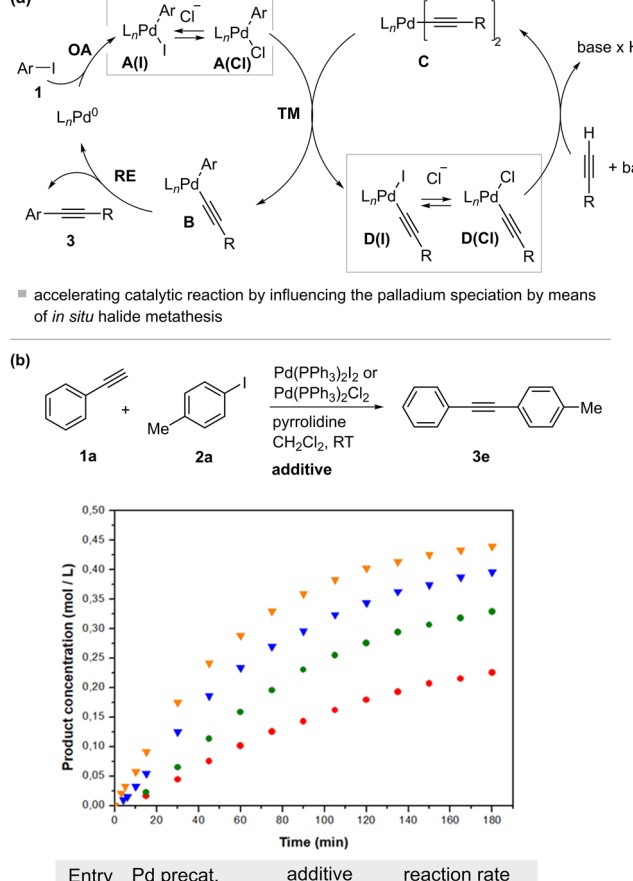

Fig. 10 Accelerating catalytic reaction by in situ halide metathesis.
**a** Influencing structures of key proposed mechanistic intermediates.
Equilibrium structures with pyrrolidine and omitted for clarity. **b** Evaluating
the effect of halide ions on the rate of catalytic reaction of **1a** and **2a** via
substituents in palladium precatalysts and by addition of additive. Conditions:
**1a** (0.5 M), **2a** (0.55 M), palladium precatalyst (0.05 M, 10 mol%),
pyrrolidine (1.0 M), CH₂Cl₂, room temperature (RT, 296 K).

addition complex during the reaction lowered the reaction rate.
Reports of synergistic dual catalysis with only one metal are
sparse and limited to a few Pd-Pd systems. Understanding the
mechanisms of such reactions is key for their rational optimiza-
tion as well as for further development of this type of cross-
coupling catalysis, as reflected by emerging one metal/two ligands
dual catalytic systems[100].

## Methods

**Preparation of palladium bisacetylides 6 from monoacetylides 5 (Method A)
or Pd(PPh₃)Cl₂ (Method B).** Method A: The reaction flask was charged with
bis(triphenylphosphine)palladium(II) acetylide halide **5** (1 equiv.) under argon
atmosphere. A 0.2 M solution of NaOH in methanol (40 mL/mmol) and alkyne **2**
(20 equiv.) were added under argon atmosphere. The mixture was stirred overnight
in the dark at room temperature under argon atmosphere. Then, the mixture was
filtered and the solid residue was washed with water, methanol and diethyl ether.
The solid residue was dried to give the desired product in pure form.

Method B: Procedure is similar as for Method A, but instead of **5** Pd(PPh₃)Cl₂
was used.

**General procedure for transmetallation reactions between 6 and 7.** A solution
of bis(triphenylphosphine)palladium aryl halide **7** (1 equiv., 0.01 M) and internal

standard (1,3,5-trimethoxybenzene) in degassed CDCl₃ was prepared in a vial
under argon atmosphere. Bis(triphenylphosphine)palladium bisacetylide **6** (1
equiv.) was added to this solution in one portion. The reaction mixture was
sonicated for 0.5 min and transferred into an NMR tube, purged with argon and
sealed. NMR spectra were recorded. The conversions to the corresponding pro-
ducts were determined by ¹H NMR by comparing the integrals of the characteristic
resonances of the starting compounds and of the desired product with internal
standard (1,3,5-trimethoxybenzene).

**Catalytic reactions between phenylacetylene (2a) and 4-iodotoluene (1a) (w/
o additive).** The reaction flask was charged with 4-iodotoluene (**1a**) (1 equiv.),
phenylacetylene (**2a**) (1.1 equiv.), tetrabutylammonium chloride (0.2 or 1.1 equiv.),
pyrrolidine (2 equiv.), and internal standard (1,3,5-trimethoxybenzene) under
argon atmosphere. Dichloromethane (2 mL/mmol of **1a**) was added under argon
atmosphere with a syringe by piercing the septum. Palladium precatalyst

Fig. 11 Investigating proposed elementary reactions with addition of
chloride ions and pyrrolidine. **a–c** Effect of pyrrolidine and chloride ions on
structures **7**. **d–f** Effect of pyrrolidine and chloride ions on structures **5**.
**g** Pyrrolidine and chloride ions have no effect on **6a**. Conditions: **5a**, **5c**, **6a**,
**7a, 7g** (0.01 M), Bu₄NCl (0.025 M) if added, pyrrolidine (0.05-0.15 M) if
added, CDCl₃, RT (296 K).

(Pd(PPh$_3$)$_2$I$_2$ or Pd(PPh$_3$)$_2$Cl$_2$, 0.1 equiv.) was added to this solution by rapidly opening the septum and flushing with argon. Aliquots (50 µL) of the reaction mixture were taken with a syringe at 15-minute intervals under argon atmosphere, transferred to a dry NMR tube containing degassed CDCl$_3$, and analyzed by NMR.

## Data availability

Additional data supporting the findings described in this manuscript are available in the Supplementary Information. The Supplementary Information file contains Supplementary Methods along with Full characterization data of compounds and experimental details (Supplementary Note 1) along with copies of $^1$H, $^{13}$C, $^{31}$P NMR spectra (Supplementary Data 1). All other data are available from the authors upon reasonable request.

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

## Acknowledgements

We gratefully acknowledge financial support from the Slovenian Research Agency (J1-9166, N1-0179, P1-0230, J1-3018). A.I. gratefully acknowledges financial support for the PhD scholarship from the Ministry of Education, Science and Sport, Republic of Slovenia. We thank Lovro Kotnik for help with preparation of some palladium acetylides. We thank Dr. Damijana Urankar from the Research Infrastructure Centre at the Faculty of Chemistry and Chemical Technology, University of Ljubljana for HRMS analyses. Dedicated to Professor Slovenko Polanc on the occasion of his 75th birthday.

## Author contributions

A.I. performed experiments; A.I and M.G. designed the experiments, performed data collection and analysis; J.K. and M.G. conceptualized the work. The manuscript was written by all authors.

## Competing interests

The authors declare no competing interests.
