## [Peer Review File · Communications Chemistry]

Reviewers' comments:

Reviewer #1 (Remarks to the Author):

The manuscript by Gazvoda and co-workers describes their efforts to uncover mechanistic insights associated to copper-free Sonogashira reaction, via a palladium-palladium transmetalation step. This is a follow-up to a previous work (reference 28: Nat. Commun. 9, 4814 (2018)) where the authors disclosed the involvement of a tandem Pd/Pd cycle. In the current manuscript, the authors carried out different kinetic experiments by NMR spectroscopy, based on proposed reactive intermediates, to identify the rate-determining step of the catalytic process. Despite this work provides some interesting mechanistic and organometallic features, there are key points that need to be further clarified. So, in its current form, I cannot recommend its publication.

1) I agree with the authors that there are few reports that postulate Pd/Pd catalytic systems (even when these dual systems could be responsible of additional already known transformations). However, the authors did not acknowledge properly previous fundamental studies on group exchange between PdII complexes that are relevant for this work. Espinet (*Organometallics* 16, 5730-5736 (1997)) along with Yamamoto (ref. 72 and *Organometallics* 16, 5354-5364 (1997)) have investigated the group exchange between Pd complexes, including the transfer of alkynyl groups (the main theme of this manuscript). The authors should include the two missing reference in the manuscript, and describe them in the introduction.

2) It is particularly strikingly to me that the authors did not provide any concentration-time plot showing the disappearance and formation of all the species involved along the different reactions (catalytic and stoichiometric). In the concentration-time data, they only showed the formation of the organic product. It would be very helpful to see the reaction evolution from the organometallic perspective. Moreover, in the provided NMR spectra related to the transmetalation reactions (S52-64), there are additional species that can be observed by ³¹P NMR that the authors didn't assigned. What can they comment on that?

3) In their previous work (ref. 28), the authors monitored the catalytic reactions at different reactions times by ³¹P NMR spectroscopy. In the NMR spectra, they observed the formation of different PdII complexes, including trans-[Pd(Ar)(CCPh)(PPh₃)₂]. This complex could be formed by the group exchange between complexes 6 and 7, and after an isomerization step, could provide the desired C-C coupling.

a. How do they explain that they observed this compound under catalytic conditions but they didn't observe in the stoichiometric experiments? They should address this discrepancy.

b. Is this related to the fact that they used a different solvent to perform the kinetic study (catalysis in DMF, kinetic study in CDCl₃)?

c. In this regard, CHCl₃ is not used as solvent in Cu-free Sonogashira cross-couplings (they should include this recent review on the topic: *RSC Adv.* 2021, 11, 6885), while DMF is. Why did they select CDCl₃ for the kinetic study? Especially when in their previous work they authors showed that CDCl₃ is not innocent (they observed the formation of Pd(PPh₃)₂Cl₂ when they dissolved Pd(PPh₃)₄ in this solvent).

4) The authors mentioned that the oxidative addition step is instantaneous. For supporting the first statement they referenced their previous work (ref. 28) and the supporting information of this manuscript. However, I couldn't find any systematic kinetic study in any of them that support this fact. In the previous work, the authors only recorded the NMR spectra for the oxidative addition of 1 and Pd(PPh₃)₄. In this manuscript, the authors synthesized a broad array of oxidative addition complexes, but the reaction conditions that they employed involved long reaction times (24 h for the

RI), heating (for RBr) or solvents that are not related to the catalytic or stoichiometric experiments. Thus, how can they be sure that, in all cases, the oxidative addition is immediate without monitoring the reactions?

5) In this manuscript, the authors mentioned that the cis isomer, which is initially formed after the oxidative addition event, could not be observed due to the higher thermodynamic stability of the cis isomer, supporting this statement in a work by Espinet and co-workers (ref. 73). It is true that the Espinet group observed that after the oxidative addition, trans compounds are the thermodynamically more stable isomers. However, in this article, the authors also were capable to observe the formation and even isolate a cis compound, cis-[Pd(C6Cl2F3)I(PPh3)2], after an oxidative addition event, and subsequently study its isomerization to the corresponding trans isomer. Indeed, few years later, the Espinet group showed the potential relevance of these cis species in transmetalation events during the Stille coupling of alkynyl stannanes and aryl iodides (*Organometallics* 2011, 30, 611-617), Did Gazvoda and co-workers consider this possibility? In addition, in this work, Espinet showed that the isomerization reactions can become the rate-determining step of a catalytic cycle. As result, they observed the accumulation of a trans complex that is completely analogous to the trans-[Pd(Ar)(CCPh)(PPh3)2] that Gazvoda observed in their *Nat. Commun.* paper. I highly recommend the authors to explore this option, especially considering that they didn't include any isomerization step in their proposed catalytic cycle.

6) For confirming the dissociation of PPh3 during the reaction process, the authors should investigate if the catalytic and stoichiometric reactions are slowed down by the addition of PPh3.

Reviewer #2 (Remarks to the Author):

Dear Editor,

Thank you for forwarding me this interesting paper investigating the mechanism of a copper-free palladium catalysed Sonogashira reaction. It is clear that significant time and effort went into the completion of this study, and insights into the mechanism of such reactions are of great interest to the general community. Whilst it is clear that the experimental work was completed to an extremely high standard, I believe there are some open questions about the conclusions that may need further investigation.

It's clear from the experimental work, that under stoichiometric conditions, a mixture of bisacetylide (6) and oxidative addition complexes (7), can lead to the formation of tolan products (3). The use of the Hammett plot to investigate the proposed transmetalation gives interesting insights, with a weak and opposing contribution from each reactant, and I think additional explanation here would be valuable as it is quite difficult to explain. The proposition that dissociation of (12) to the pre-reductive elimination complex may be consistent with the Hammett plot results, but one may then expect accumulation or observation of this species which was not apparently observed. I think a few experiments may help here: 1) looking at the order in each reactant with the expectation of an overall second order reaction if the bimolecular reaction is rate limiting, 2) doing competition experiments for the complexes to see if the Hammett by competition is identical to the Hammett by initial rate (i.e. have a mixture of 7b and 7k with limiting 6a, and use the ratio of the products formed to define the relative rates by competition), this will help determine if the selectivity determining step (presumably pi-complexation by Fig 9) is different from the rate determining step. The same could be done with two bisacetylide complexes with limiting oxidative addition complex (7). Point 2) may be quite work intensive if the complexes have expired and I do not think repeating the whole Hammett plot by

competition would be necessary, but a few examples may be informative. I think there also needs to be justification for the use of "corrected sigma values" which appear to use the sigma minus constants. The SI shows the plots with both, and I do not think using the normal sigma values will change the conclusions significantly but use of sigma minus implies significant resonance stabilisation in the transition state so would need further explanation.

The kinetics of the stoichiometric reactions are also quite unusual and appear to either slow or stall which is curious for a reaction that works well under catalytic conditions (plotting both starting material as well as product concentration in the main body would help make this obvious to the reader, although it is shown in the SI). In Fig 6, there appears to be quite an accumulation of Pd(PPh₃)₂Cl₂ after 4 hours, but this is not discussed, is the formation of this species coincident with slowing of the reaction? As the authors admit there is a large discrepancy between the catalytic and stoichiometric reaction rates, with the catalytic reaction going significantly faster and this needs further explanation. Although this could be due to the different reaction conditions as stated in the paper, I think we must be open to the possibility that an alternative mechanism could be proceeding as well.

At present, I do not think there is sufficient evidence to suggest that the catalytic reaction proceeds via the bimetallic mechanism, and whilst the presence of the chloride salt accelerates the reaction in both stoichiometric and catalytic reactions, this is not conclusive support that they proceed via the same mechanism (as iodides can in general cause issues in palladium coupling reactions). I think to support this hypothesis, a greater look into the catalytic reaction is needed. If the reaction truly proceeds by a rate-limiting, bimolecular transmetalation of two palladium partners then the reaction kinetics would be pseudo-zero order, but overall second order with respect to palladium (i.e. rate vs [Pd] = 2nd order).

It is also unclear to me how the bisacetylde complex would enter the catalytic cycle if a Pd(0) precatalyst is used. Presumably with a fast oxidative addition to form 7a, most if not all Pd will be at this complex, and with free 2a available this could go directly to B? I've noted in the previous paper that the bisacetylde complexes could form from the oxidatively added complex, so looking at the kinetics of this process in the context of the rate $7a + 2a + \text{pyrr}$ should give some beneficial information on what is happening under catalytic conditions.

If a Pd(II) complex is used, depending on the activation mechanism, some may go to Pd(0), and some may form the bisacetylde complex, and I think this is a really interesting point and different kinetics could be observed using different precatalysts if some lead to bisacetylde complexes and some don't (or in differing ratios). A further analysis of this would be of great interest to the general community to determine if these complexes, which clearly form, are on- or off-cycle, or if they provide an alternative and slower pathway to coupling than the traditional mechanism. I believe some of this was discussed in the previous publication, so this could be brought into the main text or referred to in justification or rebuttal of claims.

A final editing point, I would consider revising the statement on line 28: "Although such mechanistic analysis can readily determine the reagents present in the rate-determining step, it provides little information about the mechanistic pathway, i.e., in which step which reagent enters the catalytic cycle and how." There are plenty of examples of investigating steps beyond the RDS in catalysis without stoichiometric experiments so this may be quite a divisive statement.

In summary, I would recommend publication of this piece of work subject to major revisions of the mechanistic interpretations. Whilst I agree with many of the claims around the bimetallic reaction, this is only with respect to the stoichiometric studies and the extrapolation to the catalytic reaction

requires significant supporting information, particularly as there is clear and open evidence of a large disparity between the rates of stoichiometric and catalytic reactions.

Reviewer #3 (Remarks to the Author):

The authors present a research work on the mechanism of the Heck-Cassar alkyne coupling reaction using as catalyst palladium complexes with L = triphenylphosphine. This reaction is similar to the Sonogashira coupling reaction, but instead of using the bimetallic Pd/Cu system, only one metal, palladium, is used as the catalyst. Regarding the importance of the study, it is necessary to say that the Sonogashira reaction is extraordinarily efficient for a very wide scope of reagents. In the introduction the authors do not sufficiently explain what advantages, if any, the use of the Heck-Cassar system may have with respect to the Sonogashira reaction, so the reader cannot adequately understand the relevance of the study that follows.

The article is a continuation of the work presented by the same main author in DOI: 10.1038/s41467-018-0708, in which it is shown that the reaction mechanism consists of a bimetallic system in which the two catalytic cycles are operated by palladium and are interconnected by a transmetalation step.

The work is entitled "Elucidating the reaction mechanism of the palladium-palladium dual catalytic process through kinetic studies of the proposed elementary steps", and it presents a series of experiments that support the proposed mechanism, but other possible alternative mechanisms are not analyzed. The kinetic studies that are announced in the title are reduced to the measurement of the initial rates of several reactions, in particular of transmetalation reactions, but the authors do not extract values of rate constants, nor of the order of reaction in the different reagents, values of activation energies, equilibrium constants, etc. with which the kinetics of the reactions can be quantified. In particular, the lack of an experimental rate law prevents evaluating the mechanistic proposal and establishing the rate-determining step.

The mechanistic study is addressed by preparing some of the intermediates proposed in the working hypothesis, and verifying that they produce the expected reaction. It is verified that the addition of the oxidant ($\text{Pd}(\text{PPh}_3)_4 + \text{ArX}$) occurs and that in a basic medium complexes of the type PdArXL_2 react with alkynyls to give the palladium(II) alkynyl derivatives. These two reactions are well known and do not deserve comment. Then the article focuses on transmetalation.

The transmetalation reactions have been monitored by ^{31}P and ^1H NMR. Many unidentified signals appear in the trace spectra shown in the supplementary material. All signals at significant concentrations should be identified, since they may be reaction intermediates in the mechanism or important by-products. For instance, there is no reason to think that the reaction between $\text{trans-}[\text{Pd}(\text{CCR})\text{XL}_2]$ and $\text{trans-}[\text{Pd}(\text{CCR})_2\text{L}_2]$ to give $\text{cis-}[\text{Pd}(\text{CCR})_2\text{L}_2]$ and subsequent reductive elimination does not take place. Also, the relatively inert product $\text{trans-}[\text{PdAr}(\text{CCR})\text{L}_2]$ could be formed in the transmetalation reactions between $\text{trans-}[\text{PdArXL}_2]$ and $\text{trans-}[\text{Pd}(\text{CCR})_2\text{L}_2]$. Do these signals appear in NMR?

In the concentration/time plots only the first few minutes of the reaction are reported, so it is not possible to verify properly how the reaction evolves.

In general, there is a lack of discussion and interpretation of the data. For example, it is stated several times that the rate of transmetalation depends on the halogen in the $\text{trans-}[\text{PdArXL}_2]$ complex, so that it decreases in the order $\text{Cl} > \text{Br} > \text{I}$, but no explanation is given for this fact. Also, it is not explained why the transmetalation with $\text{trans-}[\text{Pd}(\text{CCR})_2\text{L}_2]$ is faster than with $\text{trans-}[\text{Pd}(\text{CCR})\text{XL}_2]$. Nor is it explained why alkyl alkynyls transmetallate more slowly than aryl alkynyls. A Hammett plot has been constructed in order to analyze the effect of substituents on aryls (Ar) and alkynyls (CCR),

but the results are not interpreted. What is the cause of the increase in speed with the higher electron density of the aryls in trans-[PdArXL₂]? Why do aryl alkynyls react faster when bearing EW groups? By the way, it should be stated whether the Hammett plot is built from rate constants or from initial rates.

The discussion of the stereochemistry of the transmetallation is completely ignored, as if it were not of the slightest importance.

Along the text there are several statements that are difficult to understand or to share, such as, for example, that the 6m complex (trans-[Pd(CC(2-Py))L₂]) gives unusually fast transmetallation reactions due to the coordinative capacity of pyridine. I can't imagine a transmetallation transition state where the pyridine is coordinated to the Pd receiving group while creating the Pd-C bond with the terminal carbon of the alkyne. It would be convenient to include a graph with the TS they propose. The authors cite an intermetallic complex with a short Pd-Pd distance, but in the proposed TS what is relevant is the C-Pd distance.

It is also stated that the reaction between complexes 6 and 7 gives rise to "Pd(PPh₃)₂". The observation of this complex as main Pd(0) species is extraordinarily unusual. PdL₂ complexes are relatively stable and can be isolated with very bulky phosphines (PCy₃, P(tBu)₃, etc.), but usually Pd(0) complexes with PPh₃ decompose until the amount of phosphine in the medium allows the equilibrium between [Pd(PPh₃)₃] and [Pd(PPh₃)₄]. [Pd(PPh₃)₂] is not usually an observable species. Can the authors give the reference from which they have extracted the chemical shift of this isolated species? Is decomposition to metallic palladium observed in their reactions?

On lines 272 and following it is stated that "4-coordinated palladium oxidative addition complexes, such as 7, are more likely to undergo dissociation of a PPh₃ ligand and subsequent reaction with the organometallic nucleophile than a direct associative reaction". I cannot agree with this statement. It is true that complexes like 7 can dissociate phosphine to give important intramolecular reactions, such as beta-hydrogen elimination, or reductive elimination, but intermolecular substitution reactions usually follow an associative mechanism. Another case would be if the authors had used extremely bulky phosphines, such as P(t-Bu)₃ or phosphines derived from biphenyls.

Line 281 states: "Based on the results of Hammett correlations as well as previous DFT calculations, we postulated that dissociation of the Pd-Pd complexes from 12 is most likely the rate-determining step of the process under investigation". I think that the DFT should support the experimental results, not the other way around, and that the Hammett plots are not evidence of this rds, which is far away from the step that involves RCC or Ar groups. The authors should support this statement with the experimental reaction rate law and with the measure of the activation entropy ΔS[‡].

From an experimental point of view, the measurement of the rates should be done in the presence of excess phosphine. This is quite obvious from the proposed reaction scheme in Figure 9. Due to the ability of the intermediate [Pd(PPh₃)₂] to capture PPh₃, the dissociation equilibrium of the first step (or the substitution equilibrium for the formation of the intermediate 9 by any mechanism) is progressively shifted to the right as the reaction progresses, meaning that the rate at each instant does not depend exclusively on the concentration of the reactants.

Finally, the authors compare the rate of the catalytic cycle with that of isolated reactions. They find that the rate under catalytic conditions is 19 times higher than that obtained in reductive transmetallation/elimination experiments. The authors do not satisfactorily justify this fact that questions their hypothesis, limiting themselves to saying that the conditions are not the same. In summary, the article presents a series of interesting experimental results, but its analysis and discussion do not meet the requirements of a journal such as Communications Chemistry, so I cannot recommend it to be accepted.

Response to Reviewers' comments regarding the manuscript entitled 'Elucidating reaction mechanism of palladium-palladium dual catalytic process through kinetic studies of proposed elementary steps'

Reviewer #1 (Remarks to the Author):

Comment 1:

The manuscript by Gazvoda and co-workers describes their efforts to uncover mechanistic insights associated to copper-free Sonogashira reaction, via a palladium-palladium transmetalation step. This is a follow-up to a previous work (reference 28: Nat. Commun. 9, 4814 (2018)) where the authors disclosed the involvement of a tandem Pd/Pd cycle. In the current manuscript, the authors carried out different kinetic experiments by NMR spectroscopy, based on proposed reactive intermediates, to identify the rate-determining step of the catalytic process. Despite this work provides some interesting mechanistic and organometallic features, there are key points that need to be further clarified. So, in its current form, I cannot recommend its publication.

Reply 1:

We thank the Reviewer for comments and issues raised, which helped improve the manuscript. Below we address each of the points raised, which we have also additionally discussed in the manuscript and Supplementary Information.

Comment 2:

1) I agree with the authors that there are few reports that postulate Pd/Pd catalytic systems (even when these dual systems could be responsible of additional already known transformations). However, the authors did not acknowledge properly previous fundamental studies on group exchange between PdII complexes that are relevant for this work. Espinet (Organometallics 16, 5730-5736 (1997)) along with Yamamoto (ref. 72 and Organometallics 16, 5354-5364 (1997)) have investigated the group exchange between Pd complexes, including the transfer of alkynyl groups (the main theme of this manuscript). The authors should include the two missing reference in the manuscript, and describe them in the introduction.

Reply 2:

We thank the Reviewer for pointing out the two missing references, which we cited in the introduction part of the manuscript and additionally described them in the section Transmetalation reactions.

Comment 3:

2) It is particularly strikingly to me that the authors did not provide any concentration-time plot showing the disappearance and formation of all the species involved along the different reactions (catalytic and stoichiometric). In the concentration-time data, they only showed the formation of the organic product. It would be very helpful to see the reaction evolution from the organometallic perspective.

Reply 3:

In the previous submission, the concentration-time plots for the reagents, i.e. oxidative addition complexes (**7**), palladium bisacetylides (**6**) and product formation (**3**) have been described in the kinetic studies of the transmetallation reaction in the Supplementary Information. As suggested by the Reviewer, we have included an example of concentration-time diagram for all relevant organometallic species, i.e., compounds **7**, **6**, **5**, and the tolan product **3**, for the reaction between **6m** and **7g** (Fig. 6a in the revised manuscript, see also page S53).

Comment 4:

Moreover, in the provided NMR spectra related to the transmetalation reactions (S52-64), there are additional species that can be observed by ^{31}P NMR that the authors didn't assigned. What can they comment on that?

Reply 4:

The transmetalation reactions between **6** and **7** are clean for up to 1-2 hours, as shown by the enclosed ^{31}P NMR spectra. At the onset of the transmetalation reaction, only the cross-coupled product **3** is formed, together with the monoacetylde **5** and $\text{Pd}(\text{PPh}_3)_2$, as we indicated in the manuscript. Detailed analysis of the ^1H NMR spectra when monitoring transmetalation reaction confirmed that only the above-mentioned species are present at the beginning of the transmetalation reaction. After 2 hours, additional ^{31}P resonances appeared, which could be due to decomposition of complexes and side reactions between the species formed; however, these species are not relevant to the reactions studied, so we do not assign them further. This was already stated in the manuscript and is now additionally emphasised.

Comment 5:

3) In their previous work (Ref. 28 in the manuscript), the authors monitored the catalytic reactions at different reactions times by ^{31}P NMR spectroscopy. In the NMR spectra, they observed the formation of different PdII complexes, including $\text{trans-}[\text{Pd}(\text{Ar})(\text{CCPh})(\text{PPh}_3)_2]$. This complex could be formed by the group exchange between complexes **6** and **7**, and after an isomerization step, could provide the desired C-C coupling.

a. How do they explain that they observed this compound under catalytic conditions but they didn't observe in the stoichiometric experiments? They should address this discrepancy.

Reply 5:

The $\text{trans-}[\text{Pd}(\text{Ar})(\text{CCPh})(\text{PPh}_3)_2]$, species B of the proposed mechanism, is indeed most likely the resting state resulting from isomerization under the specific catalytic conditions used in Nat. Commun. **9**, 4814 (2018), Ref. 28 in the previous version of the manuscript, Ref. 32 in the revised manuscript, i.e., Reaction a, $\text{Pd}(\text{PPh}_3)_4$, NaOMe, DMF, 50 °C, as indicated in our previous work. As indicated in the reference, the high loading (20 mol%) of the PPh_3 ligand precatalyst was used to allow monitoring of the phosphorous species by ^{31}P NMR. The reaction conditions described, i.e., the high loading with and excess of PPh_3 , DMF, and NaOMe, were likely the reason for decreasing the rate of the reductive elimination step and allowed the isomerization to $\text{trans-}[\text{Pd}(\text{Ar})(\text{CCPh})(\text{PPh}_3)_2]$, trans-B.

Comment 6:

b. Is this related to the fact that they used a different solvent to perform the kinetic study (catalysis in DMF, kinetic study in CDCl₃)?

Reply 6:

The difference most likely results from the choice of reaction conditions, i.e. solvent, base, concentration of palladium precatalyst and PPh₃.

Comment 7:

c. In this regard, CHCl₃ is not used as solvent in Cu-free Sonogashira cross-couplings (they should include this recent review on the topic: RSC Adv. 2021, 11, 6885), while DMF is. Why did they select CDCl₃ for the kinetic study? Especially when in their previous work they authors showed that CDCl₃ is not innocent (they observed the formation of Pd(PPh₃)₂Cl₂ when they dissolved Pd(PPh₃)₄ in this solvent).

Reply 7:

We thank the Reviewer to indicate RSC Adv. **11**, 6885-6925 (2021), we have included the reference in the manuscript. The indicated review article is more focused on preparative aspects of palladium catalysed coupling between aryl halides and terminal alkynes, including also examples of heterogenous catalysis. The choice of CDCl₃ as the solvent for monitoring the reactions was based mainly on the solubility of the Pd species studied in this solvent. The CDCl₃ was previously used as a solvent for monitoring similar reactions (Organometallics **16**, 5730–5736 (1997)). The studied palladium-phosphorous species are soluble in CDCl₃ at a concentration of 0.01 M (the concentration comparable to that of Pd in catalytic reactions and also still sufficiently high to allow monitoring of the reactions by NMR). In contrast, most of the Pd species studied are not soluble at these concentrations in solvents such as DMF. Moreover, some of the palladium complexes, such as oxidative palladium addition complexes, have already been described in the literature in CDCl₃ as a solvent, which facilitates the comparison of the observed ³¹P NMR resonances with the literature reports.

In our earlier paper (Nat. Commun. **9**, 4814 (2018)); Ref. 28 in previous version of the manuscript, Ref. 32 in the revised manuscript), we monitored the quality of Pd(PPh₃)₄ from different manufacturers to show the importance of using the same batch of Pd(PPh₃)₄ when performing catalytic reactions, as the reproducibility of catalytic reactions has been shown to be highly dependent on batch quality. Pd(PPh₃)₄ is manufactured from Pd(PPh₃)₂Cl₂, which remains in the sample when the conversion into Pd(PPh₃)₄ is not complete. However, after some time, Pd(PPh₃)₂Cl₂ starts to form from Pd(PPh₃)₄ in CDCl₃.

We have shown that by preparing fresh solutions of the same pure sample of Pd(PPh₃)₄ in different solvents (see spectra below) in benzene-*d*₆, THF-*d*₈, and in CDCl₃, showing the resonances that corresponds to Pd(PPh₃)₄ (δ +27 ppm) along with a broadened resonance corresponding to Pd(PPh₃)₃. As we discussed in our previous paper (see page S42 in Supplementary Information in Nat. Commun. **9**, 4814 (2018)), the broad resonance at +4.8 ppm was ascribed to Pd(PPh₃)₃. This has been confirmed by adding PPh₃ to the solution of Pd(PPh₃)₄, shifting the resonance at +4.8 ppm closer to –5 ppm. This observation is in agreement with the literature report (Eur. J. Org. Chem., 366-371 (2004)).

In CDCl_3 , $\text{Pd}(\text{PPh}_3)_2\text{Cl}_2$ was indeed formed, but in small amounts, i.e., 4% after 15 min, 10% after 2 h, suggesting that the large amount of $\text{Pd}(\text{PPh}_3)_2\text{Cl}_2$ in the ^{31}P NMR spectra reported in our previous paper was actually from residual $\text{Pd}(\text{PPh}_3)_2\text{Cl}_2$ of the sample.

$\text{Pd}(\text{PPh}_3)_4$ in $\text{THF-}d_8$:

$\text{Pd}(\text{PPh}_3)_4$ in $\text{benzene-}d_6$:

$\text{Pd}(\text{PPh}_3)_4$ in CDCl_3 :

It is noteworthy that CDCl_3 is well tolerated towards the studied palladium key species **5**, **6** and **7**, as the samples did not change noticeably when the compounds were analysed/characterised by NMR (^1H , ^{31}P , ^{13}C) and the experiments, which took longer time, gave clean spectra of these complexes (please see Supplementary Information, section Copies of NMR spectra).

Comment 8:

4) The authors mentioned that the oxidative addition step is instantaneous. For supporting the first statement they referenced their previous work (Ref. 28 in the manuscript) and the supporting information of this manuscript. However, I couldn't find any systematic kinetic study in any of them that support this fact. In the previous work, the authors only recorded the NMR spectra for the oxidative addition of **1** and $\text{Pd}(\text{PPh}_3)_4$.

Reply 8:

It is known from the literature that oxidative addition of aryl iodides to Pd^0 species occurs instantaneously (J. Organometal. Chem. **208**, 1981, 419–427; Organometallics, **9**, 1990, 2276–2282). For the studied example of oxidative addition of 4-iodotoluene to Pd^0 - PPh_3 -based species to form $\text{trans-Pd}(\text{PPh}_3)_2(\text{Ar})(\text{I})$, we performed the reaction between 4-iodotoluene and $\text{Pd}(\text{PPh}_3)_4$ in CDCl_3 and recorded ^1H and ^{31}P NMR spectra after the addition of aryl iodide to the solution of $\text{Pd}(\text{PPh}_3)_4$ in CDCl_3 . As judged by NMR, the composition of the reaction mixture did not change in time as compared to the initial spectra recorded after 10 min at RT. The NMR spectra recorded after 10, 30 and 60 min were

identical, indicating that the equilibrium was formed within a few minutes, and did not change thereafter. Thus, we concluded that the oxidative addition of Ar-I (for the studied example of 4-iodotoluene) to Pd⁰ is very rapid (instantaneous) and indeed much faster than the other steps studied.

Similarly, we performed oxidative addition of 4-iodobenzofenone **1m** to Pd(PPh₃)₄, which gave identical results - complex **7q** was formed within 10 minutes after addition of **1m**, proving that oxidative addition of aryl iodides is rapid regardless of the electronic effects of the para substituent on aryl ring (see page S151 in Supplementary Information).

Comment 9:

In this manuscript, the authors synthesized a broad array of oxidative addition complexes, but the reaction conditions that they employed involved long reaction times (24 h for the RI), heating (for RBr) or solvents that are not related to the catalytic or stoichiometric experiments. Thus, how can they be sure that, in all cases, the oxidative addition is immediate without monitoring the reactions?

Reply 9:

Palladium oxidative addition adducts **7** (Fig. 5 in the manuscript) were prepared and characterized as starting materials for transmetalation reactions with **6**, to investigate the influence of the substituents in **7** (i.e., the halide ligand and the substituents on the phenyl rings) on the course of the transmetalation reaction. The kinetics of their formation were not relevant for this purpose. Only the 4-tolyl and 4-acetyl iodide substrates were studied in more detail, as described above (see Reply 8), since oxidative addition of 4-iodotoluene was one of the elementary steps of the studied catalytic reaction between **1a** and **2a**. This was also indicated and described in the manuscript.

We indicated in the manuscript that “the rate of oxidative addition of aryl or pseudoaryl halides (Ar-X) usually follows the dissociation enthalpies of the corresponding bonds (C-X), and in this concept the order of reactivity is Ar-I > Ar-Br ≈ Ar-OTf > Ar-Cl” (Ref. 74 in the previous version of the manuscript; Ref. 84 in the revised manuscript). And that “this indicates that replacing the aryl iodide starting substrate with aryl bromide or chloride would completely change the rates of the elementary steps in the catalytic reaction.” Thus, we only indicated that oxidative addition of 4-iodotoluene was instantiations/very rapid.

Comment 10:

5) In this manuscript, the authors mentioned that the cis isomer, which is initially formed after the oxidative addition event, could not be observed due to the higher thermodynamic stability of the cis isomer, supporting this statement in a work by Espinet and co-workers (ref. 73). It is true that the Espinet group observed that after the oxidative addition, trans compounds are the thermodynamically more stable isomers. However, in this article, the authors also were capable to observe the formation and even isolate a cis compound, cis-[Pd(C6Cl2F3)I(PPh₃)₂], after an oxidative addition event, and subsequently study its isomerization to the corresponding trans isomer. Indeed, few years later, the Espinet group showed the potential relevance of these cis species in transmetalation events during the Stille coupling of alkynyl stannanes and aryl iodides (Organometallics 2011, 30, 611-617), Did Gazvoda and co-workers consider this possibility?

Reply 10:

As indicated in the Reply 9, the oxidative addition of 4-iodotoluene and Pd(PPh₃)₄ proceeds within 10 minutes, and the NMR-monitored reaction mixture does not change after this time. We were able to assign all ³¹P resonances in the NMR spectrum of the reaction mixture consisting of the palladium oxidative addition complex *trans*-Pd(PPh₃)₂(tolyl)(I) (**7a**), Pd(PPh₃)₄ and PPh₃, along with side species, i.e., O=PPh₃, Pd(PPh₃)₂(tolyl)₂, **7g** and Pd(PPh₃)₂I₂, which are results of the side reactions (see page S149).

The observation and the ability to isolate the *cis*-isomer reported by Espinet and co-workers is a result of using a specific substrate with specific substituents on the aryl ring (C₆Cl₂F₃I). In addition, the *cis* isomer has very characteristic ³¹P NMR spectra due to non-equivalent PPh₃ groups: ³¹P NMR δ 29.6/33.3/33.4 (ddt), 16.7/21.0/21.0 (ddt) (Organometallics 1998, 17, 954-959) and is therefore difficult to overlook. We neither observed these type of coupling patterns in the spectra of the oxidative addition of 4-tolyl iodide or 4-iodoacetophenone to Pd(PPh₃)₄ nor in the synthesized palladium complexes **7a-7q**, all of which showed a ³¹P resonance characteristic for the *trans* configuration of complexes **7**. The ³¹P NMR spectra of complexes **7** were in agreement with the literature values.

As we describe in the manuscript and in our earlier work (Nat. Commun. **9**, 4814 (2018)); Ref. 28 in previous version of the manuscript, Ref. 32 in the revised manuscript), the dissociative mechanism in which PPh₃ dissociates from Pd(PPh₃)₂(Ar)(X) to form Pd(PPh₃)(Ar)(X), which is the reactive species, is the most likely (this has also been observed by others for similar transmetallation processes and by us, see Ref. 28 in previous version of the manuscript, Ref. 32 in the revised manuscript and the references cited therein). However, we have considered and cited the reference given above (Organometallics **30**, 611-617 (2011)) and indicated in the manuscript that *cis*-species **7** might be involved in the transmetallation step.

Comment 11:

In addition, in this work, Espinet showed that the isomerization reactions can become the rate-determining step of a catalytic cycle. As result, they observed the accumulation of a *trans* complex that is completely analogous to the *trans*-[Pd(Ar)(CCPh)(PPh₃)₂] that Gazvoda observed in their Nat. Commun. Paper. I highly recommend the authors to explore this option, especially considering that they didn't include any isomerization step in their proposed catalytic cycle.

Reply 11:

The isomerization steps have been omitted for clarity and to simplify the presentation of the mechanistic pathway, which we have pointed out in the description of the figures. As explained above and in the manuscript, the *trans*-[Pd(Ar)(CCPh)(PPh₃)₂] species was not observed in any of the transmetallation reactions under study when monitoring them by NMR, and such a species was only observed in the catalytic reaction (Reaction a in Ref. 28 in previous version of the manuscript, Ref 32 in the revised manuscript; please see Reply 5) with specific reaction conditions. Neither *trans*-[Pd(Ar)(CCPh)(PPh₃)₂] nor *cis*-[Pd(Ar)(CCPh)(PPh₃)₂] thus accumulates during transmetallation reaction between **6** and **7**, suggesting that reductive elimination proceeds more rapidly than the transmetallation step (see discussion in the manuscript for Fig. 6 and Fig. 9).

Comment 12:

6) For confirming the dissociation of PPh₃ during the reaction process, the authors should investigate if the catalytic and stoichiometric reactions are slowed down by the addition of PPh₃.

Reply 12:

We thank the Reviewer for this comment. We performed the stoichiometric reaction between **6a** and **7a** in the presence of 4.4 equivalents of PPh₃, which stopped the reaction completely (see S161–162 in Supplementary Information). Similarly, the addition of 0.1 and 0.5 equivalents of PPh₃ to the catalytic reaction also dramatically slowed the reaction (see S163–166). This further indicates the dissociative mechanism hypothesis. We have included these observations in the manuscript and Supplementary Information.

Reviewer #2 (Remarks to the Author):

Dear Editor,

Thank you for forwarding me this interesting paper investigating the mechanism of a copper-free palladium catalysed Sonagashira reaction. It is clear that significant time and effort went into the completion of this study, and insights into the mechanism of such reactions are of great interest to the general community. Whilst it is clear that the experimental work was completed to an extremely high standard, I believe there are some open questions about the conclusions that may need further investigation.

We thank the Reviewer for the positive comment and reply to their indicated issues below point by point. We have addressed the issues raised for which we have also performed additional experiments and included the results in the manuscript and Supplementary Information.

Comment 1:

It's clear from the experimental work, that under stoichiometric conditions, a mixture of bisacetylide (**6**) and oxidative addition complexes (**7**), can lead to the formation of tolan products (**3**). The use of the Hammett plot to investigate the proposed transmetallation gives interesting insights, with a weak and opposing contribution from each reactant, and I think additional explanation here would be valuable as it is quite difficult to explain. The proposition that dissociation of (**12**) to the pre-reductive elimination complex may be consistent with the Hammett plot results, but one may then expect accumulation or observation of this species which was not apparently observed.

Reply 1:

The results of the Hammett plots indicate that both the Pd oxidative addition complex **7** and the palladium bisacetylide **6** are directly involved in the rate-determining step of the transmetallation process. As the Reviewer further notes, the substituents on aryl groups have a counterintuitive effect on the respective reagents, i.e., electron-donating groups on the palladium oxidative addition adduct, which is usually considered as electrophilic reagent, and electron-withdrawing groups on the palladium bisacetylide, which is an organometallic nucleophile, facilitate the reaction. Thus, the observed

Hammett plots can be attributed to the dissociation of the bimetallic complex **12** in Figure 9. We did not observe the accumulation of the bimetallic complex **12** (or other species, as evident from ^{31}P NMR spectra of transmetallation reactions) while monitoring the transmetallation reaction, which is perhaps not entirely unexpected, since the proposed structure of intermediate **12** may be considered an unstable intermediate/transition state.

Comment 2:

I think a few experiments may help here: 1) looking at the order in each reactant with the expectation of an overall second order reaction if the bimolecular reaction is rate limiting,

Reply 2:

We thank the Reviewer for this suggestion. We performed additional experiments to determine the order in palladium oxidative addition complex and palladium bisacetylide, i.e., **6a** and **7a**, in the transmetallation reaction studied. We performed the experiments by varying the concentrations of one of the reagents in 3 different concentrations (0.005, 0.015 and 0.025 M). For both **6a** and **7a**, the observed order of the reagents turned out to be 1, indicating that both reagents are involved in the rate-determining step of the process. We have included this observation in the manuscript and Supplementary Information (see S167–171).

Comment 3:

2) doing competition experiments for the complexes to see if the Hammett by competition is identical to the Hammett by initial rate (i.e. have a mixture of **7b** and **7k** with limiting **6a**, and use the ratio of the products formed to define the relative rates by competition), this will help determine if the selectivity determining step (presumably pi-complexation by Fig 9) is different from the rate determining step. The same could be done with two bisacetylide complexes with limiting oxidative addition complex (**7**). Point 2) may be quite work intensive if the complexes have expired and I do not think repeating the whole Hammett plot by competition would be necessary, but a few examples may be informative.

Reply 3:

We thank the Reviewer for this suggestion. We performed the proposed competition experiments using a mixture of palladium oxidative addition complexes **7a** and **7c** carrying Me and NO_2 substituents, respectively, with the palladium bisacetylide **6a** as the limiting reagent. The results of formation of the corresponding tolan **3** products were comparable to the observed Hammett plots/rates, indicating that the selectivity-determining step is not different from the rate-determining step.

Similarly, we performed transmetallation reactions of mixtures of palladium bisacetylides **6c** and **6k** bearing Me and NO_2 groups, respectively, using the oxidative palladium addition complex **7k** as the limiting reagent. Again, the formation of the tolan products **3** was similar to the formation rates of the tolan products of Hammett plots/rates.

We have included these observations in the manuscript and Supplementary Information (see S176181 in Supplementary Information).

Comment 4:

I think there also needs to be justification for the use of “corrected sigma values” which appear to use the sigma minus constants. The SI shows the plots with both, and I do not think using the normal sigma values will change the conclusions significantly but use of sigma minus implies significant resonance transferring in the transition state so would need further explanation.

Reply 4:

As indicated by the Reviewer, the conclusion of the Hammett plots is the same when using the original or the 'corrected' sigma values. A slightly better correlation was observed with the 'corrected' sigma values, which could be due to potential resonance stabilization in the proposed intermediate **12** (see Figure 9). However, since the plots are comparable and to avoid additional unsubstantiated hypotheses, we have included the plots with the original sigma values in the manuscript and the plots with the 'corrected' sigma values in the Supplementary Information. We have omitted the term 'corrected' sigma values and used σ_p^- (σ_p^-) constants instead.

Comment 5:

The kinetics of the stoichiometric reactions are also quite unusual and appear to either slow or stall which is curious for a reaction that works well under catalytic conditions (plotting both starting material as well as product concentration in the main body would help make this obvious to the reader, although it is shown in the SI). In Fig 6, there appears to be quite an accumulation of Pd(PPh₃)₂Cl₂ after 4 hours, but this is not discussed, is the formation of this species coincident with slowing of the reaction?

Reply 5:

We believe that the kinetics of the stoichiometric reactions under the conditions studied (0.01 M concentration of reagents; with concave curves and maximum rates at the beginning of the reactions) are as expected. As described in the manuscript, the transmetalation reactions proceeded cleanly, producing only the cross-coupled tolan product **3**, the monoacetylide **5**, and Pd(PPh₃)₂**8** up to 1-2 hours, as shown in the ³¹P NMR spectra in the manuscript (Fig. 6) and in Figs. S1-S10 in the Supplementary Information. Also, when monitoring the reactions by ¹H NMR, no byproducts were formed up to 1-2 hours reaction. This is also the time window in which the initial rates were determined, i.e. first 20-30 minutes after the start of the reaction. After that, the reactions slowed down, which could be due to lower concentrations of the reagents, their slow decomposition, and side reactions of decomposition products. As described in the manuscript, the ³¹P NMR spectra of the reaction mixture showed several other resonances after 4 hours. This was for the reader's information and not to investigate the side reactions in more detail, as they are not relevant to the present study.

Comment 6:

As the authors admit there is a large discrepancy between the catalytic and stoichiometric reaction rates, with the catalytic reaction going significantly faster and this needs further explanation. Although this could be due to the different reaction conditions as stated in the paper, I think we must be open to the possibility that an alternative mechanism could be proceeding as well.

Reply 6:

We thank the Reviewer for this comment, which we also consider one of the major weaknesses of the manuscript. The large discrepancy between the catalytic reaction and the stoichiometric transmetallation arose for the reason that **7** is actually a precursor of the catalytically active species, i.e., the tricoordinated palladium oxidative addition complex (which we have in fact indicated and noted in the manuscript 'Catalytically active species are usually not stable enough to be isolated, so their precursors are usually isolated. For example, Pd oxidative addition complexes **7**, trans-Pd(PPh₃)₂(Ar)(X) are isolated as precursors to catalytically active tricoordinated Pd(PPh₃)(Ar)(X) species. Precursors to catalytically active species are usually observed in reactions as the resting states of catalytic cycles' and drawn in Figure 9. Since the triphenylphosphine forms isolable palladium oxidative addition complexes in tetracoordinate form, we prepared the tricoordinated palladium oxidative addition complex Pd(P(*o*-tolyl)₃)(Ar)(I) **7a'** as a surrogate for tricoordinated Pd(PPh₃)(Ar)(I) species, which we reacted with **6a**. The reaction of tricoordinated palladium oxidative addition complex Pd(P(*o*-tolyl)₃)(Ar)(I) **7a'** proved to be considerably faster and nearly as fast as studied catalytic reaction despite the observed side reactions that can be attributed to relative instability of tricoordinated palladium complex **7a'**. For detailed information see Supp. Info. (S158–160). However, because of the relative instability of prepared tricoordinated **7a'**, tetracoordinated **7** are still more suitable for the investigation of the elementary reactions under study.

This indicates that the additional phosphine ligand, from tetracoordinated **7**, present in the stoichiometric transmetallation reaction, was the reason for the discrepancy in rates between the catalytic reaction and the stoichiometric transmetallation, which we also showed in the experiment when we added 4 equiv. to the transmetallation reaction between **6a** and **7a**, completely retarding the reaction.

We have included these observations in the manuscript and Supplementary

Information. Comment 7:

At present, I do not think there is sufficient evidence to suggest that the catalytic reaction proceeds via the bimetallic mechanism, and whilst the presence of the chloride salt accelerates the reaction in both stoichiometric and catalytic reactions, this is not conclusive support that they proceed via the same mechanism (as iodides can in general cause issues in palladium coupling reactions). I think to support this hypothesis, a greater look into the catalytic reaction is needed. If the reaction truly proceeds by a rate-limiting, bimolecular transmetallation of two palladium partners then the reaction kinetics would be pseudo-zero order, but overall second order with respect to palladium (i.e. rate vs [Pd] = 2nd order).

Reply 7:

We thank the Reviewer for this comment. The order of Pd in the catalytic reaction was discussed in our previous publication (Nat. Commun. **9**, 4814 (2018); Ref. 28 in previous version of the manuscript, Ref 32 in the revised manuscript), where the order for Pd in the catalytic reaction was found to be 1. This is not unexpected because, as we have shown, different palladium species dominate in the reaction mixture under different reaction conditions, favouring the formation of one of the key species **6** or **7**, implying that one of them is in large excess to another, and that the reaction rate depends on the formation of one of these reagents. This means that in such catalytic reaction one of the reagents

has zero order dependency, the other one, and consequently the total order in Pd in the catalytic reaction is 1.

Comment 8:

It is also unclear to me how the bisacetylide complex would enter the catalytic cycle if a Pd(0) precatalyst is used. Presumably with a fast oxidative addition to form 7a, most if not all Pd will be at this complex, and with free 2a available this could go directly to B? I've noted in the previous paper that the bisacetylide complexes could form from the oxidatively added complex, so looking at the kinetics of this process in the context of the rate $7a + 2a + \text{pyrr}$ should give some beneficial information on what is happening under catalytic conditions.

If a Pd(II) complex is used, depending on the activation mechanism, some may go to Pd(0), and some may form the bisacetylide complex, and I think this is a really interesting point and different kinetics could be observed using different precatalysts if some lead to bisacetylide complexes and some don't (or in differing ratios). A further analysis of this would be of great interest to the general community to determine if these complexes, which clearly form, are on- or off-cycle, or if they provide an alternative and slower pathway to coupling than the traditional mechanism. I believe some of this was discussed in the previous publication, so this could be brought into the main text or referred to in justification or rebuttal of claims.

Reply 8:

We thank the Reviewer for this comment which is an important one and which we discussed in our earlier paper (Nat. Commun. 9, 4814 (2018); Ref. 28 in previous version of the manuscript, Ref. 32 in the revised manuscript). As indicated by the Reviewer, both palladium species involved in transmetalation, **6** and **7**, require palladium precursors in different oxidation states for their formation, i.e., Pd(0) for the formation of the oxidative addition complex **7** and Pd(II) for the formation of palladium bisacetylide **6**. When performing the catalytic reactions, the palladium is introduced into the reaction in form of precatalyst, usually in the form of Pd(0) or Pd(II), e.g., Pd⁰(PPh₃)₄ or Pd^{II}(PPh₃)₂Cl₂. When Pd(PPh₃)₂Cl₂ is used as a precatalyst, the Pd(0) is formed after the formation of palladium bisacetylide and reductive elimination of the Hay-Glaser alkyne dimer, which initiates the formation of the oxidative addition complex and catalysis (Bull. Chem. Soc. Jpn., 77, 2033–2045 (2004)). When Pd(PPh₃)₄ is used as a precatalyst, the formation of Pd(PPh₃)₂(Ar)(X) and further transmetalation between the two species gives rise to Pd(PPh₃)₂(X)₂, a precursor for the formation of palladium bisacetylide, as shown in our previous work (see Ref. 28 in the previous version of the manuscript; Ref. 32 in the revised manuscript). In these two side reactions, the required Pd precursors are formed in small quantities and in this way the catalytic reaction is initiated. This is observed as an induction period present on the beginning of the catalytic reactions when either Pd(0) or Pd(II) is used as a precatalyst, which was described in our previous work and has also been observed by others (see Nat. Commun. 9, 4814 (2018); and references cited therein).

Comment 9:

A final editing point, I would consider revising the statement on line 28: "Although such mechanistic analysis can readily determine the reagents present in the rate-determining step, it provides little information about the mechanistic pathway, i.e., in which step which reagent enters the catalytic cycle

and how.” There are plenty of examples of investigating steps beyond the RDS in catalysis without stoichiometric experiments so this may be quite a divisive statement.

Reply 9:

We omitted the sentence from the manuscript.

Comment 10:

In summary, I would recommend publication of this piece of work subject to major revisions of the mechanistic interpretations. Whilst I agree with many of the claims around the bimetallic reaction, this is only with respect to the stoichiometric studies and the extrapolation to the catalytic reaction requires significant supporting information, particularly as there is clear and open evidence of a large disparity between the rates of stoichiometric and catalytic reactions.

Reply 10:

We thank the Reviewer for constructive comments, especially for pointing out the discrepancy in rates between catalytic and stoichiometric reactions. We hope that we have adequately addressed the issues raised and that the manuscript is suitable for publication.

Reviewer #3 (Remarks to the Author):

Comment 1:

The authors present a research work on the mechanism of the Heck-Cassar alkynylation reaction using as catalyst palladium complexes with L = triphenylphosphine. This reaction is similar to the Sonogashira coupling reaction, but instead of using the bimetallic Pd/Cu system, only one metal, palladium, is used as the catalyst. Regarding the importance of the study, it is necessary to say that the Sonogashira reaction is extraordinarily efficient for a very wide scope of reagents. In the introduction the authors do not sufficiently explain what advantages, if any, the use of the Heck-Cassar system may have with respect to the Sonogashira reaction, so the reader cannot adequately understand the relevance of the study that follows.

The article is a continuation of the work presented by the same main author in DOI: 10.1038/s41467018-0708, in which it is shown that the reaction mechanism consists of a bimetallic system in which the two catalytic cycles are operated by palladium and are interconnected by a transmetalation step.

Reply 1:

We thank the Reviewer for pointing out the importance of the Sonogashira reaction, which is indeed still the reaction of choice for the synthesis of disubstituted alkyne products due to its efficiency. However, the original Pd/Cu Sonogashira reaction requires the use of copper as a co-catalyst, the major drawbacks of which are use of environmentally unfriendly reagents and the formation of Hay-Glasser alkyne homocoupling products (this was additionally indicated in the introduction of the manuscript). Therefore, the copper-free variant, known as the copper-free Sonogashira cross-coupling or Heck-Cassar alkynylation, is gaining attention because it uses only palladium as a catalyst. Until recently, this reaction was believed to proceed via a monometallic mechanism. However, we have recently proposed that the reaction may actually proceed via two palladium catalytic cycles connected by a

transmetallation step. Understanding the reaction mechanism is crucial for the rational development of the reaction, which could establish palladium-catalyzed alkynylation as the reaction of choice for the synthesis of disubstituted alkynes in the future. In the present study, the mechanism of this reaction is analyzed with special attention to the formation of palladium bisacetylides as reactive reagents in the catalytic process, which transfer the alkyne component to the oxidative palladium addition complex in the transmetallation step, which is also analyzed in detail.

Comment 2:

The work is entitled “Elucidating the reaction mechanism of the palladium-palladium dual catalytic process through kinetic studies of the proposed elementary steps”, and it presents a series of experiments that support the proposed mechanism, but other possible alternative mechanisms are not analyzed. The kinetic studies that are announced in the title are reduced to the measurement of the initial rates of several reactions, in particular of transmetalation reactions, but the authors do not extract values of rate constants, nor of the order of reaction in the different reagents, values of activation energies, equilibrium constants, etc. with which the kinetics of the reactions can be quantified. In particular, the lack of an experimental rate law prevents evaluating the mechanistic proposal and establishing the rate-determining step.

Reply 2:

As we answer in detail below, one of the key purposes of the study was to determine the mechanistic pathway of the reaction, i.e., in which step which reagent enters the catalytic cycle and how. Specifically, that the alkyne enters the catalysis via activation into palladium bisacetylde, which transfers the alkyne reagent to the palladium oxidative addition complex in the transmetallation step (the alternative mechanistic proposal in which the alkyne reagent enters the catalytic cycle via ligand exchange in the palladium oxidation addition complex, i.e., comparison of the reactivity of alkyne and palladium bisacetylde in reaction with the palladium oxidation addition complex, was discussed in our previous study, *Nat. Commun.* **9**, 4814 (2018); Ref. 28 in previous version of the manuscript, Ref. 32 in the revised manuscript). These two steps, i.e., the formation of palladium bisacetylde, and the transmetallation between palladium bisacetylde and palladium oxidative addition complexes, which are in contrast to the previously proposed mechanisms, are studied in detail in the present manuscript. It is noteworthy that these elementary reactions of the mechanism have not been studied in detail before. We have included the analysis of the reaction between **6a** and **7a** with the calculation of the orders in each reactant, the calculation of the rate constant, its dependence on temperature, and the subsequent calculation of the transition state enthalpy and entropy, as suggested by the Reviewer. However, we believe, that the specification and extraction of other kinetic parameters is beyond the scope of the present study. We believe that the initial rate method is adequate for comparing the rates of the elementary steps of the proposed mechanisms because it allows comparison under similar reaction conditions. As indicated above, the alternative mechanisms were discussed in detail in our previous study (*Nat. Commun.* **9**, 4814 (2018); Ref. 28 in previous version of the manuscript, Ref. 32 in the revised manuscript), in particular the monometallic proposal that was previously postulated as the most likely. Other mechanistic proposals have been previously studied by others and have been omitted experimentally and/or theoretically and are also discussed in our previous study (see *Nat. Commun.* **9**, 4814 (2018); Ref. 28 in previous version of the manuscript, Ref 32 in the revised manuscript and references therein).

As we have indicated in several places in the manuscript, the conclusions about the mechanism are based on the examples studied and that under completely different conditions (or by employing different substrates in the catalytic reaction) the rates of the elementary steps would change, and to leave room for competing mechanisms we have added to the conclusion the sentences: "It should be noted that under different conditions (e.g., different ligands and reaction conditions), other mechanisms could compete with or replace the one described here. Similarly, the use of different aryl halide and alkyne starting reagents would completely change the rates of the elementary steps of the catalytic reaction."

Comment 3:

The mechanistic study is addressed by preparing some of the intermediates proposed in the working hypothesis, and verifying that they produce the expected reaction. It is verified that the addition of the oxidant ($\text{Pd}(\text{PPh}_3)_4 + \text{ArX}$) occurs and that in a basic medium complexes of the type PdArXL_2 react with alkynyls to give the palladium(II) alkynyl derivatives. These two reactions are well known and do not deserve comment. Then the article focuses on transmetalation.

Reply 3:

As the Reviewer notes, the oxidative addition of aryl halides to $\text{Pd}(\text{PPh}_3)_4$ is indeed well established to generate an oxidative addition complex. We have described in detail the base-mediated reaction between $\text{Pd}^{\text{II}}(\text{PPh}_3)_2(\text{X})_2$ species and alkynes to form palladium bisacetylides **6**, which is general and has not previously been described in as much detail as in the present manuscript.

Comment 4:

The transmetalation reactions have been monitored by ^{31}P and ^1H NMR. Many unidentified signals appear in the trace spectra shown in the supplementary material. All signals at significant concentrations should be identified, since they may be reaction intermediates in the mechanism or important by-products.

Reply 4:

We have indicated and clearly shown in the manuscript and in Supplementary Information that the transmetalation reactions between **6** and **7** are clean from the start of the reaction up to 1 (sometimes up to 2) hours, producing only the expected palladium bisacetylide **5** as a product of transmetalation and the tolan product **3** and $\text{Pd}(\text{PPh}_3)_2$ as products of reductive elimination, and therefore cannot agree with the Reviewer's statement. All transmetalation reactions have the maximum rate at the onset of the reaction and concave curves. All this indicates the direct productivity of the transmetalation reaction between **6** and **7** into expected disubstituted alkyne product **3**. The first 20-30 minutes of the reactions were considered in the extraction of reaction rates. The ^1H and ^{31}P NMR spectra of the reaction progress after 2-4 hours were included to inform the reader that after this time side reactions start to occur, but they are not relevant to the reaction studied, and therefore (all) side resonances after 2-4 hours were not assigned in the ^{31}P NMR spectra. As indicated above, we have included sentence 'After 2-4 hours, additional ^{31}P resonances began to appear, which could be due to decomposition of complexes and side reactions between the species formed, however, these species are not relevant to the reactions studied, so we did not assign them further' in the manuscript to additionally stress this.

Comment 5:

For instance, there is no reason to think that the reaction between trans-[Pd(CCR)XL₂] and trans-[Pd(CCR)₂L₂] to give cis-[Pd(CCR)₂L₂] and subsequent reductive elimination does not take place.

Reply 5:

The cis-[Pd(CCR)₂L₂, L = PPh₃] palladium bisacetylides have not been observed and reported in the literature, most likely because they are not stable and produce an alkyne homo coupling product (Hay-Glaser product) and Pd(0) after reductive elimination. However, this side reaction occurs in small amounts in the catalytic reaction to generate Pd(0) species (Bull. Chem. Soc. Jpn. **77**, 2033–2045 (2004), Organometallics **32**, 4192–4198 (2013)), which are required for the oxidative addition of Ar-X when Pd(II) is used as a precatalyst for the reaction. This is also the reason for the induction time in this reaction previously observed by us and others (see Nat. Commun. **9**, 4814 (2018); and the discussion there).

Trans-[Pd(CCR)₂L₂, L = PPh₃] is otherwise stable in solution, as shown in Supplementary Information, see Fig. S18). The NMR spectra show that the compound does not change over time. It is noteworthy that we did not observe the formation of an alkyne homocoupling product (Hay-Glaser product) by NMR during either the stoichiometric or catalytic reactions during the study.

Comment 6:

Also, the relatively inert product trans-[PdAr(CCR)L₂] could be formed in the transmetallation reactions between trans-[PdArXL₂] and trans-[Pd(CCR)₂L₂]. Do these signals appear in NMR?

Reply 6:

As we discussed in the manuscript, the signals for trans-[Pd(Ar)(CCR)L₂] were not detected in the ³¹P NMR spectra when the transmetallation reactions were observed between **6** and **7**, suggesting that reductive elimination proceeds faster than transmetallation (please see discussion by Figures 6 and 9).

Comment 7:

In the concentration/time plots only the first few minutes of the reaction are reported, so it is not possible to verify properly how the reaction evolves.

Reply 7:

Concentration-time plots for the transmetallation reactions are reported up to 4 hours (please see Supplementary Information, S84-S105), the first 20-30 min are then highlighted in separate graphs from which initial rates were determined. It can be seen that the reactions have concave curves with a maximum rate at the beginning of the reaction. One example of concentration/time plot of all relevant species **3**, **5**, **6** and **7** of transmetallation of **6m** and **7k** was also additionally included in the manuscript (Fig. 6a).

Comment 8:

In general, there is a lack of discussion and interpretation of the data. For example, it is stated several times that the rate of transmetallation depends on the halogen in the trans-[PdArXL₂] complex, so that it decreases in the order Cl > Br > I, but no explanation is given for this fact. Also, it is not explained

why the transmetallation with trans-[Pd(CCR)₂L₂] is faster than with trans-[Pd(CCR)XL₂]. Nor is it explained why alkyl alkynyls transmetallate more slowly than aryl alkynyls.

Reply 8:

Since the present manuscript reports the first detailed study of Pd-Pd transmetallation of this kind, the main idea was to show that the transmetallation studied is general, in process we aimed to describe the effect of substituents X or Ar in **7** trans-PdArXL₂ or R in trans-[Pd(CCR)₂L₂] on the course of the reaction. Our main purpose was to report on the data obtained, and we believe that further investigation of the effects of the individual substituents on any of the complexes is beyond the scope of the present study. The decrease in the rate of the transmetallation reaction in the order Cl > Br > I was also observed in other similar Pd-Sn systems, where the halide ligand promotes the cyclic transmetallation pathway and the more electronegative ligand facilitates the process (J. Am. Chem. Soc. **120**, 8978-8985 (1998), Organometallics **25**, 5788-5794 (2006)). This statement and references were added to the manuscript.

We performed the transmetallation reaction with trans-[Pd(CCR)₂L₂] (**6**) and trans-[Pd(CCR)XL₂] (**5**) and compared the rate of formation of tolan product **3** to determine which alkyne reagent was more efficient in transferring the alkyne substrate to the oxidative addition complex **7**, which was indicated in the manuscript (Fig. 7). It was found that trans-[Pd(CCR)₂L₂] **6** was more efficient, likely because it has 2 alkyne handles compared to 1 in trans-[Pd(CCR)XL₂], or because of electronic effects of the halogen ligand in **5**. This was additionally stated in the manuscript.

Comment 9:

A Hammett plot has been constructed in order to analyze the effect of substituents on aryls (Ar) and alkynyls(CCR), but the results are not interpreted. What is the cause of the increase in speed with the higher electron density of the aryls in trans-[PdArXL₂]? Why do aryl alkynyls react faster when bearing EW groups? By the way, it should be stated whether the Hammett plot is built from rate constants or from initial rates.

Reply 9:

We discussed the results of the Hammett plots in the manuscript under Fig. 8. As we indicated in the manuscript, the counterintuitive effects of substituents on phenyl groups on the oxidative addition complex, normally considered electrophilic, where electron donating substituents accelerate the reaction rate, and on the organometallic nucleophiles palladium bisacetylides, where electron withdrawing substituents increase the rates, could be explained by the proposed transmetallation pathway where dissociation of the palladium complexes occurs in the intermediate **12** shown in Fig. 9, which is consistent with the values obtained. The Hammett plots were constructed from the initial rates which was indicated in Supplementary Information and additionally in the manuscript.

Comment 10:

The discussion of the stereochemistry of the transmetallation is completely ignored, as if it were not of the slightest importance.

Reply 10:

The proposed pathway of the transmetallation reaction under study is shown in Figure 9, where the stereochemistry is also briefly discussed. We believe that the detailed stereochemical analysis of the transmetallation process is beyond the scope of the present study.

Comment 11:

Along the text there are several statements that are difficult to understand or to share, such as, for example, that the 6m complex ($\text{trans-[Pd(CC(2-Py))L}_2\text{]}$) gives unusually fast transmetallation reactions due to the coordinative capacity of pyridine. I can't imagine a transmetallation transition state where the pyridine is coordinated to the Pd receiving group while creating the Pd-C bond with the terminal carbon of the alkyne. It would be convenient to include a graph with the TS they propose. The authors cite an intermetallic complex with a short Pd-Pd distance, but in the proposed TS what is relevant is the C-Pd distance.

Reply 11:

As we have indicated, the main purpose of the present manuscript was to demonstrate the generality of the transmetallation process on various substrates. We believe that further exploration of the electronic and other aryl and alkyne substrate-induced effects on the transmetallation step is beyond the scope of the present study. Faster transmetallation rates were observed with pyridine-containing palladium bisacetylide **6m**, $\text{trans-[Pd(CC(2-Py))}_2\text{L}_2\text{]}$, which could be due to the fact that the palladium is brought closer together by the known coordinative ability of pyridine, which can act as a ligand, thus facilitating the reaction (proposed structure I below). Similarly, the alternative transmetallation pathway could operate in the case of a pyridine alkyne substrate (proposed structure II below). However, these are merely hypotheses for one specific example among many others, which don't change the conclusions regarding the generality of the transmetallation reactions studied. Since we do not have concrete evidence for this, we have deleted from the manuscript the part of hypothesis sentence 'for a Pd-Pd interconnected transition state or intermediate'.

Comment 12:

It is also stated that the reaction between complexes **6** and **7** gives rise to "Pd(PPh₃)₂". The observation of this complex as main Pd(0) species is extraordinarily unusual. PdL₂ complexes are relatively stable and can be

isolated with very bulky phosphines (Pcy₃, P(tBu)₃, etc.), but usually Pd(0) complexes with PPh₃ decompose until the amount of phosphine in the medium allows the equilibrium between [Pd(PPh₃)₃] and [Pd(PPh₃)₄]. [Pd(PPh₃)₂] is not usually an observable species. Can the authors give the reference from which they have extracted the chemical shift of this isolated species? Is decomposition to metallic palladium observed in their reactions?

Reply 12:

The reaction of **6** and **7** indeed results Pd(PPh₃)₂ as the product of reductive elimination. The stoichiometry of the transmetallation reaction between **6** and **7** produces palladium monoacetylide **5**,

Pd(PPh₃)₂(X)(CCR) and tolan product **3**, Ar-CCR and Pd(PPh₃)₂ as the products of reductive elimination. In the absence of additional PPh₃, Pd(PPh₃)₂ is observed. The synthesis of Pd(PPh₃)₂ was described in the J. Organometal. Chem. **364**, 235-244 (1989) (Preparation and reactions of bis(triphenylphosphine) palladium(0)), where ³¹P NMR δ 31.0 (s) ppm resonance is reported for this species, which is in good agreement with our observed value ³¹P NMR δ 32.9 (s) ppm, taking into account experimental differences (e.g., solvent). The reference was cited in the manuscript.

Comment 13:

On lines 272 and following it is stated that “4-coordinated palladium oxidative addition complexes, such as **7**, are more likely to undergo dissociation of a PPh₃ ligand and subsequent reaction with the organometallic nucleophile than a direct associative reaction”. I cannot agree with this statement. It is true that complexes like **7** can dissociate phosphine to give important intramolecular reactions, such as beta-hydrogen elimination, or reductive elimination, but intermolecular substitution reactions usually follow an associative mechanism. Another case would be if the authors had used extremely bulky phosphines, such as P(t-Bu)₃ or phosphines derived from biphenyls.

Reply 13:

We and others have previously shown that the dissociative mechanism of complexes **7** is much more likely for transmetallation reactions with organometallic nucleophiles (see Ref. 28 in previous version of the manuscript, Ref. 32 in the revised manuscript and references therein).

To back up our hypothesis, we performed transmetallation with addition of PPh₃, which completely stopped the reaction, indicating that the dissociation of PPh₃ from Pd(PPh₃)₂(Ar)(I) is vital for the successful outcome of the reaction.

Comment 14:

Line 281 states: “Based on the results of Hammett correlations as well as previous DFT calculations, we postulated that dissociation of the Pd-Pd complexes from **12** is most likely the rate-determining step of the process under investigation”. I think that the DFT should support the experimental results, not the other way around, and that the Hammett plots are not evidence of this rds, which is far away from the step that involves RCC or Ar groups. The authors should support this statement with the experimental reaction rate law and with the measure of the activation entropy ΔS[‡].

Reply 14:

We agree with the Reviewer that DFT should support the experimental results and not the other way around. As we indicated in the manuscript, a preliminary DFT study was performed in our previous work to support the experimental transmetallation reaction between **6** and **7** and to postulate the multistep transmetallation pathway (please see Nat. Commun. **9**, 4814 (2018); Ref. 28 in previous version of the manuscript, Ref. 32 in the revised manuscript). As we indicated in the manuscript, in previous work DFT could not distinguish whether the rate-determining step/transition state was the dissociation of palladium complexes (complex **12**, as shown in Figure 9, or the initial dissociation of PPh₃ from **7**). We believe that given the counterintuitive effects of the substituents on the aryl rings in **6** and **7**, the small ρ values in the Hammett plots and the order in **6** and **7**, which are 1 for both reagents, strongly suggest that the dissociation of bimetallic complex **12** may be the rate-determining step of the multistep transmetallation process. This was further supported by calculation of ΔS[‡] (+134 JK⁻¹mol⁻¹

₁, see S172–175). Positive value of ΔS^\ddagger suggests dissociative mechanism in transition state, which along with order in both reactants points towards **12** being the key transition state of the reaction under study. The result was included in the manuscript.

Comment 15:

From an experimental point of view, the measurement of the rates should be done in the presence of excess phosphine. This is quite obvious from the proposed reaction scheme in Figure 9. Due to the ability of the intermediate $[\text{Pd}(\text{PPh}_3)_2]$ to capture PPh_3 , the dissociation equilibrium of the first step (or the substitution equilibrium for the formation of the intermediate **9** by any mechanism) is progressively shifted to the right as the reaction progresses, meaning that the rate at each instant does not depend exclusively on the concentration of the reactants.

Reply 15:

We performed the additional experiment with excess PPh_3 (4 equiv.) for the reaction between **6a** and **7a**, which completely retarded the reaction. This is consistent with our hypothesis that the reaction proceeds via a dissociative mechanism of PPh_3 from **7**, as shown in Figure 9, with excess PPh_3 preventing dissociation of PPh_3 from **7**. Similar experiments and conclusions have been performed and observed by others for Pd/M systems, e.g., Suzuki coupling, catalytic Pd/B systems (*J. Org. Chem.* **59**, 5034-5037 (1994)) and in stoichiometric Pd/Cu systems, in which alkynyl ligand transfer between an alkynyl copper and $\text{trans-}[\text{Pd}(\text{II})(\text{Ar})(\text{PEt}_3)_2]$ species slows upon addition of excess PPh_3 (*Organometallics* **16**, 5354-5364 (1997)). Experimental studies showed the dissociation of phosphine from square-planar Pd(II) species (*J. Am. Chem. Soc.* **109**, 148-156 (1987)) and of stable three-coordinated Pd(II) species with only one bulky phosphine group (*J. Am. Chem. Soc.* **126**, 1184-1194 (2004); *J. Am. Chem. Soc.* **124**, 9346-9347 (2002)).

Comment 16:

Finally, the authors compare the rate of the catalytic cycle with that of isolated reactions. They find that the rate under catalytic conditions is 19 times higher than that obtained in reductive transmetallation/elimination experiments. The authors do not satisfactorily justify this fact that questions their hypothesis, limiting themselves to saying that the conditions are not the same. In summary, the article presents a series of interesting experimental results, but its analysis and discussion do not meet the requirements of a journal such as *Communications Chemistry*, so I cannot recommend it to be accepted.

Reply 16:

We thank the Reviewer for this comment, which we also consider one of the key weaknesses of the manuscript (the same question was raised by the Reviewer 2, so the comment is the same as the Reply 6 to the Reviewer 2).

The large discrepancy between the catalytic reaction and the stoichiometric transmetallation arose for the reason that **7** is actually a precursor of the catalytically active species, i.e., the tricoordinated palladium oxidation addition complex (which we have in fact indicated and noted in the manuscript 'Catalytically active species are usually not stable enough to be isolated, so their precursors are usually isolated. For example, Pd oxidative addition complexes **7**, $\text{trans-Pd}(\text{PPh}_3)_2(\text{Ar})(\text{X})$ are isolated as precursors to catalytically active tricoordinated $\text{Pd}(\text{PPh}_3)(\text{Ar})(\text{X})$ species. Precursors to catalytically

active species are usually observed in reactions as the resting states of catalytic cycles' and drawn in Figure 9). Since the triphenylphosphine forms isolable palladium oxidative addition complexes in tetracoordinate form, we prepared the tricoordinated palladium oxidative addition complex Pd(P(*o*-tolyl)₃)(Ar)(I) **7a'** as a surrogate for tricoordinated Pd(PPh₃)(Ar)(I) species, which we reacted with **6a**. The reaction of tricoordinated palladium oxidative addition complex Pd(P(*o*-tolyl)₃)(Ar)(I) **7a'** proved to be considerably faster and nearly as fast as studied catalytic reaction despite the observed side reactions that can be attributed to relative instability of tricoordinated palladium complex **7a'**. For detailed information see Supp. Info. (S158–160). However, because of the relative instability of prepared tricoordinate **7a'**, tetracoordinated **7** are still more suitable for the investigation of the elementary reactions under study.

This indicates that the additional phosphine ligand, from tetracoordinated **7**, present in the stoichiometric transmetallation reaction was the reason for the discrepancy in rates between the catalytic reaction and the stoichiometric transmetallation, which we also showed in the experiment when we added 4 equiv. to the transmetallation reaction between **6a** and **7a**, completely retarding the reaction.

We have included these observations in the manuscript and Supplementary Information.

We thank the Reviewer for their comments and issues raised, which we believe have not altered the conclusions of the manuscript, and we hope that now that these issues have been resolved, the manuscript is suitable for publication.

Reviewers' comments:

Reviewer #2 (Remarks to the Author):

I thank the authors for their responses to my comments and the addition of extra experiments. I'd like to reiterate that much of the content of the manuscript is of interest to the general community and describes well the behaviour of these complexes under stoichiometric conditions. I do still have a different opinion however on the interpretation with regards to the rate determining step which I describe below.

I return to my point on the order in palladium under catalytic conditions. If there is a rate determining bimolecular reaction between two palladium species then the reaction will exhibit pseudo-zero order kinetics, but a second order dependence on palladium as shown below:

Rate = $k[\text{Pd1}][\text{Pd2}]$, where $\text{Pd1} = 6$ $\text{Pd2} = 7$ (therefore zero order with respect to substrates, pseudo-zero order kinetic profile, assuming no deactivation of catalyst)

$$[\text{Pdtotal}] = [\text{Pd1}] + [\text{Pd2}]$$
$$\text{So } 2[\text{Pdtotal}] = 2([\text{Pd1}] + [\text{Pd2}])$$

Even if 99% of the catalyst sits as Pd2, doubling of the total pre-catalyst will increase the rate by 4x as both the concentration of Pd1 and Pd2 will be doubled. Just because one species is in large excess does not mean the other does not contribute to the rate equation.

e.g. from 0.01 M of precatalyst:

$$\text{Pd1} = 0.0001 \text{ M} \quad \text{Pd2} = 0.0099 \text{ M}$$

$$\text{Rate}(\text{exp1}) = k[\text{Pd1}][\text{Pd2}] = k*(0.0001*0.099) = 9.9\text{E-}7*k$$

Double catalyst: 0.02 M

$$\text{Pd1} = 0.0002 \text{ M} \quad \text{Pd2} = 0.0198 \text{ M}$$

$$\text{Rate}(\text{exp2}) = k[\text{Pd1}][\text{Pd2}] = k*(0.0002*0.0198) = 3.96\text{E-}6*k$$

$$\text{Therefore } (\text{rate}(\text{exp2})/\text{rate}(\text{exp1})) = 4$$

The other mechanistic scenarios are as follows with expected orders in Pd:

An additive limiting the concentration of 6 or 7: This goes back to my question on how exactly does the precatalyst transform to both Pd(0) for oxidative addition and Pd(II) for bisacetylide formation, and I thank the authors for their explanation. If for whatever reason doubling the loading of precatalyst does not double the concentration of both of these species, then a first order behaviour could be explained. One scenario example could be there is a minor impurity "X" where $X < \text{Pd}(\text{total})$ and the formation of 6 or 7 depends on X e.g. $\text{Precat} + X$ goes to 7. Then as you increase the total Pd content, you only increase 6, because the amount of 7 produced depends on X.

RDS from 12: If $6 + 7$ goes to 12 irreversibly then 12 will accumulate. This is not observed and therefore it could be an equilibrium. i.e. $6 + 7 = 12$ which then goes on to products. Under this scenario where the equilibrium lies on the left-hand side (and therefore 12 is not observed), it will still be second order in palladium as doubling the concentration of total catalyst will increase the concentration of 12 by ca. 4x.

RDS from B (reductive elimination): It is possible that reductive elimination is rate limiting but B is not observed due to a pre-equilibrium as well. i.e. $6 + 7 = 5 + B$, with the equilibrium on the LHS such

that only 6 or 7 are observed. In this situation a doubling of the total catalyst will only double the concentration of B, and therefore the reaction will be first order in palladium. This is the only scenario I can think of where 6 or 7 are the only observable species, the reaction kinetics are pseudo-zero order and overall 1st order in palladium, but I am happy to be corrected.

The authors previously discounted RE as rate limiting as the associated species B wasn't observed, but a pre-equilibrium could explain this.

An in-depth kinetic study of the catalytic reaction could be considered to look at this in more detail (for example, if the Hammett analysis under catalytic conditions gave the same results as stoichiometrically), confirming the order in all other reagents etc. I would also recommend derivation of the rate law so that there is no ambiguity here on what is controlling the reaction rate.

Before publication, I recommend reassessing some of the statements with regards to the rate-determining step in the context of their previous work and my comments above. I think substantially more work would be needed for a comprehensive kinetic study on the catalytic reaction and do not necessarily recommend that that is included here as the authors have now acknowledged in the manuscript that other mechanisms could compete under different conditions.

Reviewer #3 (Remarks to the Author):

Dear editor

I appreciate that this is a quite improved version of the manuscript.

In the author's own words: "the key purpose of the study was to determine the mechanistic pathway of the reaction". The work has been very much improved with new references that had been requested by reviewers and some new experiments, but has some drawbacks: The kinetic experiments are based on initial rates, in a different solvent than the one used in catalysis, in the absence of base, and in the absence of excess ligands. These two aspects are very important because they can significantly alter speciation.

The proposed reaction mechanism is perfectly acceptable in its general aspects. However, when getting down to the details of each step, some of the proposals are not solidly grounded in experiments or require more discussion.

The most important aspects of the mechanism that remain to be clarified are: the stereochemistry of the transmetalation (in *Nat. Commun.* 9, 4814 (2018) the presence of trans-[PdAr(alkynyl)L₂] is observed, and obviously the cis isomer, which gives the reductive elimination), and the reaction order of the transmetalation step with respect to palladium, which surprisingly, in a preliminary study, comes out of order one. The authors include now the experimental determination of the reaction order on each reagent in the transmetalation step, being one on each of them contrary to the observed kinetics in the previous article in *Nat. Commun.*

Some aspects should be improved:

The dependence of the rate of transmetalation on the halogen, being Cl > Br > OAc > I, is not sufficiently explained. This is important because the authors make clever use of this effect to improve the reaction, and also because it can give clues about the actual nature of the species involved in the transmetalation under catalytic conditions. Thus on page 9, the next sentence has been included: "The decrease in the rate of the transmetalation reaction in the order Cl > Br > I was also observed in other similar Pd-Sn systems, where the halide ligand promotes the cyclic transmetalation pathway and the more electronegative ligand facilitates the process. (References: *J. Am. Chem. Soc.* 120, 8978-8985 (1998), *Organometallics* 25, 5788-5794 (2006)). In these references, the transmetalation between tin and palladium was studied and on it, the activation energy for different halogens X depends on the bond dissociation energy between Pd-X or Sn-X which are very different for chlorine

bromine and iodide (Sn-Cl is much favored relative to Pd -Cl than Sn-I to Pd-I). However in a Palladium by Palladium transmetalation the bonds that are broken and created in the transition state (Pd-X and Pd-C) are of the same nature as the bond that is created and broken (Pd-C and Pd-X), so that it is not clear why the order Cl > Br > OAc > I and the comparison with Pd/Sn transmetalation are of little value.

There are some other issues, such as the speciation of Pd(0) species that is proposed, which ignores the presence of ligands (alkynes) in the solution that could stabilize the proposed [Pd(PPh₃)₂] complexes, and the absence of discussion of the trans influence of ligands that could explain the relative rates of transmetalation when [Pd(alkynyl)₂(PPh₃)₂] and [Pd(alkynyl)X(PPh₃)₂] are used as transmetalation reagent.

Response to Reviewers' comments regarding the manuscript entitled 'Elucidating reaction mechanism of palladium-palladium dual catalytic process through kinetic studies of proposed elementary steps' in Communications Chemistry

Reviewer #2 (Remarks to the Author):

Comment 1:

I thank the authors for their responses to my comments and the addition of extra experiments. I'd like to reiterate that much of the content of the manuscript is of interest to the general community and describes well the behaviour of these complexes under stoichiometric conditions. I do still have a different opinion however on the interpretation with regards to the rate determining step which I describe below.

Reply 1:

We thank the Reviewer for their positive comments and issues raised that further improved the manuscript. In the following, we address the issues raised in connection with the interpretation of the rate-determining step of the catalytic reaction. To this end, we performed additional experiments that we have included in the revised version of the manuscript and Supplementary Information. The additional experiments further connected the analysis under stoichiometric conditions with catalytic conditions.

Comment 2:

I return to my point on the order in palladium under catalytic conditions. If there is a rate determining bimolecular reaction between two palladium species then the reaction will exhibit pseudo-zero order kinetics, but a second order dependence on palladium as shown below:

Rate = $k[\text{Pd1}][\text{Pd2}]$, where $\text{Pd1} = 6$ $\text{Pd2} = 7$ (therefore zero order with respect to substrates, pseudo-zero order kinetic profile, assuming no deactivation of catalyst)

$$[\text{Pd}_{\text{total}}] = [\text{Pd1}] + [\text{Pd2}]$$

$$\text{So } 2[\text{Pd}_{\text{total}}] = 2([\text{Pd1}] + [\text{Pd2}])$$

Even if 99% of the catalyst sits as Pd2, doubling of the total pre-catalyst will increase the rate by 4x as both the concentration of Pd1 and Pd2 will be doubled. Just because one species is in large excess does not mean the other does not contribute to the rate equation. e.g. from 0.01 M of precatalyst:

$$\text{Pd1} = 0.0001 \text{ M} \quad \text{Pd2} = 0.0099 \text{ M}$$

$$\text{Rate}(\text{exp1}) = k[\text{Pd1}][\text{Pd2}] = k(0.0001 \cdot 0.0099) = 9.9\text{E-}7 \cdot k$$

Double catalyst: 0.02 M

$$\text{Pd1} = 0.0002 \text{ M} \quad \text{Pd2} = 0.0198 \text{ M}$$

$$\text{Rate}(\text{exp2}) = k[\text{Pd1}][\text{Pd2}] = k(0.0002 \cdot 0.0198) = 3.96\text{E-}6 \cdot k$$

$$\text{Therefore } (\text{rate}(\text{exp2})/\text{rate}(\text{exp1})) = 4$$

Reply 2:

We agree with the Reviewer's rationale and calculation provided. In our previous Response to Reviewers' Comments document we responded to the order in Pd for the catalytic reaction under reaction conditions given in our previous publication (Nat Comm 2018, 4, 4818), i.e. Pd(PPh₃)₄, NaOMe, DMF), which turned out to be 1, which was addressed by the Reviewer, and for which we stated in our previous article that the order 1 in Pd in the studied catalytic reaction "one could interpret this result by either reductive elimination from Cycle A or palladium bis-acetylide **6** formation from Cycle B as the potential rate-limiting steps". We have also indicated that transmetallation could still be the rate-limiting step due to "in case of prompt oxidative addition, and consistent with the results shown in Supplementary Fig. 2, then the rate of transmetallation virtually depends on the concentration of **6**. In this case, transmetallation can be well approximated as a pseudo-1st order kinetics and hence it should not be excluded from the list of possible rate-determining steps based on our preliminary kinetic study." It should be noted that transmetallation need not be the rate-determining step of the catalytic process; other steps could also be rate-determining, which would not preclude transmetallation from operating in the reaction. We have added this note in the revised version of the manuscript.

However, the reaction studied in the present manuscript describes a different catalytic system, i.e. Pd(PPh₃)₂I₂ in DCM, with an organic base (pyrrolidine), which allows easier and more accurate monitoring of the reaction. This is because the solubility of the organic base (instead of inorganic NaOMe) makes the reaction mixture homogeneous and also avoids the potential acceleration effects of NaOMe in the catalytic reaction, which were also raised by Reviewer #3 (please see Reply 2 to Reviewer #3 below in the Comments from Reviewer #3 regarding the points raised by Reviewer #1.).

To address the order in Pd in the catalytic system studied in the present manuscript, we performed a catalytic reaction between 4-tolyl iodide **1a** (0.5 M) and phenylacetylene **2a** (0.55 M) in the presence of pyrrolidine (1.0 M, 2 equiv.) as base in dichloromethane at room temperature and varied the concentration of palladium precatalyst Pd(PPh₃)₂I₂ (0.01 M, 0.02 M, 0.03 M, 0.04 M, and 0.06 M; i.e., 2 mol%, 4 mol%, 6 mol%, 8 mol%, and 12 mol%, respectively).

It is known that coordinating amine bases, such as pyrrolidine, can replace one of the phosphine ligands in palladium oxidative addition complex Pd(PPh₃)₂(Ar)(X) **7**, resulting in an equilibrium between Pd(PPh₃)₂(Ar)(X) and Pd(PPh₃)(pyrrolidine)(Ar)(X) (J. Org. Chem. **71**, 1677-1687 (2006); Chem. Eur. J. **13**, 666-676 (2007); Nat. Comm. **9**, 4814 (2018), Refs. 20, 32, 82 in the manuscript). This affects the speciation of Pd species in the reaction mixture under catalytic conditions (see Figure 2 in Nat. Comm. **9**, 4814 (2018)).

For the equilibrium (formation) between Pd(PPh₃)₂(Ar)(X) **7** and Pd(PPh₃)(pyrrolidine)(Ar)(X) please, see Supplementary Fig. 13 of Supplementary Information in Nat. Comm. **9**, 4814 (2018), and Supplementary Fig. 27 in Supplementary Information of the present manuscript.

On the other hand, pyrrolidine does not interact with the palladium bisacetylide Pd(PPh₃)₂(CCR)₂ **6** (see Supplementary Fig. 33; in the ³¹P NMR spectra of the solution of **6** in the presence of 10 equivalents of pyrrolidine, no additional resonances are seen, but after 1 hour O=PPh₃ begins to form), and have little effect on **5** (see Supplementary Fig. 14 in Supplementary Information in Nat. Comm. **9**, 4814 (2018)). Description of these effects was included in the revised version of the manuscript and in the Supplementary Information (see pages S169,170).

We hypothesized that the formation of Pd(PPh₃)(pyrrolidine)(Ar)(X) could potentially influence the course of the catalytic reaction by interacting with **7**, which could affect the transmetallation step between **6** and **7**. The order 1.6 in Pd in the catalytic reaction studied, a partial order higher than 1 and close to 2, prompted us to study the stoichiometric transmetallation reaction in the presence of pyrrolidine. To determine the order in **6a** and **7a** in the stoichiometric transmetallation reaction in the presence of pyrrolidine, we performed reactions of **6a** and **7a** in the presence of 20 equivalents of pyrrolidine (the excess/concentration of pyrrolidine that still allows monitoring of the reaction by NMR) in CDCl₃ at 302 K. We performed the experiments by varying the concentrations of one of the reagents **7a** and **6a** at 4 different concentrations (0.005 M, 0.010 M, 0.015 M, 0.025 M) in the presence of 20 equivalents (0.1 M) of pyrrolidine (see Supplementary Information S189–S194). For **6a**, the observed order turned out to be 1.2, and for **7a**, 0.5. The order in **6a** was comparable to the transmetallation reaction in the absence of pyrrolidine, where it turned out to be 1.1. On the other hand, there was a decrease in the order in **7a**, which dropped from 0.9 in the case of the transmetallation reaction in the absence of pyrrolidine to 0.5 in the presence of pyrrolidine.

Thus, the combined order of **6a** and **7a** in the stoichiometric transmetallation reaction is 1.7, which is comparable to the order of palladium in the catalytic reaction studied (1.6), strongly suggesting that the transmetallation reaction is the rate-determining step of the catalytic reaction studied.

We have included these observations in the revised version of the manuscript and Supplementary Information.

Comment 3:

The other mechanistic scenarios are as follows with expected orders in Pd: An additive limiting the concentration of **6** or **7**: This goes back to my question on how exactly does the precatalyst transform to both Pd(0) for oxidative addition and Pd(II) for bisacetylide formation, and I thank the authors for their explanation. If for whatever reason doubling the loading of precatalyst does not double the concentration of both of these species, then a first order behaviour could be explained. One scenario example could be there is a minor impurity "X" where X < Pd(total) and the formation of **6** or **7** depends on X e.g. Precat + X goes to **7**. Then as you increase the total Pd content, you only increase **6**, because the amount of **7** produced depends on X.

Reply 3:

As we stated in our earlier Response to Reviewers' Comments and in our earlier paper (Nat. Commun. **9**, 4814 (2018), ref. 32 in the revised manuscript), the formation of the catalytic amounts of either Pd(II) and Pd(0), each of which is required for one of the catalytic cycles, results from side reactions, i.e. when Pd(PPh₃)₂X₂ is used as a precatalyst, as in the case of the catalytic reaction described in the present manuscript, the Pd(0) is formed after the formation of palladium bisacetylide and reductive elimination of the Hay-Glaser alkyne dimer, which initiates the formation of the oxidative addition complex and catalysis (Bull. Chem. Soc. Jpn., **77**, 2033-2045 (2004)). In contrast, when Pd(PPh₃)₄ is used as a precatalyst, the formation of Pd(PPh₃)₂(Ar)(X) and further transmetallation between the two species leads to Pd(PPh₃)₂(X)₂, a precursor for the formation of palladium bisacetylide, as shown in our previous work (ref. 32 in the manuscript). In these two side reactions, the required Pd precursors are formed in small amounts, and in this way the catalytic reaction is initiated. This is observed as an induction period at the beginning of the catalytic reactions, described in our previous work, in the present manuscript and was as well observed by others (see Nat. Commun. **9**, 4814 (2018); and references cited therein). After the induction period, an equilibrium is formed between the key Pd species, which depends on the reaction conditions, and leads to the reaction proceeding at maximum rate.

Comment 4:

RDS from 12: If 6 + 7 goes to 12 irreversibly then 12 will accumulate. This is not observed and therefore it could be an equilibrium. i.e. 6 + 7 = 12 which then goes on to products. Under this scenario where the equilibrium lies on the left-hand side (and therefore 12 is not observed), it will still be second order in palladium as doubling the concentration of total catalyst will increase the concentration of 12 by ca. 4x.

Reply 4:

We thank the Reviewer for pointing out the equilibrium between 6 and 7 to form 12. As is usual with mechanisms, all intermediate states of the multistep TM process are (most likely) in equilibrium, which we have now additionally indicated by equilibrium arrows in Fig. 9 in the revised version of the manuscript, except for the last step (RE) in which the tolane product 3 is formed.

We performed an additional experiment in which we increased the concentration of 6a and 7a fourfold (to 0.04 M) to potentially observe the proposed intermediate 12. However, the experiment yielded only the previously observed 5a, Pd(PPh₃)₂, as a product of the transmetallation reaction, together with by-products, O=PPh₃, along with Pd(PPh₃)₂l₂, Pd(PPh₃)₂(tol)₂ that began to form after 1 h after the start of the reaction. We have included this observation in the Supplementary Information (see Supplementary Fig. 16). Even if 12 is indeed the intermediate before the rate-determining step, the lack of observation of 12 is not unexpected in our opinion, since 12 could be considered a transition state rather than an intermediate and the former are not normally observed as stable species. As indicated in the previous response, the experiments shown in Fig. 9 for the transmetallation of 6a and 7a in the absence of an additive, i.e., the order in 6a and 7a being 1 for both reagents, as well as the positive ΔS and Hammett plot results, suggest that the rate-determining step of the multistep transmetallation process could be the dissociation of palladium complexes from the intermediate 12.

Comment 5:

RDS from B (reductive elimination): It is possible that reductive elimination is rate limiting but B is not observed due to a pre-equilibrium as well. i.e. $6 + 7 = 5 + B$, with the equilibrium on the LHS such that only 6 or 7 are observed. In this situation a doubling of the total catalyst will only double the concentration of B, and therefore the reaction will be first order in palladium. This is the only scenario I can think of where 6 or 7 are the only observable species, the reaction kinetics are pseudo-zero order and overall 1st order in palladium, but I am happy to be corrected.

Reply 5:

We thank the Reviewer for this comment and acknowledge that it is possible that reductive elimination is the rate-determining step of the transmetallation process studied, since in the case of transmetallation reactions between palladium oxidative addition complexes **7** and palladium bisacetylides **6** into **5**, Pd(PPh₃)₂ and toluene product **3**, we are actually observing two elementary steps, transmetallation and reductive elimination. However, analysis of the order in **6a** and **7a** in the stoichiometric transmetallation reaction in the absence of the additives revealed that the order is 1 for both **6a** and **7a**, whereas in the presence of pyrrolidine it is 1.2 for **6a** and 0.5 for **7a**. Moreover, the order in palladium in the catalytic reaction is 1.6, suggesting that one of the steps within the multistep transmetallation process is the rate-determining step in the reaction studied. However, it could also be that both **6a** and **7a** contribute to the formation of the key intermediate **B** for reductive elimination, as suggested by the Reviewer, with the equilibrium being on the LHS. We have added a note in the revised version of the manuscript that reductive elimination may also be the rate-determining step of the process under study.

Comment 6:

The authors previously discounted RE as rate limiting as the associated species B wasn't observed, but a pre-equilibrium could explain this.

Reply 6:

We agree with the Reviewer, please see Reply 5.

Comment 7:

An in-depth kinetic study of the catalytic reaction could be considered to look at this in more detail (for example, if the Hammett analysis under catalytic conditions gave the same results as stoichiometrically), confirming the order in all other reagents etc. I would also recommend derivation of the rate law so that there is no ambiguity here on what is controlling the reaction rate.

Reply 7:

As indicated in Reply 2, we performed the analysis of the catalytic reaction studied, which showed that the order in Pd is 1.6, which is in good agreement with the combined order in **6a** and **7a**, the proposed key intermediates of the transmetallation step, which is 1.7 under similar reaction conditions in the stoichiometric transmetallation reaction.

As we indicated in the manuscript, changing the substrates, i.e., using other aryl halides, e.g., instead of aryl iodide, aryl bromide, or aryl chloride, would completely change the rates of the elementary steps of the catalytic reaction, in which case, for example, oxidative addition would most likely become the rate-determining step of the process. Similarly, the effects of substituents on the phenyl ring of the aryl iodide as well as on the alkyne handle of the terminal alkyne reagent could (and probably would) also completely change the rates of the elementary steps of the process. Previously, competitive Hammett studies for a similar reaction, i.e., the palladium-catalyzed cross-coupling of aryl halides with terminal alkynes, found that the rate-determining step of the catalytic process was altered by the use of different substituents on the phenyl ring of the alkyne reagent (*Organometallics* **27**, 24902498 (2008)). Similarly, in our previous study (*Nat. Comm* **9**, 4814 (2018)), we observed that the order in Pd was 1 in the palladium-catalyzed reaction between aryl iodide and phenylacetylene (when NaOMe was chosen as base and DMF as solvent), suggesting that one of the steps involving one palladium (oxidative addition, palladium bisacetylide formation, or reductive elimination) is most likely the rate-determining step of the process. However, this does not mean that transmetallation is not involved in the catalytic process, but that under certain reaction conditions transmetallation proceeds faster than one of the above steps. A similar observation was made when studying the catalytic reaction of the original Pd/Cu-catalyzed Sonogashira coupling, where oxidative addition was found to be the rate-determining step of the process rather than Pd-Cu transmetallation, as one would expect (*Can. J. Chem.* **86**, 410-415 (2008)).

In the revised version of the manuscript, we pointed out that although transmetallation is not the rate-determining step of the catalytic process, this does not mean that it is not operating in the reaction.

We believe that describing all the effects of the substituents on both substrates is beyond the scope of the present study. Herein, we have described the approach to analyzing the reaction mechanism of a bimetallic catalytic process, i.e., a dual palladium-palladium catalytic process, based on disassembly of the reaction mechanism into elementary steps and studying them independently. We believe we have shown that Pd-Pd transmetallation is general to the complexes studied and that it proceeds in the catalytic reaction studied, where it is also shown to be the rate-determining step of the process.

Comment 8:

Before publication, I recommend reassessing some of the statements with regards to the rate-determining step in the context of their previous work and my comments above. I think substantially more work would be needed for a comprehensive kinetic study on the catalytic reaction and do not necessarily recommend that that is included here as the authors have now acknowledged in the manuscript that other mechanisms could compete under different conditions.

Reply 8:

As suggested by the Reviewer, we additionally studied the catalytic reaction under investigation, which further connected the analysis of reactions under stoichiometric conditions with those under catalytic conditions. We have included the results in the revised version of the manuscript and in the Supplementary Information. We also reassessed some statements in the revised version of the manuscript related to the rate-determining step as described above. We hope that the manuscript is now more suitable for publication.

Reviewer #3 (Remarks to the Author):

Comment 1:

Dear editor I appreciate that this is a quite improved version of the manuscript. In the author's own words: "the key purpose of the study was to determine the mechanistic pathway of the reaction". The work has been very much improved with new references that had been requested by reviewers and some new experiments, but has some drawbacks: The kinetic experiments are based on initial rates, in a different solvent than the one used in catalysis, in the absence of base, and in the absence of excess ligands. These two aspects are very important because they can significantly alter speciation.

Reply 1:

We thank the Reviewer for their positive comments and appreciate the comments and issues that helped improve the manuscript.

As we indicated in the previous Response to Reviewers' Comments document we have shown that both the catalytic and stoichiometric reactions proceed at very similar rates in dichloromethane and chloroform. We performed the catalytic reaction between **1a** and **2a** in both dichloromethane and chloroform, where comparable rates were observed (please see Supplementary Information S66–69), and the same was true for the stoichiometric transmetallation reactions between **6a** and **7a**, which were performed in CDCl₃ and CD₂Cl₂ (please see S70–74). Thus, the solvents CH₂Cl₂ and CHCl₃ have no significant effect on the reactions studied. We have additionally emphasized this in the revised version of the manuscript.

We have previously described that the excess ligand PPh₃ retards the transmetallation reaction, suggesting a dissociative mechanism, and that the tricoordinated palladium oxidative addition complex **7'** is a reactive species involved in the transmetallation step.

We thank the Reviewer for pointing out the speciation under different reaction conditions, as this aspect was not well described in the previous version of the manuscript. It is true that the reaction conditions used in the catalytic reaction studied, the use of pyrrolidine as a base that can also act as a ligand, can change the speciation of the Pd complexes in the reaction mixture, which we discussed in our previous publication (Nat. Commun. **9**, 4814 (2018), ref. 32 in the manuscript). In the revised version of the manuscript, we additionally describe the effect of pyrrolidine on the studied palladium key species, i.e., **5**, **6**, **7**, as follows. It is known that coordinating amine bases, such as pyrrolidine, can replace one of the phosphines in **7**, leading to an equilibrium between **7** and **7'(pyr)**, Pd(PPh₃)(pyrrolidine)(Ar)(X) (Chem. Eur. J. **13**, 666–676 (2007), Nat. Commun. **9**, 4814 (2018), J. Org. Chem. **71**, 1677–1687 (2006); refs. 20, 32, 85 in manuscript), with $K = 0.15$ (CDCl₃, 300 K), $K = \frac{[\text{Pd}(\text{PPh}_3)(\text{pyrrolidine})(\text{Ar})(\text{X})]_x[\text{PPh}_3]}{[\text{Pd}(\text{PPh}_3)_2(\text{Ar})(\text{X})]_x[\text{pyrrolidine}]}$, please see Supplementary Information pages S24,25 in Nat. Comm. **9**, 4818 (2018)). On the other hand, pyrrolidine has little effect on structure **5** ($K = 0.03$ (CDCl₃, 300 K), $K = \frac{[\text{Pd}(\text{PPh}_3)(\text{pyrrolidine})(\text{CCAr})(\text{X})]_x[\text{PPh}_3]}{[\text{Pd}(\text{PPh}_3)_2(\text{CCAr})(\text{X})]_x[\text{pyrrolidine}]}$, see Supplementary Information pages S26,27 in Nat. Comm. **9**, 4818 (2018)), and little effect on **6**, (see Supplementary

Figure 33). We have included these notions in the revised version of the manuscript and Supplementary Information.

In addition, we performed oxidative addition of Pd(PPh₃)₄ and **1a** in the presence of pyrrolidine, which proved instantaneous, as in the absence of pyrrolidine (Supplementary Figs. 26 and 27). We have included this observation in the revised version of the manuscript. The initial formation of palladium bisacetylide **6a** from Pd(PPh₃)₂I₂ and the regeneration of palladium bisacetylide **6a** from **5a** were carried out in the presence of pyrrolidine. The pyrrolidine also affected the order in **6a** and **7a** in the transmetalation step, as we have described in detail above (please see Reply 2 to Reviewer 2), where order in **6a** remained unchanged but in the case of **7a** dropped from 0.9 in the absence to 0.5 in the presence of pyrrolidine. The absolute reaction rates of **7a** and **6a** slowed down in the presence of pyrrolidine, which can be considered as an additional coordinating ligand, similar to the case of addition of excess PPh₃ to the reaction of **7a** and **6a**, indicating a dissociative reaction pathway of transmetalation and that the tricoordinated palladium oxidation addition complex **7'** is the reactive species in the transmetalation step, as we have indicated above and in the manuscript.

The results where we considered speciation as a consequence of pyrrolidine as a coordinating base in the reaction mixtures did not change the conclusions on the rates of the elementary steps studied.

Comment 2:

The proposed reaction mechanism is perfectly acceptable in its general aspects. However, when getting down to the details of each step, some of the proposals are not solidly grounded in experiments or require more discussion.

The most important aspects of the mechanism that remain to be clarified are: the stereochemistry of the transmetalation (in Nat. Commun. 9, 4814 (2018) the presence of trans-[PdAr(alkynyl)L₂] is observed, and obviously the cis isomer, which gives the reductive elimination), and the reaction order of the transmetalation step with respect to palladium, which surprisingly, in a preliminary study, comes out of order one. The authors include now the experimental determination of the reaction order on each reagent in the transmetalation step, being one on each of them contrary to the observed kinetics in the previous article in Nat. Commun.

Reply 2:

The stereochemistry of the transmetalation process described in the manuscript is tentatively proposed based on experimental data observed during the study of the transmetalation reaction of **6** and **7**. As described in the manuscript, the proposed transmetalation most likely proceeds via a dissociative pathway from **7** to **7-PPh₃**, which then reacts with palladium bisacetylide **6**. In the revised version of the manuscript, we have included additional discussion of the stereochemistry of the proposed transmetalation with the emphasis on the trans effect of ligands in Pd complexes.

As mentioned earlier, the order 1 in Pd in the catalytic reaction in our previous work (Nat. Comm. 9, 4814 (2018)), may suggest that one of the other steps involving one palladium, e.g. oxidative addition, palladium bisacetylide formation, or reductive elimination, could be the rate-determining steps of the catalytic reaction of **1a** and **2a** in the presence of Pd(PPh₃)₄, NaOMe, DMF (see Reply 2 to Reviewer 2 for more details), but this does not mean that transmetalation does not operate in the catalytic

process. For example, as mentioned above, in the Pd-Cu Sonogashira reaction, oxidative addition was found to be the rate-determining step of the process (Can. J. Chem. **86**, 410-415 (2008)), rather than Pd-Cu transmetallation as one would expect, but this does not preclude Pd-Cu from operating in the reaction.

In the present study, the order of the palladium key species involved in transmetallation step **6a** and **7a** shown in Figure 9 was determined in the absence of additives and resulted in order 1 in **6a** and **7a**, respectively. All other kinetic parameters shown in Figure 9 are the result of studies of the transmetallation reaction of **6a** and **7a** in the absence of additives, as are the other transmetallation studies between **6** and **7** in Figures 6, 7, and 8. Since this is the first systematic study of such a Pd-Pd transmetallation reaction, our first objective was to investigate the generality of the Pd-Pd transmetallation reaction and the effect of the ligands present in **6** and **7**, i.e., halide ligands in **7**, substituents on alkyne handles in **6**, and on aryl rings in **7**, on the progress of the reaction. This was done to simplify the analysis and was additionally emphasized in the manuscript.

However, when comparing the rates of the proposed elementary steps of the catalytic cycles under stoichiometric conditions with the catalytic reaction studied, we considered the possible effect of the additive, i.e., the coordinating base pyrrolidine in the present study, on the rates of the elementary steps. As described above, the order in Pd in the catalytic reaction studied was 1.6 and the combined order of **6a** (0.5) and **7a** (1.2) in the stoichiometric transmetallation reaction in the presence of pyrrolidine was 1.7, indicating that the transmetallation step is the rate-determining step of the catalytic reaction studied (and that it indeed proceeds in the reaction studied).

We discussed in the previous Response to Reviewers' Comments document that in our previous study (Nat Comm **9**, 4814, (2018)) we observed *trans*-[PdAr(alkynyl)L₂] under the reaction Conditions A: Pd(PPh₃)₄, NaOMe, DMF, 50 °C. The observation of *trans*-[PdAr(alkynyl)L₂] is most likely the result of specific reaction conditions, solvent, and base that could initiate isomerization, or it could also be the result of the different stereochemical pathway of the transmetallation process in the presence of NaOMe, as suggested by the Reviewer. We did not observe the *cis*-isomer **B** in any of the studies, which is not surprising since such a structure tends to reductively eliminate the tolane product **3** and would show characteristic resonances in the ³¹P NMR spectrum due to two nonequivalent phosphorus atoms. However, the reaction conditions described in the present manuscript do not include NaOMe that could lead to the above effects, and we believe that describing the effects of NaOMe on the transmetallation step is beyond the scope of the present study.

Comment 3:

Some aspects should be improved: The dependence of the rate of transmetalation on the halogen, being Cl > Br > OAc > I, is not sufficiently explained. This is important because the authors make clever use of this effect to improve the reaction, and also because it can give clues about the actual nature of the species involved in the transmetallation under catalytic conditions. Thus on page 9, the next sentence has been included: "The decrease in the rate of the transmetalation reaction in the order Cl > Br > I was also observed in other similar Pd-Sn systems, where the halide ligand promotes the cyclic transmetalation pathway and the more electronegative ligand facilitates the process. (References: J. Am. Chem. Soc. **120**, 8978-8985 (1998), Organometallics **25**, 5788-5794 (2006)). In these references,

the transmetalation between tin and palladium was studied and on it, the activation energy for different halogens X depends on the bond dissociation energy between Pd-X or Sn-X which are very different for chlorine bromine and iodide (Sn-Cl is much favored relative to Pd -Cl than Sn-I to Pd-I). However in a Palladium by Palladium transmetalation the bonds that are broken and created in the transition state (Pd-X and Pd-C) are of the same nature as the bond that is created and broken (Pd-C and Pd-X), so that it is not clear why the order Cl > Br > OAc > I and the comparison with Pd/Sn transmetalation are of little value.

Reply 3:

We thank the Reviewer for these comments, as the manuscript indeed lacked a discussion of the trans effect of ligands in the transformations described. We agree that the comparison of the Pd-Sn transmetalation process with the described Pd-Pd transmetalation does not provide sufficient context because different types of bonds are broken and formed—we included the indicated sentence because we observed a similar effect of halogen ligands in palladium oxidative addition complexes on the Pd/Pd transmetalation process studied. However, we agree that the comparison is misplaced and have therefore rewrote the sentence. It should be noted that there are few such studies, so the comparison with the literature cannot be made in detail.

Observations on the effect of halide ligands in palladium oxidative addition complexes have also been made in reactions with secondary amines, where the cleavage (reaction) of tricoordinated palladium oxidation addition complexes bearing Cl, Br, and I halogen atoms has been observed, with the order of reactivity being Cl > Br > I (Organometallics **15**, 2755-2763 (1996)). The process is similar in some aspects to the proposed transmetalation pathway from Figure 9 in the manuscript, the reaction pathway of tricoordinated **7-PPh₃** with **6** to **12**, with the amine acting as a nucleophile in place of the alkyne handle of **6**.

However, since the above-mentioned Pd-Sn system and the described reaction of palladium oxidation addition complexes with amines are not fully comparable to the Pd-Pd system studied, we have summarized these references in the revised version of the manuscript as "Similar studies of halide ligands in palladium oxidation addition complexes revealed..."

In the Pd-Pd transmetalation reaction studied, similar types of bonds are indeed broken and formed, as indicated by the Reviewer, but the complexes in which these bonds are formed and broken have different substituents (in the trans position) in the key intermediate **12**, i.e. alkynyl and aryl ligands. The aryl ligand, which has a larger trans effect according to the Pt-based studies, could cause the cleavage of the halogen from **12**, leading to the formation of **B-PPh₃** and **5**. Furthermore, an irreversible reductive elimination step could also be considered to drive the process. This was additionally discussed in the manuscript.

We did not observe species **B**, *trans*-Pd(PPh₃)₂(Ar)(CCR), in the transmetalation reaction studied, and therefore it could be that after dissociation of complex **12**, the tricoordinated Pd(PPh₃)(Ar)(CCR) directly eliminates the tolane product **3** or associates with PPh₃ into short-lived *cis*-**B**, which eliminates **3**, as we indicated in the manuscript.

Comment 4:

There are some other issues, such as the speciation of Pd(0) species that is proposed, which ignores the presence of ligands (alkynes) in the solution that could stabilize the proposed $[\text{Pd}(\text{PPh}_3)_2]$ complexes, and the absence of discussion of the trans influence of ligands that could explain the relative rates of transmetalation when $[\text{Pd}(\text{alkynyl})_2(\text{PPh}_3)_2]$ and $[\text{Pd}(\text{alkynyl})\text{X}(\text{PPh}_3)_2]$ are used as transmetalation reagent.

Reply 4:

We thank the reviewer for this comment. As discussed in the previous Response to Reviewers' Comments document the stoichiometry of transmetallation reaction between **6** and **7** produces **5**, tolane product **3** and $[\text{Pd}(\text{PPh}_3)_2]$. The observed ^{31}P NMR resonance of the latter is in good agreement with the ^{31}P NMR value reported in the literature (J. Organomet. Chem. **364**, 235–244 (1989)), as we have previously discussed and indicated in the manuscript. The alkyne ligands present in the solution could potentially stabilize the proposed $[\text{Pd}(\text{PPh}_3)_2]$, which is also known from the literature (Acc. Chem. Res. **33**, 314–321 (2000), Ref. 71 in the manuscript), and which we have now additionally indicated in the revised manuscript.

We thank the Reviewer for pointing out the lack of discussion of the trans influence on the transmetallation reaction with the complexes trans- $[\text{Pd}(\text{alkynyl})\text{X}(\text{PPh}_3)_2]$ (**5, D**). It is known that ligands in square planar d^8 complexes, such as trans- $[\text{Pd}(\text{alkynyl})_2(\text{PPh}_3)_2]$ (**6, C**) and trans- $[\text{Pd}(\text{alkynyl})\text{X}(\text{PPh}_3)_2]$ (**5, D**) can significantly affect the rate of elimination of the ligand in the trans position. Strongly σ -donating and π -accepting ligands such as alkynylides accelerate the elimination of the ligand in trans position to a much greater extent than halides, which do not have similar π -accepting properties, which is probably the reason for the higher reactivity of palladium bisacetylides **C** compared to palladium monoacetylides **D**. In the revised version of the manuscript, we added the sentence of trans influence on the reactivity of **D**.

Comments from reviewer #3 regarding the points raised by reviewer #1.

Comment 1

Comment 4:

Moreover, in the provided NMR spectra related to the transmetalation reactions (S52-64), there are additional species that can be observed by ^{31}P NMR that the authors didn't assigned. What can they comment on that?

Reply 4:

The transmetallation reactions between **6** and **7** are clean for up to 1-2 hours, as shown by the enclosed ^{31}P NMR spectra. At the onset of the transmetallation reaction, only the cross-coupled product **3** is formed, together with the monoacetylides **5** and $\text{Pd}(\text{PPh}_3)_2$, as we indicated in the manuscript. Detailed analysis of the ^1H NMR spectra when monitoring transmetallation reaction confirmed that only the above-mentioned species are present at the beginning of the transmetallation reaction. After 2 hours, additional ^{31}P resonances appeared, which could be due to decomposition of complexes and side reactions between the species formed; however, these species are not relevant to the reactions

studied, so we do not assign them further. This was already stated in the manuscript and is now additionally emphasised.

This is OK, since they use initial rates. Also because the increase in the intensity of the decomposition products does not seem to be related with the reaction. That is, these signals probably are not intermediates in steady state during the creation course.

Reply 1:

We thank the Reviewer for this comment.

Comment 2

Comment 5:

3) In their previous work (Ref. 28 in the manuscript), the authors monitored the catalytic reactions at different reactions times by ³¹P NMR spectroscopy. In the NMR spectra, they observed the formation of different PdII complexes, including trans-[Pd(Ar)(CCPh)(PPh₃)₂]. This complex could be formed by the group exchange between complexes 6 and 7, and after an isomerization step, could provide the desired C-C coupling.

a. How do they explain that they observed this compound under catalytic conditions but they didn't observe in the stoichiometric experiments? They should address this discrepancy.

Reply 5:

The trans-[Pd(Ar)(CCPh)(PPh₃)₂], species B of the proposed mechanism, is indeed most likely the resting state resulting from isomerization under the specific catalytic conditions used in Nat. Commun. 9, 4814 (2018), Ref. 28 in the previous version of the manuscript, Ref. 32 in the revised manuscript, i.e., Reaction a, Pd(PPh₃)₄, NaOMe, DMF, 50 °C, as indicated in our previous work. As indicated in the reference, the high loading (20 mol%) of the PPh₃ ligand precatalyst was used to allow monitoring of the phosphorous species by ³¹P NMR. The reaction conditions described, i.e., the high loading with and excess of PPh₃, DMF, and NaOMe, were likely the reason for decreasing the rate of the reductive elimination step and allowed the isomerization to trans-[Pd(Ar)(CCPh)(PPh₃)₂], trans-B.

Authors have answered the question, they think that the trans product is formed by isomerization. However I don't agree with the authors. Very often the regiochemistry of the transmetalations produces both isomers, cis and trans in different ratios (there are examples with Zn, Au, Cu, and Sn transmetalating reagents). This is dependent on the solvent and the species involved. I think that the authors underestimate the role of the solvent and of the additives. In this regard, the use of NaOMe under catalytic conditions could lead to the formation of products of the type [PdAr(OMe)L₂], which, following the tendency observed by the authors, should transmetalate faster than the halogen-Pd derivatives, and perhaps with other regiochemistry. By the way, this could also explain the differences in the reaction rates observed for the catalytic and stoichiometric reactions which is other issue that the authors do not address very accurately, in my opinion. On the other hand, the use of an excess of PPh₃ that the authors use to explain the isomerization, should in fact make this isomerization slower, not faster. Contrarily to the authors explanation, the low loading of PPh₃ should allow the isomerization of trans-[Pd(Ar)(CCPh)(PPh₃)₂] to cis and then reductive elimination, making difficult its

observation even when the transmetalation is not regioselective. On the contrary, high loading of PPh₃ should make the isomerization difficult, allowing the observation of trans-[Pd(Ar).(CCPh)(PPh₃)₂].

Reply 2:

We thank the Reviewer for this comment because it is important. Like the Reviewer points out the presence of NaOMe in the catalytic reaction described in our earlier work, i.e., Conditions A, Pd(PPh₃)₄, NaOMe, DMF, 50 °C, could lead to the formation of more reactive palladium species, e.g., palladium oxidative addition complexes of the type Pd(PPh₃)_x(Ar)(OMe), which could react with different regioselectivity in the transmetalation step, forming directly trans-Pd(PPh₃)₂(Ar)(CCAr), and also react faster. Thus, under these conditions, transmetalation would no longer be the rate-determining step of the catalytic process, as suggested by the observed order in Pd which was found to be 1 in the catalytic reaction in the previous study under these reaction conditions (see Reply 2 to Reviewer 2 for more details and Nat. Comm. **9**, 4814 (2018)). Moreover, the presence of additional PPh₃ in the catalytic reaction, as suggested by the Reviewer, would slow down the isomerization of the formed trans-[Pd(Ar)(CCPh)(PPh₃)₂], which would be observed under these conditions.

However, we would like to emphasize that the reaction conditions for the catalytic reaction as well as the stoichiometric conditions described in the present manuscript are different from those in the previous study. The insufficient solubility of complexes **6a** and **7a** in DMF-d₇, which makes monitoring of the reaction in this solvent unsuitable, and NaOMe as an inorganic base, which makes the system heterogeneous, are not suitable for kinetic analysis. As we reported in a previous study (Nat. Commun. **9**, 4814 (2018)), the reaction conditions for catalytic reactions, i.e. Conditions A and Conditions B, were taken from the literature (Conditions A: J. Organomet. Chem. **93**, 253-257 (1975); Conditions B: ACS Catal. **2**, 135-144 (2012)) to describe the palladium species present in the reaction mixture under two different catalytic conditions in terms of catalyst precursor, base, solvent, and reaction temperature. We believe addressing the effects of NaOMe on the transmetalation step is beyond the scope of the present study. As we mentioned earlier, the conditions described in the present study include a soluble amine base, pyrrolidine, which makes the reaction mixture homogeneous and allows monitoring of the reactions. As we have already pointed out in previous Response to Reviewers' Comments document, the rate of catalytic reaction and the stoichiometric transmetalation of tricoordinated **7a'** and **6a** are in good agreement. Moreover, the order in Pd for the catalytic reaction and the combined order in **6a** and **7a** in the stoichiometric transmetalation reaction in the presence of pyrrolidine are in agreement. The fact that PPh₃ slows down the transmetalation process, indicating that the tricoordinated palladium oxidative addition complex **7a'** is the reactive species in the transmetalation reaction under the conditions studied.

Comment 3:

Comment 6:

b. Is this related to the fact that they used a different solvent to perform the kinetic study (catalysis in DMF, kinetic study in CDCl₃)?

Reply 6:

The difference most likely results from the choice of reaction conditions, i.e. solvent, base, concentration of palladium precatalyst and PPh₃.

I think it can be related, since the isomerization is usually produced in there-coordinated intermediates, coordinating solvents should affect the rate of this reaction.

Reply 3:

We agree with the Reviewer. As mentioned above, the catalytic reactions in our previous study were carried out under two different reaction conditions, Condition A and Condition B, where different Pd species were observed. In the present study, the conditions for the catalytic reaction were DCM, pyrrolidine, and Pd(PPh₃)₂X₂, and we performed stoichiometric reactions in line with these conditions. We would like to emphasize that in no reaction employing Condition B trans-Pd(PPh₃)₂(CCAr)(Ar) was observed, either in the catalytic or in the stoichiometric reaction. Please also see Reply 2.

Comment 4:

Comment 7:

c. In this regard, CHCl₃ is not used as solvent in Cu-free Sonogashira cross-couplings (they should include this recent review on the topic: RSC Adv. 2021, 11, 6885), while DMF is. Why did they select CDCl₃ for the kinetic study? Especially when in their previous work they authors showed that CDCl₃ is not innocent (they observed the formation of Pd(PPh₃)₂Cl₂ when they dissolved Pd(PPh₃)₄ in this solvent).

Reply 7:

We thank the Reviewer to indicate RSC Adv. 11, 6885-6925 (2021), we have included the reference in the manuscript. The indicated review article is more focused on preparative aspects of palladium catalysed coupling between aryl halides and terminal alkynes, including also examples of heterogenous catalysis. The choice of CDCl₃ as the solvent for monitoring the reactions was based mainly on the solubility of the Pd species studied in this solvent. The CDCl₃ was previously used as a solvent for monitoring similar reactions (Organometallics 16, 5730–5736 (1997)). The studied palladium-phosphorous species are soluble in CDCl₃ at a concentration of 0.01 M (the concentration comparable to that of Pd in catalytic reactions and also still sufficiently high to allow monitoring of the reactions by NMR). In contrast, most of the Pd species studied are not soluble at these concentrations in solvents such as DMF. Moreover, some of the palladium complexes, such as oxidative palladium addition complexes, have already been described in the literature in CDCl₃ as a solvent, which facilitates the comparison of the observed ³¹P NMR resonances with the literature reports. In our earlier paper (Nat. Commun. 9, 4814 (2018)); Ref. 28 in previous version of the manuscript, Ref. 32 in the revised manuscript), we monitored the quality of Pd(PPh₃)₄ from different manufacturers to show the importance of using the same batch of Pd(PPh₃)₄ when performing catalytic reactions, as the reproducibility of catalytic reactions has been shown to be highly dependent on batch quality. Pd(PPh₃)₄ is manufactured from Pd(PPh₃)₂Cl₂, which remains in the sample when the conversion into Pd(PPh₃)₄ is not complete. However, after some time, Pd(PPh₃)₂Cl₂ starts to form from Pd(PPh₃)₄ in CDCl₃. We have shown that by preparing fresh solutions of the same pure sample of Pd(PPh₃)₄ in different solvents (see spectra below) in benzene-d₆, THF-d₈, and in CDCl₃, showing the resonances

that corresponds to Pd(PPh₃)₄ (δ +27 ppm) along with a broadened resonance corresponding to Pd(PPh₃)₃. As we discussed in our previous paper (see page S42 in Supplementary Information in Nat. Commun. 9, 4814 (2018)), the broad resonance at +4.8 ppm was ascribed to Pd(PPh₃)₃. This has been confirmed by adding PPh₃ to the solution of Pd(PPh₃)₄, shifting the resonance at +4.8 ppm closer to -5 ppm. This observation is in agreement with the literature report (Eur. J. Org. Chem., 366-371 (2004)).

I think the selection of CDCl₃ is a bad choice. CD₂Cl₂ or THF would be much better. The kinetic behaviour of the systems after the first minutes is very poor (see for instance graphics in pg S81, S82, S87, S96, S101, S104). The authors use only the first minutes for the initial rates method, and show 4 hours in conc. / time plots in the SI. However, these 4 hours do not cover even a half-live period in most cases (see for instance graphics for 7a in S90, particularly 7c in S105, but also 7l in S93, 7m in S96, 7o in S102, or 7q in S99), so it is not possible to verify properly how the reaction evolves. I think that the poor kinetic behaviour is related to the use of CDCl₃ as solvent and also to the absence of added PPh₃ in the reaction medium, that allows the decomposition of the system. Note that most of the graphics above mentioned show that the reactions slow down with time faster than expected.

Reply 4:

As we indicated in the previous Response to Reviewers' Comments document the choice of CDCl₃ as solvent was based on the solubility of the Pd species monitored in this solvent and the literature data used to compare synthesized and observed species where CDCl₃ was also chosen as solvent. Similarly, CDCl₃ has been used to monitor similar reactions in previous studies (*Organometallics* **16**, 5730-5736 (1997)). We have shown that both the catalytic and stoichiometric reactions proceed similarly in both CH₂Cl₂ (CD₂Cl₂) and CHCl₃ (CDCl₃), which we have also indicated in the manuscript (see Reply 2 to Reviewer 2, for further details see Supplementary Information S66-78). We additionally performed the stoichiometric transmetallation reaction in THF-*d*₈ and monitored its rate and speciation. The speciation of the transmetallation reaction of **6a** and **7a** in CDCl₃ and in CH₂Cl₂ was similar, as was the rate; on the other hand, the ³¹P NMR spectrum of the transmetallation in THF-*d*₈ was slightly more complex (see Supplementary Fig. 17), with the solubility of species **6a** and **7a** also slightly lower. Moreover, THF is rarely the solvent of choice for the palladium-catalyzed coupling of aryl halides with terminal alkynes. As we mentioned earlier, complexes **6a** and **7a** are very poorly soluble in DMF-*d*₇. Therefore, despite its drawbacks, we believe that CDCl₃ has proven to be the solvent of choice for monitoring the stoichiometric reactions involving the described palladium complexes.

Typically, transmetallation reactions were monitored for up to 8 hours. However, as we have discussed previously, side species begin to form 2 hours after the onset of the reaction, and the reactions slow down, as discussed by us and also acknowledged by the Reviewer in Comments from Reviewer #3 regarding the points raised by Reviewer #1 (please see Comment 1). As we indicated in the manuscript, side species began to form in the transmetallation reaction after 2 hours that are not relevant to the transmetallation reaction under study, so we did not assign them further. For this reason, we also believe that the evolution of the reaction after 2 hours is not so relevant (however, we have provided examples of the evolution of the transmetallation reaction within a 12-hour window in Supplementary Information, see Supplementary Figure 18).

As we have already shown, the addition of PPh₃ slows down the transmetallation, which is most likely due to preventing the dissociation of PPh₃ from the 4-coordinated palladium oxidative addition

complex, as we discussed in the manuscript. The 4-coordinated palladium oxidative addition complexes **7** are the precursors of the catalytically active tricoordinated complex **7'**, as we described in the manuscript. As we have shown, the transmetalation reaction of **7a'** with **6a** is rapid and completed in 30 minutes. Thus, compared to the reaction of **7a'** and **6a**, the reaction of tetracoordinated palladium oxidative addition complex **7a** and palladium bisacetylide **6a** is indeed a transmetalation reaction in the presence of an additional PPh_3 . As we indicated in previous Response to Reviewers' Comments, after some time (2-4 h) the by-products form, causing the reaction to stall. We believe that the reaction profiles of the transmetalation reactions are as expected and produce the maximum rates at the beginning of the reaction.

Regarding the poor graphics: The diagrams on pages S81 and S82 (pages S86 and S87 in the revised Supplementary Information) describe the formation of toluene product **3** in the reaction of **7a** and palladium monoacetylides **5b** and **5c** bearing a bromide and chloride ligand, respectively. It is noteworthy that these reactions proceed very slowly and that in the diagrams on pages S81 and S82 the y-axis with the indicated concentration of toluene product **3** is shown in very small concentration. Below is an example of a graph from page S82 (S87 in the revised Supplementary Information) plotted on a "global" scale (the "zoomed" version of the graph was given to show the extraction of the initial rate from the data obtained). We have added the "global" graph along with the "zoomed" version described for these two cases.

"Zoomed" version of the graph from page S82 (S87 in the revised version of the Supplementary Information):

The graph of the same reaction on "global scale" with corrected graphics:

The other graphs given by the Reviewer (S87, S96, S101, S104, **7a** in S90, **7c** in S105, **7l** in S93, **7m** in S96, **7o** in S102, **7q** in S99, and some others) have been corrected as given below for **7m** in S96. We performed the analysis of the recorded ^1H NMR spectra for the kinetic analysis of the transmetallation reaction using the software for automatic integration of selected indicative proton resonances for substrates and products, as indicated in Supplementary Information. In a few cases, the slight shift of the x-axis (slight deviation during the automatic calibration of the spectra) of the ^1H NMR spectra resulted in a slight deviation of the integrated region in the spectra and led to incorrect concentrations. We have reviewed and corrected all spectra for similar discrepancies and included the corrected graphs in the revised versions of Supplementary Information and manuscript.

For example, see the previous graph on page S96:

And corrected graph:

It is noteworthy that these corrections did not change the results/conclusions of the manuscript.

Reviewers' comments:

Reviewer #2 (Remarks to the Author):

Dear Editor,

I thank the authors once again for the additional experiments and addressing the comments of the reviewers. Before recommending publication, I believe the following questions must be answered.

- 1) Is it feasible that this mechanism is operating under catalytic conditions?
- 2) Is there evidence that this mechanism *is* operating under catalytic conditions.
- 3) Is the bimetallic transmetalation (TM) the RDS or not.

I think question 1 has been answered with the numerous detailed stoichiometric experiments which describe the behaviour of these complexes. I think this section is fit for publication and is of interest to the organometallic chemistry community. It is quite clear that this reactivity could be operating under catalytic conditions, but whether it is the predominate mechanism, or indeed whether it is an undesired pathway is the next question.

Questions 2 and 3 are more difficult to answer, especially as this mechanism could be operating without transmetalation being the RDS, and therefore be kinetically invisible. If this mechanism is operating, and TM is the RDS then, as discussed, we expect to see 2nd order reactivity with respect to palladium. The authors report an order of 1.6 by means of a log-log plot. However, looking at the plot as a simple rate vs concentration graph (using the data that is reported in the SI, page S183) reveals that up to 8 mol%, the reaction is very clearly 1st order with respect to palladium (it does not pass through the origin, which can cause issues with log-log plots. There may be impurities preventing reaction at lower catalyst concentrations, hence no zero intercept). I have added the rate vs concentration as a separate attachment. However, from 8 mol% to 12 mol% there is a huge rate increase. This large increase is somewhat hidden when presented as a log-log plot, but as a rate vs concentration plot, this is a clear outlier in the data (the authors could also consider looking at the data by the $t(\text{cat})^n$ approach by Jordi Bures - "A Simple Graphical Method to Determine the Order in Catalyst" - <https://onlinelibrary.wiley.com/doi/10.1002/anie.201508983>)

This could indicate two regimes, where the bimetallic mechanism operates at higher loadings, and a traditional mechanism operating at lower concentrations (which could make sense as the 2nd order process slows substantially at lower concentrations). Exploring more loadings (particularly between 8 and 12 mol%, and higher) would help determine whether a change in behaviour is indeed occurring or whether 12 mol% is just an outlier. Saying this, however, the clear 1st order behaviour from 2 to 8 mol% does not agree with the bimetallic transmetalation as the RDS up to this loading. Fortunately, many of the experiments performed were at 10 mol%, so the results presented could be in the second regime – but this would have to be confirmed.

Looking at the stoichiometric experiments, I think the summation of 0.5 and 1.2 to give 1.7 is a coincidence. Looking at the data for the 0.5 order in the SI (S194), there is likely to be significant error on this value due to the spread of the data points. If there is indeed an order of 0.5, this would need further explanation, but I think more likely there is some error in the data due to the very low concentration and long reaction times (a statistical analysis of the linear regression could demonstrate this). I would recommend removing this section unless the 0.5 order can be confirmed and explained (e.g. an unreactive dimeric species?).

With the observation of 1st order behaviour in the catalyst up to 8 mol%, it is difficult to know what to recommend as this result is not consistent with the conclusions presented. There is the potential here for a more in-depth kinetic study, looking at the orders of all components in the catalytic reaction to try and determine the rate determining step. However, this may strongly depend on the loading chosen given the behaviour discussed above. The answer may be that the bimetallic mechanism is not operating under catalytic conditions in some circumstances such as at low loadings, and therefore this would have to be addressed in the manuscript.

Unfortunately, the latest result of the order in catalyst has returned more questions than answers. I recommend the authors explore the palladium loading in more detail, exploring what is happening at the higher loading and if indeed there is a shift to second order behaviour with increased loadings. It would be interesting to know whether the order in the other components (i.e. the alkyne) changes with higher loadings. If the reaction is 1st order in alkyne at low loadings, but zero at higher, it could be indicative of a shift from a rate determining binding of the alkyne to the bimetallic mechanism. My suspicion is that the system is highly complex, and a thorough mechanistic study would require a large time investment but would be a very interesting article to read. The scope could be increased for RPKA / VTNA type analysis, Hammett plots and the potential for catalyst deactivation. If this is beyond the scope, then exploring the catalyst loading and orders in reagents would be a minimum here.

Alternatively, the authors could scale back some of the conclusions, leaving things open ended for a future kinetic study. In this scenario, I would recommend the article is adapted to state that the authors have investigated the reaction by deconstructing the stoichiometric reaction into its constituent parts to build a picture of how it could behave catalytically. There is some evidence to support this (with some very interesting insights), but also some evidence to the contrary (e.g., the order with respect to palladium), and mention this is under investigation. Any mention of transmetalation as the RDS would have to be removed unless the rate equation is determined. The rate determining step of the reaction can only be determined once the orders in all of the components have been assessed. I would still suggest performing more catalyst loadings as a minimum, however.

Reviewer #3 (Remarks to the Author):

The discussion in the manuscript has been improved significantly, so I think it can be published as it is, but please, in Figure 9 the geometry of intermediate B-PPh₃ should be changed from planar trigonal to "T-shape trigonal", otherwise it should be interpreted as a Transition State in an isomerization process.

Response to Reviewers' comments regarding the manuscript entitled 'Elucidating reaction mechanism of palladium-palladium dual catalytic process through kinetic studies of proposed elementary steps' in Communications Chemistry

Reviewer #2 (Remarks to the Author):

Comment 1:

Dear Editor,

I thank the authors once again for the additional experiments and addressing the comments of the reviewers. Before recommending publication, I believe the following questions must be answered.

- 1) Is it feasible that this mechanism is operating under catalytic conditions?
- 2) Is there evidence that this mechanism *is* operating under catalytic conditions.
- 3) Is the bimetallic transmetalation (TM) the RDS or not.

Reply 1:

We thank the Reviewer for their positive comments and the issues raised, which we address point by point below and for which we have also performed additional experiments. In short, we thank the reviewer for pointing out some discrepancies within the order in palladium and suggesting additional experiments regarding kinetic analysis (question 3), as well as for re-evaluating some of the claims about RDS that we addressed by performing additional experiments and adjusting the claims in the manuscript. We discuss this in detail below. We believe that we have already provided sufficient evidence for questions (1) and (2), as we discuss in detail below.

Comment 2:

I think question 1 has been answered with the numerous detailed stoichiometric experiments which describe the behaviour of these complexes. I think this section is fit for publication and is of interest to the organometallic chemistry community. It is quite clear that this reactivity could be operating under catalytic conditions, but whether it is the predominate mechanism, or indeed whether it is an undesired pathway is the next question.

Reply 2:

We thank the Reviewer to indicate that the question 1 has been answered adequately. As noted by the Reviewer, we have attempted to demonstrate the feasibility and productivity of the proposed elementary reactions by stoichiometric reactions of independently prepared proposed reaction intermediates (or their precursors). As also noted by the Reviewer, the rates of these stoichiometric reactions, including Pd-Pd transmetalation, are comparable to the rate of the catalytic reaction when aryl halide and terminal alkyne are used as starting reagents in the presence of palladium precatalyst and base.

Furthermore, we also believe that we have shown that the proposed bimetallic mechanism is the predominant mechanism for the reaction studied. The key experiment here was to compare the feasibility (productivity) of key steps of the proposed mechanisms that should be operating for the transformation under study, the Pd-catalysed alkynylation of aryl halides, (i) the monometallic mechanism, that was previously considered as the most plausible (Fig. 1b), and the new proposed bimetallic mechanism (Fig. 1c). *We have described in detail previously (Ref. 32, Nat. Commun. 9, 4814 (2018)) and indicated in the manuscript that under identical reaction conditions the reaction of the palladium oxidative addition complex 7a with the palladium bisacetylide 6a to form the tolane product 3 is instantaneous and the tolane product is formed in a concave shaped curve with maximum rate at the beginning of the reaction. On the other hand, in the reaction of the palladium oxidative addition complex 7a with the phenylacetylene 2a in the presence of a base (pyrrolidine) (the key proposed step of the monometallic mechanism), there is an induction period at the beginning of the reaction in which palladium acetylides are formed to initiate the reaction; and when maximum rate in this reaction is reached, after the induction period, its value is almost identical as in the reaction of palladium oxidative addition complex 7a with palladium bisacetylide 6a under identical conditions (for more details, please see Fig. 5 and discussion in Nat. Commun. 9, 4814 (2018)).*

In other words, the main difference from the earlier monometallic mechanistic proposal (Fig. 1b in the manuscript), the mechanistic proposal previously stated in the literature as the most likely one, and our bimetallic mechanistic proposal (Fig. 1c), is how the alkyne reagent is transferred to the palladium oxidative addition complex. It is noteworthy that despite extensive experimental and theoretical efforts to support previous monometallic mechanism, many questions were still open and the proposed model remained unconfirmed. Incidentally, thorough computational studies revealed, for example, a high activation barrier for the formation of the η^2 -complex from the acetylene and the oxidative adduct **A** (ACS Catal. 2, 135-144 (2012); Acc. Chem. Res. 46, 2626-2634 (2013); Organometallics 35, 1036-1045 (2016). For more discussion please see Nat. Comm. 9, 4814 (2018), and references therein). As described in the previous paragraph, the reaction of palladium bisacetylide or terminal alkyne with the palladium oxidative addition complex proceeds differently under identical conditions. The reaction with palladium bisacetylide is instantaneous and produces the disubstituted alkyne product in the concave shaped curve with maximum rate at the beginning of the reaction. In contrast, in the reaction with the alkyne there is an induction period at the beginning of the reaction (in which palladium acetylides are formed, and then react under the pathway of bimetallic mechanism). Therefore this reaction, the key step in the proposed monometallic mechanism of Fig. 1b (**A** + **2**), is not (as) productive as the TM reaction with palladium bisacetylide (**A** + **C**).

Moreover, *we have previously shown that under the reaction conditions for the catalytic reaction studied (which correspond to the standard reaction conditions for this type of transformation and was taken from the literature, ACS Catal. 2, 135-144 (2012)), palladium bisacetylides are formed and are present in the reaction mixture during catalysis, as revealed by ³¹P NMR when monitoring the reaction (for more discussion please see Fig. 2 in Nat. Commun. 9, 4814 (2018)). As we describe in the present manuscript, the palladium bisacetylide is formed at a comparable rate to the rate of the catalytic reaction (this step is most likely also the RDS of the catalytic process under study, as we describe below).*

Moreover, as indicated in the manuscript, we have previously reported an improved synthetic protocol using two different palladium precatalysts simultaneously, one for the formation of palladium bisacetylide and one for the palladium oxidative addition complex (Org. Lett. 22, 4938-4943 (2020)), and others have also pointed out the plausibility of Pd-Pd transmetalation for the palladium-catalysed alkynylation of aryl halide under different reaction conditions (Chem. Commun. 55, 4973-4976 (2019);

J. Chem. Sci. **132**, 4 (2020); ChemistrySelect **5**, 2925-2934 (2020)), suggesting an operative bimetallic mechanism in the palladium-catalysed cross-coupling between aryl halides and terminal alkynes.

It follows that the Pd-Pd transmetalation reaction between palladium bisacetylides and palladium oxidative addition complexes, which we have shown to produce disubstituted alkynes at rates comparable to the catalytic reaction in stoichiometric reaction, proceeds in the catalytic reaction studied. We are not aware of any plausible mechanistic proposals for this type of transformation other than the two we have described above and in the manuscript.

We have added a short paragraph in the revised version of the manuscript that summarizes the above considerations.

Comment 3:

Questions 2 and 3 are more difficult to answer, especially as this mechanism could be operating without transmetalation being the RDS, and therefore be kinetically invisible. If this mechanism is operating, and TM is the RDS then, as discussed, we expect to see 2nd order reactivity with respect to palladium. The authors report an order of 1.6 by means of a log-log plot. However, looking at the plot as a simple rate vs concentration graph (using the data that is reported in the SI, page S183) reveals that up to 8 mol%, the reaction is very clearly 1st order with respect to palladium (it does not pass through the origin, which can cause issues with log-log plots. There may be impurities preventing reaction at lower catalyst concentrations, hence no zero intercept). I have added the rate vs concentration as a separate attachment. However, from 8 mol% to 12 mol% there is a huge rate increase. This large increase is somewhat hidden when presented as a log-log plot, but as a rate vs concentration plot, this is a clear outlier in the data (the authors could also consider looking at the data by the $t(\text{cat})^n$ approach by Jordi Bures - "A Simple Graphical Method to Determine the Order in Catalyst" - <https://onlinelibrary.wiley.com/doi/10.1002/anie.201508983>)

Reply 3:

We are very grateful to the Reviewer for pointing out the discrepancy in the log-log plot for the order in Pd, which was given as 1.6. As the Reviewer noted, there was a huge increase in rate from the point for 8 mol% to 12 mol%, which was in fact due to an experimental error in determining the rate for the reaction with 12 mol% Pd catalyst loading. We included 3 additional concentration points for the Pd precatalyst (1, 10, 15 mol%) in the corrected plot on page S193 (Supplementary Figure 41) of the revised version of the Supplementary Information. The corrected analysis showed the order in Pd 1 for all data points included.

In addition to the above correction, we have also thoroughly reviewed all the other kinetic experiments and redone some of them, as well as rewritten the part of the kinetic analysis, as we describe below.

As suggested by the Reviewer, we calculated the order in the catalyst using the $t(\text{cat})^n$ approach according to Angew. Chem. Int. Ed. **55**, 16084-16087 (2016). The analysis resulted the order in Pd 1.2 and is included on page S203 in the revised version of the Supplementary Information. We also used this approach to determine order in other reagents, as we explain below in more detail, which is described on pages S203-S210 in the revised version of the Supplementary Information.

If the transmetalation is not an RDS and the order in Pd is not 2, then the mechanistic analysis under synthetically relevant conditions will overlook (not consider) the transmetalation step as a plausible

step in the reaction, as the Reviewer has indicated, and as we have also indicated in the manuscript. We discuss this in more detail below and also consider this to be one of the key messages of the manuscript.

Comment 4:

This could indicate two regimes, where the bimetallic mechanism operates at higher loadings, and a traditional mechanism operating at lower concentrations (which could make sense as the 2nd order process slows substantially at lower concentrations). Exploring more loadings (particularly between 8 and 12 mol%, and higher) would help determine whether a change in behaviour is indeed occurring or whether 12 mol% is just an outlier. Saying this, however, the clear 1st order behaviour from 2 to 8 mol% does not agree with the bimetallic transmetalation as the RDS up to this loading. Fortunately, many of the experiments performed were at 10 mol%, so the results presented could be in the second regime – but this would have to be confirmed.

Reply 4: Please see Reply 3 above.

Comment 5:

Looking at the stoichiometric experiments, I think the summation of 0.5 and 1.2 to give 1.7 is a coincidence. Looking at the data for the 0.5 order in the SI (S194), there is likely to be significant error on this value due to the spread of the data points. If there is indeed an order of 0.5, this would need further explanation, but I think more likely there is some error in the data due to the very low concentration and long reaction times (a statistical analysis of the linear regression could demonstrate this). I would recommend removing this section unless the 0.5 order can be confirmed and explained (e.g. an unreactive dimeric species?).

Reply 5:

In the first part of the manuscript, we present the analysis of the transmetallation reaction of **6a** and **7a** to determine the order in each reagent, as well as several other experiments to gain insights into this process that have not been described previously. To simplify the analysis, it was performed without any additives.

In the catalytic reaction studied, pyrrolidine is present as a base that can also act as a coordinating ligand, as we describe in the manuscript. Pyrrolidine can form equilibria with several Pd species proposed as intermediates. We have indicated in the manuscript, and was also reported by others (Chem. Eur. J. **13**, 666-676 (2007); Nat. Commun. **9**, 4814 (2018); J. Org. Chem. **71**, 1677-1687 (2006)) that the addition of pyrrolidine to the palladium oxidative addition complex leads to the Pd(PPh₃)(Pyr)(Ar)(X) complex **7a'(pyr)**. We have shown that tricoordinated palladium oxidative addition complex Pd(Ar)(L)(X) is a reactive species in the transmetalation step. Therefore, the Pd(PPh₃)(Pyr)(Ar)(X) **7a'(pyr)** is indeed not reactive toward the reaction with palladium bisacetylides.

Interestingly, on the other hand, pyrrolidine does not interact with palladium bisacetylde **6a**.

Therefore, we hypothesised that the presence of pyrrolidine might affect the order of **7a** and **6a** in the transmetalation described above, and thus performed an analysis of the order in **7a** and **6a** in the transmetalation reaction in the presence of pyrrolidine.

Analysis of the orders in **7a** and **6a** in the presence of pyrrolidine, pages S216-221 in the Supplementary Information, revealed order 1 in the palladium bisacetylide and 0.5 in the palladium oxidation complex. The order in the palladium oxidative addition complex **7a** was determined at four different concentrations (pages S219-220 in the revised version of the Supplementary Information), and the second-degree polynomial function describes well the concentrations obtained over time, and the R² value is above 99% for all four examples. Moreover, the ln/ln plot for order in **7a** (page S221) is 0.54 with an R² value of 0.89.

Therefore, we believe that this observation is not due to experimental error. It could be that when there is a large excess of pyrrolidine over the palladium oxidative addition complex, which can occur under catalytic reaction conditions, the order of **7a** is limited toward 0, making it even more difficult to observe transmetallation as a rate-determining step by traditional methods of kinetic analysis under synthetically relevant conditions (if transmetallation would be the rate-determining step of the process).

However, since the revised order in Pd turned out to be 1 for the catalytic reaction, we removed this section (discussion) from the manuscript as suggested by the Reviewer. However, we left the observation of the orders in the transmetallation reaction in the presence of pyrrolidine in the Supplementary Information and included the above note in the revised version of the manuscript.

Comment 6:

With the observation of 1st order behaviour in the catalyst up to 8 mol%, it is difficult to know what to recommend as this result is not consistent with the conclusions presented. There is the potential here for a more in-depth kinetic study, looking at the orders of all components in the catalytic reaction to try and determine the rate determining step. However, this may strongly depend on the loading chosen given the behaviour discussed above. The answer may be that the bimetallic mechanism is not operating under catalytic conditions in some circumstances such as at low loadings, and therefore this would have to be addressed in the manuscript.

Reply 6:

Here we will start with the statement from the manuscript that *"It should be noted that transmetallation need not be the rate-determining step to assume that it occurs in the process, since, for example, oxidative addition was found to be the rate-determining step in the Pd-Cu Sonogashira reaction studied, which does not preclude Pd-Cu transmetallation from proceeding in the reaction."*

The error in determining the Pd order, as discussed in Reply 3, led to an incorrect conclusion regarding the RDS of the process, and we are again grateful to the Reviewer for raising the issues discussed above. However, we do not believe that the mechanism changes with catalyst loading and disagree with the Reviewer's statement that the bimetallic mechanism does not operate when (if) the order in Pd is 1. As indicated in the first paragraph of this Reply, the oxidative addition step was found to be RDS in the Pd-Cu Sonogashira reaction, but this does not change the conclusion that the Pd-Cu transmetallation step is operating in this reaction. We have revised the discussion section on RDS in the revised version of the manuscript; it turned out that RDS is most likely the formation of palladium bisacetylides (please see Reply 7). In addition, as discussed in previous Response to Reviewers' Comments documents we have left room in the manuscript conclusion for other potential mechanisms that could operate in this transformation under different reaction conditions, i.e., "It should be noted

that under different conditions (e.g., different ligands and reaction conditions), other mechanisms could compete with or replace the one described here."

Comment 7:

Unfortunately, the latest result of the order in catalyst has returned more questions than answers. I recommend the authors explore the palladium loading in more detail, exploring what is happening at the higher loading and if indeed there is a shift to second order behaviour with increased loadings. It would be interesting to know whether the order in the other components (i.e. the alkyne) changes with higher loadings. If the reaction is 1st order in alkyne at low loadings, but zero at higher, it could be indicative of a shift from a rate determining binding of the alkyne to the bimetallic mechanism. My suspicion is that the system is highly complex, and a thorough mechanistic study would require a large time investment but would be a very interesting article to read. The scope could be increased for RPKA / VTNA type analysis, Hammett plots and the potential for catalyst deactivation. If this is beyond the scope, then exploring the catalyst loading and orders in reagents would be a minimum here.

Reply 7:

We thank the Reviewer for this comment and, as requested, have performed additional experiments to determine the order in the reagents by kinetic analysis under synthetically relevant conditions and have also performed VTNA analysis.

In summary, the kinetic analysis under synthetically relevant conditions obtained from the slopes of the linear functions of the \ln/\ln plots of the maximum rates obtained by derivation of the sigmoid curves from the fit of the experimental data, against the corresponding concentration for each reagent and catalyst, gave the order in Pd 1.1, -0.15 for aryl iodide, 0.3 for phenylacetylene, and 1.1 for pyrrolidine (pages S184-S202 in the revised version of the Supplementary Information). We also performed variable time normalisation analyses (VTNA) to determine the order in the catalyst and reagents. These gave an order of 1.2 in Pd, -0.08 in aryl iodide, 0.37 in phenylacetylene, and 1.1 in pyrrolidine (pages S203-S210 in the revised version of the Supplementary Information).

The orders obtained, the order 1 in Pd, 0 in aryl halide, 0.35 in alkyne, 1 in base, suggest that one of the transition states within the multistep process of formation of palladium bisacetylide **6a** from **5a** is most likely the rate-determining step of the catalytic reaction studied.

In light of this result, we revisited the rates of the elementary steps in Table 1. We had previously assumed that transmetallation was the rate-determining step of the catalytic process studied, since we had observed the lowest rate for this step when comparing the rates of elementary reactions under similar reaction conditions. Moreover, this seemed reasonable given the previously obtained order in Pd, which was higher than 1. We reexamined the transmetallation reaction and (as we have previously indicated in the Supplementary Information) included the notion to the revised version of the manuscript: "While the transmetallation between **6a** and **7a'** was the slowest of the reactions studied (Table 1, entry 2), due to the instability of **7a'**, i.e., aryl group scrambling and side reactions, (J. Am. Chem. Soc. **16**, 6313-6315 (1991); J. Am. Chem. Soc. **33**, 8576-8581 (1995)), a lower than potential rate was observed during the reaction; otherwise, the rate would likely be higher than that observed for the formation of **6a** from the described **5a**."

It is noteworthy that the presented kinetic analysis, by comparing the rates of elementary steps under similar reaction conditions, indicated the above-described palladium bisacetylide formation **6a** from

5a as the rate-determining step of the process, with a very similar value of the rate as in the catalytic reaction (Table 1).

In light of the above, we have rewritten the discussion in the manuscript considering the determination of RDS in the relevant parts of the manuscript.

Comment 8:

Alternatively, the authors could scale back some of the conclusions, leaving things open ended for a future kinetic study. In this scenario, I would recommend the article is adapted to state that the authors have investigated the reaction by deconstructing the stoichiometric reaction into its constituent parts to build a picture of how it could behave catalytically. There is some evidence to support this (with some very interesting insights), but also some evidence to the contrary (e.g., the order with respect to palladium), and mention this is under investigation. Any mention of transmetalation as the RDS would have to be removed unless the rate equation is determined. The rate determining step of the reaction can only be determined once the orders in all of the components have been assessed. I would still suggest performing more catalyst loadings as a minimum, however.

Reply 8:

Please see Reply 7 for discussion about kinetic analysis under synthetically relevant conditions. We thank the Reviewer for these constructive comments as they have improved the manuscript substantially.

As suggested by the Reviewer, we have removed the statements about transmetalation as RDS of the reaction, also in the part about halogen metathesis. We have rewritten this part of the manuscript, focusing more on the importance of careful selection of precatalysts when they are in the form of halides, and on the effect of the base that can act as a ligand. It is noteworthy that when studying the kinetics of such a reaction under synthetically relevant conditions, halide metathesis may occur in situ, which may affect the structure of key mechanistic intermediates and the rates of elementary reactions of the mechanism. There are reports in the literature analysing the kinetics of such a reaction using, for example, Pd(PPh₃)₂Cl₂ for the reaction of aryl iodides as starting substrates (see, e.g., ACS Catal. **2**, 135-144 (2012)). We have included this notion in the revised version of the manuscript not mentioning the specific example nor the references.

In conclusion, it should be pointed out once again that a mechanistic analysis under synthetically relevant conditions cannot provide insight into the reaction pathway of the monometallic dual catalytic process if the transmetalation is not RDS (as in the case of the catalytic system under study). On the other hand, the disassembly approach described here can easily determine the feasibility of the proposed elementary steps and the interconnection of the intermediates, and give indications of possible RDS. We believe that we have provided sufficient evidence that the studied reaction indeed proceeds via a Pd-Pd transmetalation step and that the described approach is useful for such a mechanistic analysis.

Reviewer #3 (Remarks to the Author):

Comment 1:

The discussion in the manuscript has been improved significantly, so I think it can be published as it is, but please, in Figure 9 the geometry of intermediate B-PPh₃ should be changed from planar trigonal to "T-shape trigonal", otherwise it should be interpreted as a Transition State in an isomerization process.

Reply 1:

We thank the Reviewer for their positive comments. We have changed the structure of intermediate **B-PPh₃** from planar to "T-shape trigonal" in Figure 9.